# Efficient Perplexity Bound and Ratio Matching in Discrete Diffusion Language Models

**Etrit Haxholli**     **Yeti Z. Gürbüz**     **Oğul Can**     **Eli Waxman**
MetaDialog Research
{etrith, yeti, ogulc, elib}@metadialog.com

## Abstract

While continuous diffusion models excel in modeling continuous distributions, their application to categorical data has been less effective. Recent work has shown that ratio-matching through *score-entropy* within a continuous-time discrete Markov chain (CTMC) framework serves as a competitive alternative to autoregressive models in language modeling. To enhance this framework, we first introduce three new theorems concerning the KL divergence between the data and learned distribution. Our results serve as the discrete counterpart to those established for continuous diffusion models and allow us to derive an improved upper bound of the perplexity. Second, we empirically show that ratio-matching performed by minimizing the *denoising cross-entropy* between the clean and corrupted data enables models to outperform those utilizing score-entropy with up to 10% lower perplexity/generative-perplexity, and 15% faster training steps. To further support our findings, we introduce and evaluate a novel CTMC transition-rate matrix that allows prediction refinement, and derive the analytic expression for its matrix exponential which facilitates the computation of conditional ratios thus enabling efficient training and generation.

## 1 Introduction

Modeling data distributions is a fundamental task in machine learning. In the case of continuous data distributions, recent advancements in continuous diffusion models (Sohl-Dickstein et al., 2015; Ho et al., 2020; Song et al., 2020b) have demonstrated impressive capabilities in generating data samples and performing density estimation (Song et al., 2021a;c; Haxholli & Lorenzi, 2023; Kingma et al., 2021a). Despite these achievements, the application of such models to categorical data distributions, like language, remains limited, as continuous diffusion models generally underperform compared to autoregressive models in these scenarios (Chen et al., 2023; Gulrajani & Hashimoto, 2024; Li et al., 2022; Dieleman et al., 2022; Strudel et al., 2022). To address this, recent research has focused on the development of discrete diffusion models (Austin et al., 2021; Campbell et al., 2022; Meng et al., 2022; Lou et al., 2024; Sahoo et al., 2024; Shi et al., 2024; Ou et al., 2025) which offer distinct advantages compared to autoregressive models, such as the ability to infill various parts of a sequence non-sequentially and have the potential to reduce computing time and expenses in generating lengthy sequences (Deschenaux & Gulcehre, 2024; Christopher et al., 2024).

Evaluating discrete diffusion models, however, presents a practical challenge due to the difficulty in calculating the perplexity, unlike in autoregressive models where this computation is straightforward. Although a recent perplexity bound has been proposed by Lou et al. (2024), no tightness guarantees exist. In this paper, we present three theorems concerning the Kullback-Leibler (KL) divergence between the data distribution and the learned distribution in discrete diffusion models (Lou et al., 2024). These results serve as the discrete analogue of the continuous diffusion theorems provided in Song et al. (2021b). One of our key contributions is Theorem 4, which provides an upper bound ($J_2$) on the cross-entropy between the data and learned distributions, offering a more direct way of bounding the perplexity. This bound is computationally more efficient than the existing bound in Lou et al. (2024), and empirical results suggest that it is also slightly tighter.

---

Code is available at: https://github.com/MetaDialog-Research/PBRC

In addition, inspired by model reparametrizations (Ho et al., 2020; Karras et al., 2022; Lou et al., 2024), this paper examines the ratio-matching training objective SEDD in Lou et al. (2024). We highlight that the sole unknowns implicitly learned by the model are the per-token marginal probabilities across the vocabulary, conditioned on the current perturbed sequence. Consequently, rather than modeling the ratios directly, we employ a weighted version of the denoising cross-entropy loss $L_{ll}$ proposed in Campbell et al. (2022), which also mirrors the cross-entropy loss utilized in continuous diffusion models, as described in Dieleman et al. (2022). We show empirically that by modifying the reconstruction of the scores, training with cross-entropy outperforms direct ratio matching for all types of tested discrete diffusion dynamics. We name this strategy of using cross-entropy for training and the adjusted ratio reconstruction for generation, cross-entropy discrete diffusion (CEDD). Similar advantageous results when utilizing the cross-entropy loss have been reported in studies of *absorb* discrete diffusion, as shown in concurrent research by Sahoo et al. (2024); Shi et al. (2024); Ou et al. (2025). These particular results, up to a scaling factor, are specific cases within the broader CEDD framework. CEDD improves upon SEDD by circumventing the learning of conditional ratios—which are analytically determinable— focusing instead on learning the mixing weights that constitute the necessary marginal ratios for generation. This focus not only conserves modeling resources but is also particularly advantageous when the distribution of conditional ratios is complex.

To illustrate our point, we design a new transition-rate matrix named *roulette* diffusion, and derive its matrix exponential. The roulette diffusion is an interpolation between the absorb and the uniform diffusion. In the forward process, a token can transition to any state until it hits the absorb state, that is, until it is masked. In return, the reverse process begins with a sequence of masked tokens, which are gradually unmasked, and where the unmasked tokens can be refined. Intuitively. this capability should be important for discrete diffusion models (Deschenaux & Gulcehre, 2024), and can be useful for downstream tasks, as shown in our spelling correction experiments. Moreover, during the reverse process, the scores/ratios of unmasked tokens have much larger magnitudes than those of masked tokens, posing a significant learning challenge for the network due to output scale variability. Employing the CEDD strategy mitigates this challenge as demonstrated experimentally.

In summary, the main **contributions** of this paper include:

- Providing 3 new theorems concerning the KL divergence and cross entropy between the data and the learned distribution. Improving model evaluation through the bound provided in Theorem 4.
- Introducing a new transition-rate matrix (roulette diffusion) that allows token correction after unmasking in the reverse process. Deriving its matrix exponential which enables efficient training using SEDD and generation when CEDD is employed.
- Comparing the performance of SEDD and CEDD experimentally in the task of language modelling on absorb, uniform and roulette diffusion models. Showing that CEDD outperforms SEDD in all cases in terms of perplexity.

## 2 PRELIMINARIES AND NOTATION

### 2.1 MARKOV CHAINS OVER FINITE-STATE SPACES

A discrete-time Markov Chain in a finite-state space is a stochastic process $X_1, X_2, \ldots, X_{\bar{T}}$, where each state $X_t$ depends solely on the preceding one. The states $X_t$ can take values from $\{1, 2, \ldots, S\}$, and $\bar{T}$ represents the number of time steps. The probability of being in state $x$ at time $t$ is

$$p_t(X_t = x) = \sum_{y=1}^{S} p_t(X_t = x, X_{t-1} = y) = \sum_{y=1}^{S} p_{t|t-1}(X_t = x | X_{t-1} = y) p_{t-1}(X_{t-1} = y). \quad (1)$$

Placing all such probabilities $p_t(X_t = x)$ in a vector $\boldsymbol{s}_t$, such that $\boldsymbol{s}_t(x) = p_t(X_t = x)$, gives

$$\boldsymbol{s}_t = \boldsymbol{P}\boldsymbol{s}_{t-1}, \text{ where } \boldsymbol{P}(x, y) = p_{t|t-1}(X_t = x | X_{t-1} = y). \quad (2)$$

One can generalize such processes into Continuous Time Markov Chains (CTMCs) where $t \in [0, \bar{T}]$, (Anderson, 2012). For simplicity, we make the choice $\bar{T} = 1$. To construct a CTMC, one first chooses a transition-rate matrix $\boldsymbol{Q}_t$, which has the property that its non-diagonal elements are non-negative, and the elements in each of its columns add to zero (Suhov & Kelbert, 2008). Given an

initial probability distribution $s_0$, the equation below fully determines the evolution of the probability with respect to time:

$$\frac{ds_t}{dt} = Q_t s_t. \tag{3}$$

In addition, we choose $Q_t = \sigma'(t)Q$, where $Q$ is itself a constant transition-rate matrix and where function $\sigma$ is monotonically increasing, and satisfies $\sigma(0) = 0$ as well as $\lim_{t \to 1} \sigma(t) = T$. In this setting, the distribution over states at time $t$ is the solution of the linear ODE in Equation (3), that is, $s_t = e^{\sigma(t)Q} s_0$.

Matrices $Q_t$ are chosen such that: a) the matrix exponential $e^{\sigma(t)Q}$ is easy to calculate, which is essential as $p_{t|0}(x|y) = e^{\sigma(t)Q}(x, y)$; and b) $s_1$ is an easy reference distribution to sample from (Austin et al., 2021; Campbell et al., 2022).

Finally, similar to diffusion processes in continuous spaces, the continuous-time Markov chain in Equation (3) also admits a reverse process (Kelly, 1979; Sun et al., 2023):

$$\frac{ds_{1-t}}{dt} = \bar{Q}_{1-t} s_{1-t}, \tag{4}$$

where $\bar{Q}_t(x, y) = Q_t(y, x)\frac{p_t(x)}{p_t(y)}$ for $x \neq y$, and $\bar{Q}_t(x, x) = -\sum_{y \neq x} \bar{Q}_t(y, x)$. Since we can easily sample from the reference distribution, the only unknowns preventing us from being able to run backwards are the ratios $\frac{p_t(x)}{p_t(y)}$ also known as concrete scores (Meng et al., 2022; Lou et al., 2024). Once such ratios are modeled using a neural network, we can generate samples from the learned data distribution $p_0^\theta$ by discretizing Equation (4) as follows:

$$p(x_{t-\epsilon} = y \mid x_t = x) = \delta_x(y) + \bar{Q}_t(y, x)\epsilon + O(\epsilon^2). \tag{5}$$

Additional details are provided in Appendix D.

## 2.2 SEDD: Estimating the Ratios via Score Entropy

As pointed out in the previous subsection, we wish to model the ratios $\frac{p_t(y)}{p_t(x_t)}$ via a neural network $s_\theta(x_t, t)_y$, for example by minimizing the score entropy loss (Lou et al., 2024):

$$\mathbb{E}_{x_t \sim p_t} \sum_{y \neq x_t} w_{x_t, y} \ell\left(\frac{p_t(y)}{p_t(x_t)}, s_\theta(x_t, t)_y\right), \text{ for } \ell(a, b) = (b - a \log b + K(a)), \tag{6}$$

and $K(a) = a(\log a - 1)$. In Lou et al. (2024), $w_{x_t, y} = Q_t(x_t, y)$, and furthermore they show that an equivalent loss is the following:

$$\mathbb{E}_{x_0 \sim p_0, x_t \sim p_{t|0}(\cdot|x_0)} \sum_{y \neq x_t} w_{x_t, y} \ell\left(\frac{p_{t|0}(y|x_0)}{p_{t|0}(x_t|x_0)}, s_\theta(x_t, t)_y\right), \tag{7}$$

which side-steps the problem of not knowing the marginal ratios $\frac{p_t(y)}{p_t(x_t)}$, by employing $p_{t|0}(i|j) = e^{\sigma(t)Q}(i, j)$. A more detailed derivation of Equation (7) can be found in Appendix A.2.

## 2.3 Discrete Diffusion for Language Modeling - Token Level Transitions

In the case of Language Modeling, we write a sequence of length $L$ from the data distribution as $x_0$, where $x_0 = (x_0^1, x_0^2, ..., x_0^L)$ and $x_0^i \in \text{Vocab} = \{1, 2, ..., V\}$. The number of possible sequences, that is, the number of states $S$ is $V^L$. Unfortunately, this implies that it is not computationally feasible to model the ratios of probabilities between the current state $x_t$ and all other states $y$, since the output of our neural network would have to be $V^L$ dimensional (Campbell et al., 2022).

We follow the usual approach (Campbell et al., 2022; Lou et al., 2024) to mitigate this issue, which is to select a sparse matrix $Q_t(S \times S)$, such that each entry $Q_t(x, y)$ for two sequences $x, y$ that differ in more than one token will be zero. The forward process that such a $Q_t$ defines, can equivalently described as follows: at each discretized step, only one uniformly randomly chosen token from the current sequence can be modified, according to a token level forward diffusion process $Q_t^{tok}(V \times V)$. More formally, for $x = (x_0^1, ..., x^i, ..., x_0^L)$ and $y = (x_0^1, ..., \hat{x}^i, ..., x_0^L)$, if $x^i \neq \hat{x}^i$

we have $\boldsymbol{Q}_t(\boldsymbol{x}, \boldsymbol{y}) = \boldsymbol{Q}_t^{tok}(x^i, y^i)$, otherwise $\boldsymbol{Q}_t$ is zero in other non-diagonal entries. For such a sparse choice of $\boldsymbol{Q}_t$, and $\boldsymbol{y}$ which only differs from $\boldsymbol{x}$ at a single position $i$, Expression (5) becomes

$$
p(x_{t-\epsilon} = \boldsymbol{y} \mid x_t = \boldsymbol{x}) = \begin{cases} 1 - \sum_{z^i \in \text{Vocab} \setminus x^i} \boldsymbol{Q}_t^{tok}(x^i, z^i) \frac{p_t(\boldsymbol{z})}{p_t(\boldsymbol{x})} \epsilon + O(\epsilon^2). & \text{if } \boldsymbol{y} = \boldsymbol{x} \\ \boldsymbol{Q}_t^{tok}(x^i, y^i) \frac{p_t(\boldsymbol{y})}{p_t(\boldsymbol{x})} \epsilon + O(\epsilon^2). & \text{if } \boldsymbol{y} \neq \boldsymbol{x} \end{cases} \tag{8}
$$

where $\boldsymbol{z}$ denotes a sequence that is identical to $\boldsymbol{x}$ everywhere, but position $i$. Thus, the usual approach entails only modeling the ratios between $\boldsymbol{x}$ and 'neighbours' $\boldsymbol{y}$ which only differ from $\boldsymbol{x}$ by one token. The number of such neighbours is $L \times V$, that is $V$ per each of the $L$ positions, hence the output of the network is $L \times V$ coinciding with that of transformers in autoregressive language models. It should be pointed out that one can indeed use Expression (7) for training, due to the fact that tokens are perturbed independently from one another in the forward process $p_{t|0}(\boldsymbol{x}_t|\boldsymbol{x}_0) = \prod_j p_{t|0}(x_t^j|x_0^j)$, and thus

$$
\frac{p_{t|0}(\boldsymbol{y}|\boldsymbol{x}_0)}{p_{t|0}(\boldsymbol{x}_t|\boldsymbol{x}_0)} = \frac{\prod p_{t|0}(y^j|x_0^j)}{\prod p_{t|0}(x_t^j|x_0^j)} = \prod \frac{p_{t|0}(y^j|x_0^j)}{p_{t|0}(x_t^j|x_0^j)} = \frac{p_{t|0}(y^i|x_0^i)}{p_{t|0}(x_t^i|x_0^i)} = \frac{e^{\boldsymbol{Q}_t^{tok}}(y^i, x_0^i)}{e^{\boldsymbol{Q}_t^{tok}}(x_t^i, x_0^i)}, \tag{9}
$$

Finally, the noise schedule $\sigma_t$ is typically loglinear $-\log(1 - (1 - \epsilon)t)$ or geometric $\sigma_{min}^{1-t} \cdot \sigma_{max}^t$.

## 3 METHODOLOGY AND THEORETICAL RESULTS

In Subsection 3.1, we provide results related to the cross entropy and the KL divergence between the data and the learned distribution in the CTMC (discrete diffusion) framework. The first three theorems therein can be considered as the discrete diffusion analog of the ones given in (Song et al., 2021b). Importantly, Theorem 4 provides an upper bound ($J_2$) on the cross entropy between the data and learned distribution which can be used to bound the perplexity, and which does not depend on the function $K$ (Equation 6). We emphasize that the results hold for general CTMCs, and not only in the special case of token-level transitions. From the second subsection onwards, we operate in the token-level transition framework. More precisely, in Subsection 3.2, we introduce the roulette transition-rate matrix, and provide an expression for its exponential. In Subsection 3.3, we state Proposition 6, which enables a more efficient estimation of $J_2$. In Subsection 3.4, we highlight that similarly to the continuous case (Dieleman et al., 2022), the ratios can be modeled using $L_{ll}$ from Campbell et al. (2022), and present how this approach is adapted in our experimental setup.

### 3.1 CROSS ENTROPY AND KL DIVERGENCE RESULTS

We begin by finding an upper bound for the KL divergence between the data and the learned distribution. The proofs are provided in Appendix A.1.

**Theorem 1.** *Define a CTMC with transition matrix $\boldsymbol{Q}_t$ that runs from time $0$ to $1$. The true reverse process defines a probability evolution $p_t$ from $p_1$ to the data distribution $p_0$, while the learned reverse process induces the evolution $p_t^\theta$ from the reference distribution $p_1^\theta = p_r$ to the approximation of the data distribution $p_0^\theta$. In this setting, the following KL divergence bound holds*

$$
D_{KL}(p_0||p_0^\theta) \leq \int_0^1 \mathbb{E}_{x_t \sim p_t} \sum_{y \neq x_t} \boldsymbol{Q}_t(x_t, y) \ell \left( \frac{p_t(y)}{p_t(x_t)}, s_\theta(x_t, t)_y \right) dt + D_{KL}(p_1||p_r). \tag{10}
$$

*where $\ell(a, b) = (b - a \log b + K(a))$ and $K(a) = a(\log a - 1)$.*

The following theorem provides an expression for the entropy of the data distribution. Furthermore, it provides sufficient conditions for the bound given above to become tight.

**Theorem 2.** *Denote the intermediate distributions at time $t$ determined by the true reverse process, and by the learned reverse process with $p_t$ and $p_t^\theta$, respectively. We can write the entropy of the data distribution $H(p_0)$ as*

$$
H(p_0) = H(p_1) - \int_0^1 \mathbb{E}_{x_t \sim p_t} \sum_y \boldsymbol{Q}_t(x_t, y) K \left( \frac{p_t(y)}{p_t(x_t)} \right) dt. \tag{11}
$$

*In addition, if the learned ratios $s_\theta(x_t, t)_y$ equal $\frac{p_t^\theta(y)}{p_t^\theta(x_t)}$ and $p_1 = p_1^\theta := p_r$, then the inequality in Theorem 1, becomes an equality.*

A particular case where the conditions of the theorem above hold is when $s_\theta(x_t, t)_y = \frac{p_t(y)}{p_t(x_t)}$ as then $\frac{p_t^\theta(y)}{p_t^\theta(x_t)} = \frac{p_t(y)}{p_t(x_t)} = s_\theta(x_t, t)_y$. The third theorem gives an upper bound of the negative log-likelihood at a single point. This is a central result in Lou et al. (2024, Theorem 3.6), but we restate it here for completeness, and provide an alternative, more detailed proof in Appendix A.1.

**Theorem 3.** *Let $p_0^\theta$ denote the learned distribution from which the reverse process samples. The negative log-probability of a state $x_0$ being sampled by the reverse process can be bounded from above as follows,*

$$-\log p_0^\theta(x_0) \le \int_0^1 \mathbb{E}_{x_t \sim p_{t|0}(\cdot|x_0)} \sum_{y \neq x_t} \boldsymbol{Q}_t(x_t, y)\ell\left(\frac{p_{t|0}(y|x_0)}{p_{t|0}(x_t|x_0)}, s_\theta(x_t, t)_y\right) dt$$

$$+ D_{KL}(p_{1|0}(\cdot|x_0)\|p_r). \tag{12}$$

Since the noise schedule in (Lou et al., 2024; Ou et al., 2025) is chosen such that $p_1 \approx p_r$ and thus $D_{KL}(p_{1|0}\|p_r) \approx 0$, one can take the expectation $\mathbb{E}_{x_0}$ on both sides of Inequality (12) to get a bound on the cross entropy $\frac{1}{L}H(p_0, p_0^\theta) = \mathbb{E}_{x_0}[-\frac{1}{L}\log(p_0^\theta(x_0))]$. One approach is to compute the RHS in Expression (12) for each point $x_0$ and then average results (Appendix B.5), which can be computationally expensive. Instead, we can divide by $L$ and take the expectation with regards to data distribution on both sides of Expression (12) as in Ou et al. (2025), and calculate

$$J_1 = \frac{1}{L}\mathbb{E}_{t \sim U(0,1)}\mathbb{E}_{x_0 \sim p_0(x_0)}\mathbb{E}_{x_t \sim p_{t|0}(\cdot|x_0)} \sum_{y \neq x_t} \boldsymbol{Q}_t(x_t, y)\ell\left(\frac{p_{t|0}(y|x_0)}{p_{t|0}(x_t|x_0)}, s_\theta(x_t, t)_y\right). \tag{13}$$

Using Theorem 1 and 2, we provide another, direct upper bound on the cross-entropy between the data and learned distributions, which evades the computation of $K$.

**Theorem 4.** *Under the conditions stated in Theorem 1, the following inequality for the cross entropy between the data and the learned distribution holds:*

$$H(p_0, p_0^\theta) \le \int_0^1 \mathbb{E}_{x_t \sim p_t} \sum_{y \neq x_t} \boldsymbol{Q}_t(x_t, y)\bar{\ell}\left(\frac{p_t(y)}{p_t(x_t)}, s_\theta(x_t, t)_y\right) dt$$

$$- \int_0^1 \mathbb{E}_{x_t \sim p_t} \sum_{y \neq x_t} \boldsymbol{Q}_t(y, x_t)dt + H(p_1, p_r), \text{ where } \bar{\ell}(a,b) = (b - a\log b). \tag{14}$$

The second term $-\int_0^1 \mathbb{E}_{x_t \sim p_t} \sum_{y \neq x_t} \boldsymbol{Q}_t(y, x_t)dt$ and third one $H(p_1, p_r) \approx H(p_r)$ can be analytically computed as shown in Section 3.3, Proposition 6. Finally, the first term can be rewritten as

$$\mathbb{E}_{t \sim U(0,1)}\mathbb{E}_{x_0 \sim p_0(x_0)}\mathbb{E}_{x_t \sim p_{t|0}(\cdot|x_0)} \sum_{y \neq x_t} \boldsymbol{Q}_t(x_t, y)\bar{\ell}\left(\frac{p_{t|0}(y|x_0)}{p_{t|0}(x_t|x_0)}, s_\theta(x_t, t)_y\right). \tag{15}$$

Therefore, due to Theorem 4, we instead propose to use

$$J_2 = \frac{1}{L}\left[\mathbb{E}_{t \sim U(0,1)}\mathbb{E}_{x_0 \sim p_0(x_0)}\mathbb{E}_{x_t \sim p_{t|0}(\cdot|x_0)} \sum_{y \neq x_t} \boldsymbol{Q}_t(x_t, y)\bar{\ell}\left(\frac{p_{t|0}(y|x_0)}{p_{t|0}(x_t|x_0)}, s_\theta(x_t, t)_y\right)\right.$$

$$\left. + H(p_r) - \int_0^1 \mathbb{E}_{x_t \sim p_t} \sum_{y \neq x_t} \boldsymbol{Q}_t(y, x_t)dt\right]. \tag{16}$$

In both cases $Perplexity = \exp(\mathbb{E}_{x_0}[-\frac{1}{L}\log(p_0^\theta(x_0))]) = \exp(\frac{1}{L}H(p_0, p_0^\theta)) \le e^{J_1}, \; e^{J_2}$.

### 3.2 ROULETTE DISCRETE DIFFUSION

Typically, matrices $\boldsymbol{Q}^{tok}$ are defined as $\boldsymbol{Q}^{tok} = \boldsymbol{P}^{tok} - \boldsymbol{I}$, where $\boldsymbol{P}^{tok}$ is idempotent, since this implies that $(\boldsymbol{Q}^{tok})^2 = -\boldsymbol{Q}^{tok}$. This last property of $\boldsymbol{Q}^{tok}$ greatly simplifies the calculation of $e^{\sigma(t)\boldsymbol{Q}^{tok}}$, as by using the Taylor series, $e^{\sigma(t)\boldsymbol{Q}^{tok}} = \boldsymbol{I} + \boldsymbol{Q}^{tok}(1 - e^{-\sigma(t)})$. Usually, the following two

matrices $\boldsymbol{P}^{tok}$ are chosen: a) $\boldsymbol{P}^{tok}_{uniform}(V \times V)$ where each entry is set to $\frac{1}{V}$, and b) $\boldsymbol{P}^{tok}_{absorb}(n \times n)$ in which the last row is full of ones while all other elements are 0, where $n = V + 1$. While absorb diffusion often outperforms the uniform one in standard evaluations, the latter is more practical for some tasks like spelling correction, where refining tokens is crucial. We propose another transition-rate matrix whose exponential can be analytically calculated. To our knowledge this is the only matrix with such a property presented so far apart from the absorb and uniform ones. We refer to this new discrete diffusion process as *roulette* diffusion. The corresponding $\boldsymbol{P}^{tok}_{roulette}(n \times n)$ is a matrix, such that $\boldsymbol{P}^{tok}_{roulette}(i \neq n, j \neq n) = \frac{1}{V}(1 - p_m)$, $\boldsymbol{P}^{tok}_{roulette}(n, j \neq n) = p_m$, $\boldsymbol{P}^{tok}_{roulette}(i \neq n, n) = 0$ and $\boldsymbol{P}^{tok}_{roulette}(n, n) = 1$. We notice that for $p_m = 1$, roulette diffusion coincides with absorb diffusion, while for $p_m = 0$ it coincides with the uniform diffusion. Intuitively, a token can transit from a non-absorb state to a non-absorb state with probability $\frac{1}{V}(1 - p_m)$, until it hits the absorb state (with probability $p_m$) and then remains there. While this matrix is not idempotent, one can still calculate its exponential as stated in the following proposition (proved in Appendix A.3.3):

**Proposition 5.** *If we denote with* $\boldsymbol{Y}_t$ *the matrix exponential of* $\sigma_t \boldsymbol{Q}^{tok}_{roulette} = \sigma_t \left( \boldsymbol{P}^{tok}_{roulette} - \boldsymbol{I} \right)$, *then* $\boldsymbol{Y}_t(i \notin \{j, n\}, j \neq n) = e^{-\sigma_t p_m} \frac{1}{n-1}(1 - e^{-(1-p_m)\sigma_t})$, $\boldsymbol{Y}_t(i \neq n, i \neq n) = e^{-\sigma_t p_m}(1 - \frac{n-2}{n-1}(1 - e^{-(1-p_m)\sigma_t}))$, $\boldsymbol{Y}_t(n, j \neq n) = 1 - e^{-\sigma_t p_m}$, $\boldsymbol{Y}_t(i \neq n, n) = 0$, *and* $\boldsymbol{Y}_t(n, n) = 1$.

The noise schedule used is the roulette-loglinear noise $-\frac{1}{p_m} \log(1 - (1 - \epsilon)t)$. In the reverse process, when a token is unmasked it can still be corrected with probability directly related to $p_m$ as shown in Appendix A.3.4. A generalization for time-evolving $p_m$ is given in Appendix A.3.5, and the corresponding $\boldsymbol{Q}^{tok}_{eroulette}(t)$ is named *eroulette*. Therein (Proposition 7), it is shown that the exponential matrix can be calculated as in the previous proposition, by substituting $p_m$ with $p_m(t)$.

## 3.3 EFFICIENT ESTIMATION OF $J_2$

In this subsection we provide Proposition 6, which shows that the second and third term on the RHS of Expression (14) can be computed efficiently:

**Proposition 6.** *In the case of the roulette diffusion with roulette-loglinear noise,* $H(p_r) = 0$ *and* $-\int_0^1 \mathbb{E}_{\boldsymbol{x}_t \sim p_t} \sum_{\boldsymbol{y} \neq \boldsymbol{x}_t} \boldsymbol{Q}_t(\boldsymbol{y}, \boldsymbol{x}_t) dt = \left(1 - \frac{1-p_m}{n-1}\right) \frac{L}{p_m}(\epsilon - 1)$. *For the absorb diffusion, we have* $H(p_r) = 0$ *and* $-\int_0^1 \mathbb{E}_{\boldsymbol{x}_t \sim p_t} \sum_{\boldsymbol{y} \neq \boldsymbol{x}_t} \boldsymbol{Q}_t(\boldsymbol{y}, \boldsymbol{x}_t) dt = L(\epsilon - 1)$. *Finally, in the case of uniform diffusion,* $H(p_r) = L \log(V)$ *and* $-\int_0^1 \mathbb{E}_{\boldsymbol{x}_t \sim p_t} \sum_{\boldsymbol{y} \neq \boldsymbol{x}_t} \boldsymbol{Q}_t(\boldsymbol{y}, \boldsymbol{x}_t) dt = -\left(1 - \frac{1}{V}\right) L \int_0^1 \sigma'(t) dt$.

## 3.4 MODELING RATIOS VIA CEDD

For sequences $\boldsymbol{x}, \boldsymbol{y}$ which only differ at some position $i$, we can write

$$\frac{p_t(\boldsymbol{y})}{p_t(\boldsymbol{x}_t)} = \Sigma_{h \in [V]} \frac{p_{t|0}(\boldsymbol{y}^i|h)}{p_{t|0}(\boldsymbol{x}_t{}^i|h)} p^i_{0|t}(h|\boldsymbol{x}_t), \tag{17}$$

where $p^i_{0|t}(\boldsymbol{x}^i_0|\boldsymbol{x}_t) = \sum_{\{\boldsymbol{x}^1_0, \ldots, \boldsymbol{x}^L_0\} \setminus \boldsymbol{x}^i_0} p_{0|t}(\boldsymbol{x}_0|\boldsymbol{x}_t)$, as shown in Equation (85), Appendix A.2.2. Since conditional ratios $\frac{p_{t|0}(\boldsymbol{y}^i|h)}{p_{t|0}(\boldsymbol{x}_t{}^i|h)}$ are known, we can choose to reparametrize the score as

$$s^i_\theta(\boldsymbol{x}_t, t)_{\boldsymbol{y}} = \Sigma_{h \in [V]} \frac{p_{t|0}(\boldsymbol{y}^i|h)}{p_{t|0}(\boldsymbol{x}_t{}^i|h)} f^i_\theta(\boldsymbol{x}_t, t)[h], \tag{18}$$

where $f_\theta(\boldsymbol{x}_t, t)$, is a neural network, with a matrix output of shape $L \times V$, whose elements in each row $i$ add to 1, that is $\sum_h f^i_\theta(\boldsymbol{x}_t, t)[h] = 1$. Intuitively, given the current perturbed sequence $\boldsymbol{x}_t$, prediciton $f^i_\theta(\boldsymbol{x}_t, t)[h]$ gives the probability that the pre-perturbation token at position $i$ used to be $h$. In Appendix A.2.2, we explain how the loss in Expression (7) can be optimized by minimizing the following cross-entropy loss:

$$L_{ll} = -\mathbb{E}_{t \sim U(0,1)} \mathbb{E}_{\boldsymbol{x}_0 \sim p_0(\boldsymbol{x}_0)} \mathbb{E}_{\boldsymbol{x}_t \sim p_{t|0}(\cdot|\boldsymbol{x}_0)} \sum_{i=1}^{L} w(t) \log f^i_\theta(\boldsymbol{x}_t, t)[\boldsymbol{x}^i_0]. \tag{19}$$

Thus, we can learn the ratios via $L_{ll}$ (Campbell et al., 2022) for any type of diffusion model. This approach is analogous to the one in continuous diffusion models for language modeling used in

Dieleman et al. (2022), as shown in Appendix A.2.3. In addition, in Appendix B.3.2, we provide our original motivation for the reparametrization given in (18).

In the uniform case, models using the direct reparametrization in (18) underperform SEDD in terms of perplexity. Indeed, the model is too confident in its predictions when $t \to 0$ as it does not benefit form the neural network regularization due to the incorporation of the true conditional ratios. Thus, inside the ratios $\frac{p_{t|0}(\boldsymbol{y}^i|h)}{p_{t|0}(\boldsymbol{x}_t{}^i|h)}$ we rescale $\sigma_t^\theta < 0.0015$, by setting $\sigma_t^\theta = 0.0015$. We also perform rescaling in the case of roulette diffusion dynamics, as when time is close to 0 most tokens are unmasked. The same strategy is applied, as before, only to ratios corresponding to unmasked token, by rescaling $\sigma_t^\theta$ when $\sigma_t^\theta < 0.5$ as follows: $\sigma_t^{scaled,\theta} = \log(1.1\sigma_t^\theta + 1.1)$. Naturally, for the sake of rigor, these are also the models we employ to generate samples, whose quality is measured in terms of generative perplexity. Further, in order to evade metric hacking, we take care to not modify the $\sigma_t$ in the metrics $J_1$ and $J_2$ used to evaluate the models. Additional details and motivations for such rescalings are provided in the last paragraph of Appendix B.3.3. When $w(t) = 1$ we refer to the method as CEDD, while when $w(t) = \log\left(e + \frac{0.3}{t}\right)$ we use CEDD*.

## 4 EXPERIMENTS

We now empirically validate the approaches and theoretical contributions presented in the previous section. In Subsection 4.1, we compare the generative perplexities of models trained on OpenWeb-Text (Gokaslan & Cohen, 2019) with SEDD and SEDD scaled (SEDDs, see Appendix B.3.2 for details) versus those trained with CEDD and CEDD*. We keep all other variables unchanged for a fair comparison, finding that CEDD outperforms SEDD in all cases. The tests are conducted for the absorb, uniform and roulette diffusion dynamics. In Subsection 4.2, we evaluate the perplexity of the models, by calculating the upper bound on 5 different datasets, namely: 1BW, LAMBADA, PTB, Wikitext2 and Wikitext103 (Chelba et al., 2013; Paperno et al., 2016; Marcus et al., 1993; Merity et al., 2016). For the sake of reproducing the results of previous work we use $J_1$ as a metric. Finally, having computed the results using $J_1$, in Subsection 4.3 we also re-evaluate the models via $J_2$. Our findings suggest that the bound provided by $J_2$ is slightly tighter. In the last subsection we compare SEDD and CEDD* on a spell-checking task.

Our model employs the transformer architecture as described by Lou et al. (2024), with no modifications; more details can be found in Appendix B.2. The algorithms for training via SEDD, and sampling unconditionally and conditionally can be found in the Appendix of Lou et al. (2024). On the other hand, we provide the algorithm for training using CEDD in Algorithm 1, Appendix B.1. In all cases, samples are generated using tau-leaping (Gillespie, 2001; Campbell et al., 2022), which performs an update at each position simultaneously for each reverse time step. All models were trained for 400k parameter updates unless stated otherwise.

### 4.1 GENERATIVE PERPLEXITY COMPARISONS

We compare the generative perplexities (GenPerp) of identical networks trained using SEDD, SEDDs, CEDD and CEDD*. To evaluate the generative perplexity of a model, we generate samples from that model, and use a GPT-2 large model in order to assess the likelihood of the generated samples. However, this metric can be unreliable, as models such as GPT-2 large are not perfect themselves, and they tend to assign high probability to some unlikely sequences, such as those that contain repetitive tokens. Such biased samples can be generated by increasing the step size, while maintaining the number of reverse steps. To ensure a fair comparison, we evade such approaches. Furthermore, recently Zheng et al. (2024) showed that the sampling procedure in Lou et al. (2024) suffers from numerical precision issues. To address this, they proposed fixing the categorical sampling to 64-bit floating-point precision—a strategy that we have also adopted. In Appendix B.6, we also provide perplexity results as evaluated by LLama 3.1 8B (Dubey et al., 2024).

Initially, we perform a grid search to find the best $p_m$ for the roulette case. Out of the 4 values, 0.95, 0.65, 0.35, 0.05, we found $p_m = 0.95$ performs best (Appendix B.6, Table 5), but the optimum is likely reached when $p_m = 1$. However, $p_m = 0.95$, enables equipping unmasking dynamics with the correction mechanism, which can be useful for some tasks like spelling correction. Therefore, in what follows, when the roulette case is concerned, it is implied that $p_m = 0.95$. In Table 1, we provide the results when the sequence length is 128 and the number of reverse steps is 1024, when using the analytic sampler. Generated samples can be found in Appendix C.1.

## 4.2 PERPLEXITY COMPARISONS

In this subsection, we compare the performance of identical networks trained using SEDD, SEDDs, CEDD and CEDD*, in terms of $\bar{J}_1$. We do not shuffle the test set. Results for sequence length $L = 128$ are provided in Table 1. Plots illustrating cummulative performance trajectories across the testing sets are provided in Appendix B.6.

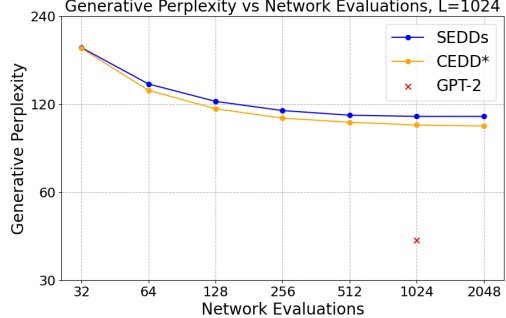

| | |
|---|---|
| **SEDDs** | **Seeing them is a treat,** because I'm coaching. It's a full week for me. "On Saturday night, Gonzalez spoke to media and said he thinks he could work out the fine fit to make the made |
| **CEDD*** | **Seeing them is a treat,** though it was only in her presence. Just being able to think about all the soldiers out here helped make the place even look like their countries and the rest of |
| **GPT-2** | **Seeing them is a treat,** and ultimately all the practical work should be well underway for everyone to enjoy. All Milan, Italy-based staff members will be given access to the dev portal |

Figure 1: Scaling of Generative Perplexity vs sampling steps for SEDDs (loaded) and CEDD* absorb.

Figure 2: Filtered samples from SEDDs and CEDD* absorb, L=1024. The conditional part is highlighted in bold.

Table 1: Results comparing SEDD, SEDD scaled (SEDDs), CEDD and CEDD*. Lower is better.

| Model (L=128) | GenPerp | LAMBADA | WikiText2 | PTB | WikiText103 | 1BW |
|---|---|---|---|---|---|---|
| SEDD Absorb | 172.35 | 70.07 | 75.20 | 240.43 | 74.79 | 88.99 |
| SEDDs Absorb | 166.35 | 67.05 | 69.37 | 208.69 | 69.17 | 83.87 |
| CEDD Absorb | 148.21 | 65.18 | 65.66 | 199.69 | 65.62 | 79.83 |
| CEDD* Absorb | **143.86** | **64.60** | **65.04** | **192.99** | **64.69** | **79.81** |
| SEDD Roulette | 178.94 | 72.07 | 80.13 | 230.74 | 79.68 | 93.45 |
| SEDDs Roulette | 172.93 | 69.10 | 74.38 | **209.12** | 74.16 | 88.02 |
| CEDD Roulette | 167.67 | 69.77 | 72.91 | 227.16 | 72.49 | **86.55** |
| CEDD* Roulette | **158.56** | **67.84** | **70.54** | 216.91 | **70.18** | 86.76 |
| SEDD Uniform | 169.66 | 80.74 | 91.79 | 252.81 | 91.40 | 102.75 |
| SEDDs Uniform | 163.88 | 81.13 | 89.21 | **228.37** | 88.56 | 100.80 |
| CEDD Uniform | **161.84** | **80.27** | **87.91** | 279.65 | **87.46** | **99.34** |
| CEDD* Uniform | 175.42 | 82.54 | 89.68 | 289.09 | 88.90 | 106.32 |
| DFM $k_t = t$ | **145.48** | **71.90** | **71.20** | 221.15 | **70.84** | **82.63** |
| DFM $k_t = t^2$ | 152.70 | 72.31 | 72.87 | **215.30** | 72.55 | 85.82 |

We also compare our approach against models trained utilizing Discrete Flow matching (DFM) (Gat et al., 2024). In Table 1, we present results when comparing against flows with convex interpolants, where we chose schedules $k_t = t$, as in Campbell et al. (2024), as well as $k_t = t^2$. The perplexity bound for these models is calculated using Expression (24) in Haxholli et al. (2024).

It can be seen that CEDD* absorb performs best overall, thus we compare this model, against CEDDT, that is, CEDD with the scaling loss used in the SOTA discrete diffusion model (Sahoo et al., 2024). Interestingly, our scaling CEDD* outperforms that of CEDDT, despite its theoretical support with regards to the score entropy loss. The results can be found in Table 2.

We also train 3 absorb models, namely SEDDs, CEDD*, CEDDT, as well as GPT2, with a sequence length of 1024. Results are provided in Table 2, where it can be seen that overall GPT-2 performs best. The gaps between SEDDs, CEDD*, CEDDT are reduced, likely since by seeing more tokens they all approach their optimal performance. However, models trained with CEDD/CEDD* converge faster to the optimum in terms of number of parameter updates. In Appendix B.6, Table 7 and Figure 8, we show the difference in performance between CEDD* and SEDDs absorb during

and after training for 20k parameter updates. In addition, training with CEDD (and its variants) is roughly 15% faster per iteration, due to the simplified loss function. Furthermore, by incorporating the $f_\theta^i(\boldsymbol{x}_t, t)$ to $s_\theta^i(\boldsymbol{x}_t, t)$ scaling in the timestep, absorb models trained with CEDD/CEDD* generate sequences 2% faster than those trained with SEDDs.

Table 2: Results comparing SEDDs (retrained), CEDD*, CEDDT and GPT-2 (retrained) in terms of generative perplexity, and perplexity on 5 test sets. Number of generation steps is 1024.

| Model (Absorb) | GenPerp | LAMBADA | WikiText2 | PTB | WikiText103 | 1BW |
|---|---|---|---|---|---|---|
| SEDDs L=128 | 166.35 | 67.05 | 69.37 | 208.69 | 69.17 | 83.87 |
| CEDD* L=128 | **143.86** | **64.60** | **65.04** | **192.99** | **64.69** | **79.81** |
| CEDDT L=128 | 154.04 | 68.24 | 68.61 | 204.76 | 68.10 | 81.81 |
| SEDDs L=1024 | 105.27 | **52.18** | 42.02 | 117.00 | 41.83 | 80.79 |
| CEDD* L=1024 | **101.83** | 52.70 | **41.57** | **115.99*** | **41.31** | **77.96** |
| CEDDT L=1024 | 108.88 | 53.20 | 42.24 | 121.05 | 42.07 | 78.10 |
| D3PM L=1024 | - | 93.47 | 77.28 | 200.82 | 75.16 | 138.92 |
| PLAID L=1024 | - | 57.28 | 51.80 | 142.60 | 50.86 | 91.12 |
| GPT-2 L=1024 | **41.02*** | **49.02*** | **37.68*** | 134.13 | **37.55*** | **58.92*** |

### 4.3 Comparing the Two Upper Bounds

Lastly, we compare the two upper bounds $J_1$ and $J_2$. The bound $J_2$ is shown empirically to be slightly tighter than $J_1$, supporting the importance of Theorem 4 and Proposition 6. We estimate the integral with respect to time in both $J_1$ and $J_2$ by randomly sampling time points from a uniform distribution in the interval $(e^{-4}, 1 - e^{-4})$, and the averaging the loss. The procedure of comparing these bounds is explained in detail in Appendix B.4. We present the results for several models in Table 3, while we show the testing plots for CEDD* absorb (L=1024) in Figure 3. The proposed bound $J_2$ gives a lower bound in every single case, as it can also be seen in Table 6, Appendix B.6, where we provide the equivalent of Table 1 when $J_2$ is utilized.

Table 3: Results comparing $J_1$ and $J_2$ for the best performing models of each category.

| Model/L/Perplexity-Bound | LAMBADA | WikiText2 | PTB | WikiText103 | 1BW |
|---|---|---|---|---|---|
| SEDDs absorb/1024/ $\exp(J_1)$ | 52.18 | 42.02 | 117.00 | 41.83 | 80.79 |
| SEDDs absorb/1024/ $\exp(J_2)$ | **51.78** | **41.76** | **115.97** | **41.51** | **80.53** |
| CEDD* absorb/1024/ $\exp(J_1)$ | 52.70 | 41.57 | 115.99 | 41.31 | 77.96 |
| CEDD* absorb/1024/ $\exp(J_2)$ | **52.10** | **41.18** | **115.03** | **40.98** | **77.28** |
| CEDD* absorb/128/ $\exp(J_1)$ | 64.60 | 65.04 | 192.99 | 64.69 | 79.81 |
| CEDD* absorb/128/ $\exp(J_2)$ | **64.11** | **64.54** | **191.38** | **64.30** | **79.17** |
| CEDD* roulette/128/ $\exp(J_1)$ | 67.84 | 70.54 | 216.91 | 70.18 | 86.76 |
| CEDD* roulette/128/ $\exp(J_2)$ | **67.27** | **69.61** | **213.90** | **69.45** | **85.64** |
| CEDD uniform/128/ $\exp(J_1)$ | 80.27 | 87.91 | 279.65 | 87.46 | 99.34 |
| CEDD uniform/128/ $\exp(J_2)$ | **79.46** | **86.82** | **276.61** | **86.52** | **98.44** |

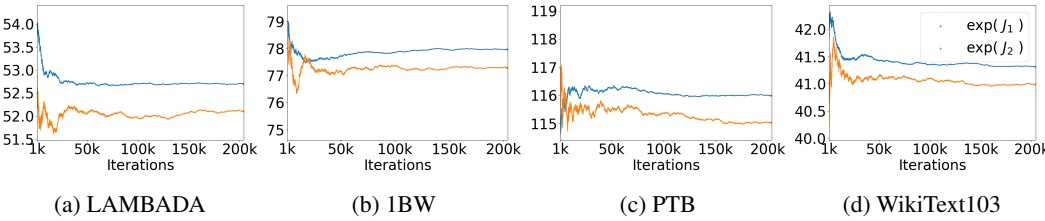

(a) LAMBADA      (b) 1BW      (c) PTB      (d) WikiText103

Figure 3: Upper bounds $J_1$ and $J_2$ of CEDD* absorb L=1024 for different testing sets.

## 4.4 SPELLING CORRECTION

We evaluate our uniform and roulette diffusion models on a character-level unsupervised spell-checking task (Hoogeboom et al., 2021). We train CEDD* and SEDD models on 'War and Peace' (3.3M tokens), and test on 'Crime and Punishment' (CAP) and 'Pride and Prejudice' (PAP). The test set is contaminated with mistakes (5% of characters), and the model chooses the most likely correction. Results when training for 25k/50k iterations are provided in Table 4 and Appendix B.6.

Table 4: Correction accuracy percentages. 25k training iterations, batch size of 32 and $L = 128$.

| Model (L=128) | CEDD* Uniform | SEDD Uniform | CEDD* Roulette | SEDD Roulette |
|---|---|---|---|---|
| PAP | 89.5 | 86.9 | **89.7** | 85.1 |
| CAP | 89.5 | 87.5 | **90.3** | 85.8 |

## 5 RELATED WORK AND FUTURE OUTLOOK

**The continuous diffusion** approach has demonstrated excellent performance in modeling continuous data distributions (Song et al., 2020a; 2021c; Kingma et al., 2021b; Nichol & Dhariwal, 2021; Saharia et al., 2022; Ramesh et al., 2022), leading to numerous efforts to adapt it for language modeling tasks (Chen et al., 2023; Gulrajani & Hashimoto, 2024; Li et al., 2022; Dieleman et al., 2022; Strudel et al., 2022; Gong et al., 2022; Mahabadi et al., 2023). Although initial results were not competitive, recent advancements have reduced the performance gap with autoregressive models. **Discrete diffusion models** offer an alternative approach for modeling categorical distributions like language data. Originally proposed by Hoogeboom et al. (2021); Austin et al. (2021), the framework was extended to continuous time by Campbell et al. (2022). Both strands of work employ a combination of the variational lower bound and cross entropy loss, the latter being central to our training approach and corresponding to the strategy employed by Dieleman et al. (2022) in the continuous case. The cross-entropy loss is also derived inSahoo et al. (2024); Shi et al. (2024); Ou et al. (2025), but only for the absorb transition-rate matrix. In addition, the cross entropy loss is similar to the loss employed in Sun et al. (2023), however in their case, one conditions on the current sequence $x_t^{-i}$ without the token at position $i$, and attempts to maximize the likelihood of $x_t^i$. In contrast, Lou et al. (2024) proposed modeling ratios of probabilities (Meng et al., 2022) directly using score entropy. **The evaluation** of such models can be performed by using Theorem 3, as originally derived and used in Lou et al. (2024). Inspired by this result, we formulate and prove the discrete version of the rest of the theorems in Song et al. (2021b), which provide important information regarding the KL divergence between the data and learned distributions in CTMCs, and which justify the usage of $J_2$. **Transition-rate matrices** are an essential component in CTMCs, as demonstrated by the performance discrepancy between the absorb and uniform matrices (Lou et al., 2024; Campbell et al., 2022). We introduced roulette matrix, an interpolation between the two and derived its exponential. **Regarding future work**, numerous avenues remain open for optimization within the diffusion model framework such as the exploration of the Eroulette matrix. Additionally, the exploration of noise schedules in discrete diffusion models remains relatively nascent, as similar to Lou et al. (2024), we did not systemically explore noise schedules. In our experiments, cross entropy loss is modulated by heuristically chosen time-dependent weighting. Investigating and establishing a general optimal weighting schedule could further refine performance metrics. Finally, studying scaling laws for discrete diffusion models and establishing their practical utility in downstream tasks is crucial.

## 6 CONCLUSION

In this work, we provided three new theorems concerning the KL divergence between the data and the learned distribution, improving model evaluation through the bound presented in Theorem 4. We also introduced a new transition-rate matrix that allows for token correction after unmasking in the reverse process, and derived its exponential matrix to enable efficient training/sampling. Finally, we proposed favoring denoising cross entropy loss over score entropy for training discrete diffusion models due to the findings in the experiments we conducted.

ETHICS STATEMENT

This paper introduces research that progresses the field of natural language generation. Beyond the pre-existing ethical concerns associated with this domain, such as e.g. bias, toxicity, and the generation of deceptive content, our methodology poses no unique risks. This is because our work is primarily theoretical and not of a magnitude that could cause any distinct issues.

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

**APPENDIX**

## A    THEORETICAL RESULTS

### A.1    KL DIVERGENCE THEOREMS

**Lemma 1.** *Let $q_1^*$ and $\hat{q}_1$ denote two initial distributions and define two Continuous-Time Markov Chains (CTMCs) running from time 1 to 0, with transition-rate matrices $\boldsymbol{Q}_t^*(y, x) = \boldsymbol{Q}_t(x, y)r(x, y)$ and $\hat{\boldsymbol{Q}}_t(y, x) = \boldsymbol{Q}_t(x, y)\hat{r}(x, y)$, where $r(x, y)$ and $\hat{r}(x, y)$ are chosen such that they ensure $\boldsymbol{Q}_t^*$ and $\hat{\boldsymbol{Q}}_t$ are indeed transition-rate matrices. When the first process (with matrix $\boldsymbol{Q}_t^*(y, x)$) is applied to initial distribution $q_1^*$ and the second (with matrix $\hat{\boldsymbol{Q}}_t(y, x)$) to initial distribution $\hat{q}_1$, they define distributions $q_t^*$ and $\hat{q}_t$ at time t, and $q_0^*$, $\hat{q}_0$ at the end of the diffusion, for which*

$$D_{KL}(q_0^*||\hat{q}_0)$$

$$\leq \int_0^1 \mathbb{E}_{x_t \sim q_t^*} \sum_{y \neq x_t} \boldsymbol{Q}_t(x_t, y)\ell\left(r(x_t, y), \hat{r}(x_t, y)\right) dt + D_{KL}(q_1^*||\hat{q}_1), \tag{20}$$

*where $\ell(a, b) = (b - a \log b + K(a))$ and $K(a) = a(\log a - 1)$.*

*Proof.* From the data processing inequality applied to the reverse of the processes defined in the Lemma, we have $D_{KL}(q_0^*||\hat{q}_0) \leq D_{KL}(q^*||\hat{q})$ where $q^*$ is the measure over the space of paths generated by the first process, while $\hat{q}$ is the measure over the space of paths generated by second one. Using the expression for the KL divergence between path measures of two CTMCs found in (Opper & Sanguinetti, 2007, Section 2.1), and by substituting $q_t(x) := q^*(x_t)$, $g(x'|x) = \boldsymbol{Q}_t^*(y, x_t)$ and $f(x'|x) = \hat{\boldsymbol{Q}}_t(y, x_t)$, we have:

$$D_{KL}(q^*||\hat{q}) = \int_0^T \mathbb{E}_{x_t \sim q_t^*} \sum_{y \neq x_t} \left( \boldsymbol{Q}_t^*(y, x_t) \log \frac{\boldsymbol{Q}_t^*(y, x_t)}{\hat{\boldsymbol{Q}}_t(y, x_t)} + \hat{\boldsymbol{Q}}_t(y, x_t) - \boldsymbol{Q}_t^*(y, x_t) \right) dt$$

$$+ D_{KL}(q_1^*||\hat{q}_1). \tag{21}$$

Now we focus on the expression inside the sum:

$$\boldsymbol{Q}_t^*(y, x_t) \log \frac{\boldsymbol{Q}_t^*(y, x_t)}{\hat{\boldsymbol{Q}}_t(y, x_t)} + \hat{\boldsymbol{Q}}_t(y, x_t) - \boldsymbol{Q}_t^*(y, x_t) = \tag{22}$$

$$\boldsymbol{Q}_t(x_t, y)r(x_t, y) \log \frac{\boldsymbol{Q}_t(x_t, y)r(x_t, y)}{\boldsymbol{Q}_t(x_t, y)\hat{r}(x_t, y)} + \boldsymbol{Q}_t(x_t, y)\hat{r}(x_t, y) - \boldsymbol{Q}_t(x_t, y)r(x_t, y) = \tag{23}$$

$$\boldsymbol{Q}_t(x_t, y)r(x_t, y)\left(\log r(x_t, y) - \log \hat{r}(x_t, y)\right) + \boldsymbol{Q}_t(x_t, y)\hat{r}(x_t, y) - \boldsymbol{Q}_t(x_t, y)r(x_t, y) = \tag{24}$$

$$\boldsymbol{Q}_t(x_t, y)r(x_t, y)\left(\log r(x_t, y) - 1\right) - \boldsymbol{Q}_t(x_t, y)r(x_t, y) \log \hat{r}(x_t, y) + \boldsymbol{Q}_t(x_t, y)\hat{r}(x_t, y) = \tag{25}$$

$$\boldsymbol{Q}_t(x_t, y)\left[K\left(r(x_t, y)\right) + \left(\hat{r}(x_t, y) - r(x_t, y) \log \hat{r}(x_t, y)\right)\right] = \boldsymbol{Q}_t(x_t, y)\ell\left(r(x_t, y), \hat{r}(x_t, y)\right), \tag{26}$$

which concludes the proof. We highlight that the CTMCs defined in the Lemma are completely arbitrary and not necessarily related to one-another, as the choices of $r(x, y)$ and $\hat{r}(x, y)$ can completely overwrite the matrix $\boldsymbol{Q}_t(x, y)$. $\square$

**Theorem 1.** *Define a CTMC with transition matrix $\boldsymbol{Q}_t$ that runs from time 0 to 1. The true reverse process defines a probability evolution $p_t$ from $p_1$ to the data distribution $p_0$, while the learned reverse process induces the evolution $p_t^\theta$ from the reference distribution $p_1^\theta = p_r$ to the approximation of the data distribution $p_0^\theta$. In this setting, the following KL divergence bound holds*

$$D_{KL}(p_0||p_0^\theta) \leq \int_0^1 \mathbb{E}_{x_t \sim p_t} \sum_{y \neq x_t} \boldsymbol{Q}_t(x_t, y)\ell\left(\frac{p_t(y)}{p_t(x_t)}, s_\theta(x_t, t)_y\right) dt + D_{KL}(p_1||p_r). \tag{27}$$

*Proof.* In Lemma 1, we set $r(x, y) = \frac{p_t(y)}{p_t(x)}$ and $\hat{r}(x_t, y) = s_\theta(x_t, t)_y$. This implies that $q_t^* = p_t$ and $\hat{q}_t = p_t^\theta$. In particular, $q_0^* = p_0$, $\hat{q}_0 = p_0^\theta$ and $q_1^* = p_1$, $\hat{q}_1 = p_1^\theta := p_r$. Plugging everything in Expression (20) gives the desired result. $\square$

**Theorem 3.** *Let $p_0^\theta$ denote the learned distribution from which the reverse process samples. The negative log-probability of a state $x_0$ being sampled by the reverse process can be bounded from above as follows,*

$$-\log p_0^\theta(x_0) \leq \int_0^1 \mathbb{E}_{x_t \sim p_{t|0}(\cdot|x_0)} \sum_{y \neq x_t} \boldsymbol{Q}_t(x_t, y) \ell\left(\frac{p_{t|0}(y|x_0)}{p_{t|0}(x_t|x_0)}, s_\theta(x_t, t)_y\right) dt$$

$$+ D_{KL}(p_{1|0}(\cdot|x_0)||p_r). \tag{28}$$

*Proof.* In Lemma 1 we set $r(x, y) = \frac{p_{t|0}(y|x_0)}{p_{t|0}(x_t|x_0)}$ and $\hat{r}(x, y) = s_\theta(x_t, t)_y$, where $p_{t|0}$ is the probability over states at time $t$ determined by a CTMC with transition-rate matrix $\boldsymbol{Q}_t$ applied to initial distribution $\delta(x = x_0)$. As such, we have $q_t^* = p_{t|0}$ and $\hat{q}_t = p_t^\theta$. In particular, $q_0^* = \delta(x = x_0)$, $\hat{q}_0 = p_0^\theta$ and $q_1^* = p_{1|0}$, $\hat{q}_1 = p_1^\theta := p_r$. Plugging everything in Expression (20):

$$D_{KL}(\delta(x = x_0)||p_0^\theta) \leq \int_0^1 \mathbb{E}_{x_t \sim p_{t|0}(\cdot|x_0)} \sum_{y \neq x_t} \boldsymbol{Q}_t(x_t, y) \left[ K\left(\frac{p_{t|0}(y|x_0)}{p_{t|0}(x_t|x_0)}\right) \right.$$

$$\left. + \left(s_\theta(x_t, t)_y - \frac{p_{t|0}(y|x_0)}{p_{t|0}(x_t|x_0)} \log s_\theta(x_t, t)_y\right) \right] dt + D_{KL}(p_{1|0}(x_0)||p_r).$$

This concludes the proof as $D_{KL}(\delta(x = x_0)||p_0^\theta) = -\log(p_0^\theta(x_0))$. $\qquad\square$

**Theorem 2.** *Denote the intermediate distributions at time $t$ determined by the true reverse process, and by the learned reverse process with $p_t$ and $p_t^\theta$, respectively. We can write the entropy of the data distribution $H(p_0)$ as*

$$H(p_0) = H(p_1) - \int_0^1 \mathbb{E}_{x_t \sim p_t} \sum_y \boldsymbol{Q}_t(x_t, y) K\left(\frac{p_t(y)}{p_t(x_t)}\right) dt. \tag{29}$$

*In addition, if the learned ratios $s_\theta(x_t, t)_y$ equal $\frac{p_t^\theta(y)}{p_t^\theta(x_t)}$ and $p_1 = p_1^\theta := p_r$, then the inequality in Theorem 1, becomes an equality.*

*Proof.* The cross entropy between the true data distribution, and the modeled data distribution satisfies the following:

$$H(p_0, p_0^\theta) - H(p_1, p_1^\theta) = \int_1^0 \frac{\partial}{\partial t} H(p_t, p_t^\theta) dt \tag{30}$$

To keep things clear we focus on $\frac{\partial}{\partial t} H(p_t, p_t^\theta)$.

$$\frac{\partial}{\partial t} H(p_t, p_t^\theta) = \frac{\partial}{\partial t} \int -p_t(x_t) \log(p_t^\theta(x_t)) dx_t = \int -\frac{\partial}{\partial t}[p_t(x_t) \log(p_t^\theta(x_t))] dx_t \tag{31}$$

$$= -\left(\int \frac{\partial p_t(x_t)}{\partial t} \log(p_t^\theta(x_t)) dx_t + \int \frac{p_t(x_t)}{p_t^\theta(x_t)} \frac{\partial p_t^\theta(x_t)}{\partial t} dx_t\right). \tag{32}$$

We can use the Kolmogorov forward equations (Equation 3), that is, $\sum_y \boldsymbol{Q}_t(x_t, y) p_t(y) = \frac{\partial p_t(x_t)}{\partial t}$ and $\frac{\partial p_t^\theta(x_t)}{\partial t} = \sum_y \boldsymbol{Q}_t(x_t, y) p_t^\theta(y)$ to get

$$= -\left(\int \sum_y \boldsymbol{Q}_t(x_t, y) p_t(y) \log(p_t^\theta(x_t)) dx_t + \int \frac{p_t(x_t)}{p_t^\theta(x_t)} \sum_y \boldsymbol{Q}_t(x_t, y) p_t^\theta(y) dx_t\right) \tag{33}$$

$$= -\left(\int \sum_y \boldsymbol{Q}_t(x_t, y) \frac{p_t(y)}{p_t(x_t)} \log(p_t^\theta(x_t)) p(x_t) dx_t + \int p_t(x_t) \sum_y \boldsymbol{Q}_t(x_t, y) \frac{p_t^\theta(y)}{p_t^\theta(x_t)} dx_t\right) \tag{34}$$

$$= -\mathbb{E}_{x_t \sim p_t} \sum_y \boldsymbol{Q}_t(x_t, y) \left(\frac{p_t(y)}{p_t(x_t)} \log(p_t^\theta(x_t)) + \frac{p_t^\theta(y)}{p_t^\theta(x_t)}\right) \tag{35}$$

$$= -\mathbb{E}_{x_t \sim p_t} \sum_y \boldsymbol{Q}_t(x_t, y) \left( \frac{p_t^\theta(y)}{p_t^\theta(x_t)} - \frac{p_t(y)}{p_t(x_t)} \log \frac{p_t^\theta(y)}{p_t^\theta(x_t)} \right) \tag{36}$$

$$+ \mathbb{E}_{x_t \sim p_t} \sum_y \boldsymbol{Q}_t(x_t, y) [\frac{p_t(y)}{p_t(x_t)} \log p_t^\theta(y)]. \tag{37}$$

The second term above $\mathbb{E}_{x_t \sim p_t} \sum_y \boldsymbol{Q}_t(x_t, y)[\frac{p_t(y)}{p_t(x_t)} \log p_t^\theta(y)]$ is zero. Indeed,

$$\mathbb{E}_{x_t \sim p_t} \sum_y \boldsymbol{Q}_t(x_t, y)[\frac{p_t(y)}{p_t(x_t)} \log(p_t^\theta(y))] = \sum_{x_t} \sum_y \boldsymbol{Q}_t(x_t, y) p_t(x_t)[\frac{p_t(y)}{p_t(x_t)} \log(p_t^\theta(y))] \tag{38}$$

$$= \sum_{x_t} \sum_y \boldsymbol{Q}_t(x_t, y) p_t(y) \log(p_t^\theta(y)) = \sum_y \left( \sum_{x_t} \boldsymbol{Q}_t(x_t, y) \right) p_t(y) \log(p_t^\theta(y)) = 0, \tag{39}$$

since for any column of a transition-rate matrix the sum of that columns elements is 0, therefore $\sum_x \boldsymbol{Q}_t(x_t, y) = 0$. In total we have:

$$H(p_0, p_0^\theta) - H(p_1, p_1^\theta) = \int_1^0 -\mathbb{E}_{x_t \sim p_t} \sum_y \boldsymbol{Q}_t(x_t, y) \left( \frac{p_t^\theta(y)}{p_t^\theta(x_t)} - \frac{p_t(y)}{p_t(x_t)} \log \frac{p_t^\theta(y)}{p_t^\theta(x_t)} \right) dt \tag{40}$$

$$= \int_0^1 \mathbb{E}_{x_t \sim p_t} \sum_y \boldsymbol{Q}_t(x_t, y) \left( \frac{p_t^\theta(y)}{p_t^\theta(x_t)} - \frac{p_t(y)}{p_t(x_t)} \log \frac{p_t^\theta(y)}{p_t^\theta(x_t)} \right) dt. \tag{41}$$

This allows the derivation of the expression for entropy of the data distribution, by taking $p_t^\theta = p_t$

$$H(p_0) = H(p_1) + \int_0^1 \mathbb{E}_{x_t \sim p_t} \sum_y \boldsymbol{Q}_t(x_t, y) \left( \frac{p_t(y)}{p_t(x_t)} - \frac{p_t(y)}{p_t(x_t)} \log \frac{p_t(y)}{p_t(x_t)} \right) dt, \tag{42}$$

thus

$$H(p_0) = H(p_1) - \int_0^1 \mathbb{E}_{x_t \sim p_t} \sum_y \boldsymbol{Q}_t(x_t, y) K \left( \frac{p_t(y)}{p_t(x_t)} \right) dt. \tag{43}$$

If we assume $\frac{p_t^\theta(y)}{p_t^\theta(x_t)} = s_\theta(x_t, t)_y$ and $p_1 = p_1^\theta := p_b$, from Equation (41) we get

$$H(p_0, p_0^\theta) - H(p_1) = \int_0^1 \mathbb{E}_{x_t \sim p_t} \sum_y \boldsymbol{Q}_t(x_t, y) \left( s_\theta(x_t, t)_y - \frac{p_t(y)}{p_t(x_t)} \log(s_\theta(x_t, t)_y) \right) dt, \tag{44}$$

and furthermore, from Equation (43) it is trivial that

$$-H(p_0) + H(p_1) = \int_0^1 \mathbb{E}_{x_t \sim p_t} \sum_y \boldsymbol{Q}_t(x_t, y) K \left( \frac{p_t(y)}{p_t(x_t)} \right) dt, \tag{45}$$

thus adding this last equation and Equation (44) we get

$$D_{KL}(p_0 || p_0^\theta) \tag{46}$$

$$= \int_0^1 \mathbb{E}_{x_t \sim p_t} \sum_y \boldsymbol{Q}_t(x_t, y) \left[ K \left( \frac{p_t(y)}{p_t(x_t)} \right) + \left( s_\theta(x_t, t)_y - \frac{p_t(y)}{p_t(x_t)} \log s_\theta(x_t, t)_y \right) \right] dt. \tag{47}$$

Since we know that $\frac{p_t(x_t)}{p_t(x_t)} = 1$, and $s_\theta(x_t, t)_{x_t} = \frac{p_t^\theta(x_t)}{p_t^\theta(x_t)} = 1$, we have

$$\left[ K \left( \frac{p_t(x_t)}{p_t(x_t)} \right) + \left( s_\theta(x_t, t)_{x_t} - \frac{p_t(x_t)}{p_t(x_t)} \log s_\theta(x_t, t)_{x_t} \right) \right] = -1 + 1 = 0 \tag{48}$$

so we can change the sums $\sum_y$ to simply $\sum_{y \neq x_t}$, as follows

$$D_{KL}(p_0 || p_0^\theta) \tag{49}$$

$$= \int_0^1 \mathbb{E}_{x_t \sim p_t} \sum_{y \neq x_t} \boldsymbol{Q}_t(x_t, y) \left[ K \left( \frac{p_t(y)}{p_t(x_t)} \right) + \left( s_\theta(x_t, t)_y - \frac{p_t(y)}{p_t(x_t)} \log s_\theta(x_t, t)_y \right) \right] dt. \tag{50}$$

Finally, since $p_1 = p_1^\theta := p_b$, then $D_{KL}(p_1 || p_r) = 0$, therefore we can add it to the right side, finishing the proof. $\qquad \square$

We remark that as $s_\theta(x_t, t)_y \to \frac{p_t(y)}{p_t(x_t)}$, then $p_t(i)^\theta \to p_t(i)$, and thus $\frac{p_t^\theta(y)}{p_t^\theta(x_t)} \to \frac{p_t(y)}{p_t(x_t)}$, as such $s_\theta(x_t, t)_y \to \frac{p_t^\theta(y)}{p_t^\theta(x_t)}$. Therefore, the bound in Theorem 1 becomes tighter as the model improves.

**Theorem 4.** *Under the conditions stated in Theorem 1, the following inequality for the cross entropy between the data and the learned distribution holds:*

$$H(p_0, p_0^\theta) \leq \int_0^1 \mathbb{E}_{x_t \sim p_t} \sum_{y \neq x_t} \boldsymbol{Q}_t(x_t, y) \bar{\ell}\left(\frac{p_t(y)}{p_t(x_t)}, s_\theta(x_t, t)_y\right) dt$$

$$- \int_0^1 \mathbb{E}_{x_t \sim p_t} \sum_{y \neq x_t} \boldsymbol{Q}_t(y, x_t) dt + H(p_1, p_r), \tag{51}$$

*where $\bar{\ell}(a, b) = (b - a \log b)$.*

*Proof.* From Theorem 1, we have

$$D_{KL}(p_0 || p_0^\theta) - D_{KL}(p_1 || p_r)$$

$$\leq \int_0^1 \mathbb{E}_{x_t \sim p_t} \sum_{y \neq x_t} \boldsymbol{Q}_t(x_t, y) \left[K\left(\frac{p_t(y)}{p_t(x_t)}\right) + \left(s_\theta(x_t, t)_y - \frac{p_t(y)}{p_t(x_t)} \log s_\theta(x_t, t)_y\right)\right] dt. \tag{52}$$

Since we know that $\frac{p_t(x_t)}{p_t(x_t)} = 1$, we can manually set $s_\theta(x_t, t)_{x_t} = 1$, and we get

$$\left[K\left(\frac{p_t(x_t)}{p_t(x_t)}\right) + \left(s_\theta(x_t, t)_{x_t} - \frac{p_t(x_t)}{p_t(x_t)} \log s_\theta(x_t, t)_{x_t}\right)\right] = -1 + 1 = 0, \tag{53}$$

therefore

$$H(p_0, p_0^\theta) - H(p_0) - D_{KL}(p_1 || p_r)$$

$$\leq \int_0^1 \mathbb{E}_{x_t \sim p_t} \sum_y \boldsymbol{Q}_t(x_t, y) \left[K\left(\frac{p_t(y)}{p_t(x_t)}\right) + \left(s_\theta(x_t, t)_y - \frac{p_t(y)}{p_t(x_t)} \log s_\theta(x_t, t)_y\right)\right] dt. \tag{54}$$

from which we deduce,

$$H(p_0, p_0^\theta)$$

$$\leq H(p_0) + \int_0^1 \mathbb{E}_{x_t \sim p_t} \sum_y \boldsymbol{Q}_t(x_t, y) K\left(\frac{p_t(y)}{p_t(x_t)}\right) dt$$

$$+ \int_0^1 \mathbb{E}_{x_t \sim p_t} \sum_y \boldsymbol{Q}_t(x_t, y) \left(s_\theta(x_t, t)_y - \frac{p_t(y)}{p_t(x_t)} \log s_\theta(x_t, t)_y\right) dt + D_{KL}(p_1 || p_r). \tag{55}$$

Since from Theorem 2 we have that

$$H(p_0) + \int_0^1 \mathbb{E}_{x_t \sim p_t} \sum_y Q_t(x, y) K\left(\frac{p_t(y)}{p_t(x_t)}\right) dt = H(p_1), \tag{56}$$

hence we can write

$$H(p_0, p_0^\theta)$$

$$\leq \int_0^1 \mathbb{E}_{x_t \sim p_t} \sum_y \boldsymbol{Q}_t(x_t, y) \left(s_\theta(x_t, t)_y - \frac{p_t(y)}{p_t(x_t)} \log s_\theta(x_t, t)_y\right) dt + H(p_1) + D_{KL}(p_1 || p_r),$$

and therefore

$$H(p_0, p_0^\theta) \leq \int_0^1 \mathbb{E}_{x_t \sim p_t} \sum_{y \neq x_t} \boldsymbol{Q}_t(x_t, y) \left(s_\theta(x_t, t)_y - \frac{p_t(y)}{p_t(x_t)} \log s_\theta(x_t, t)_y\right) dt$$

$$+ \int_0^1 \mathbb{E}_{x_t \sim p_t} \boldsymbol{Q}_t(x_t, x_t) dt + H(p_1, p_r). \tag{57}$$

The fact that $\boldsymbol{Q}_t(x_t, x_t) = -\sum_{y \neq x_t} \boldsymbol{Q}_t(y, x_t)$ concludes the proof. $\qquad\square$

**Proposition 6.** *In the case of the roulette diffusion with roulette-loglinear noise, $H(p_r) = 0$ and $-\int_0^1 \mathbb{E}_{\boldsymbol{x}_t \sim p_t} \sum_{\boldsymbol{y} \neq \boldsymbol{x}_t} \boldsymbol{Q}_t(\boldsymbol{y}, \boldsymbol{x}_t) dt = \left(1 - \frac{1-p_m}{n-1}\right) \frac{L}{p_m} (\epsilon - 1)$. For the absorb diffusion, that is when $p_m \to 1$, we have $-\int_0^1 \mathbb{E}_{\boldsymbol{x}_t \sim p_t} \sum_{\boldsymbol{y} \neq \boldsymbol{x}_t} \boldsymbol{Q}_t(\boldsymbol{y}, \boldsymbol{x}_t) dt = L(\epsilon - 1)$. Finally, for the uniform diffusion, $H(p_r) = L \log(V)$ and $-\int_0^1 \mathbb{E}_{\boldsymbol{x}_t \sim p_t} \sum_{\boldsymbol{y} \neq \boldsymbol{x}_t} \boldsymbol{Q}_t(\boldsymbol{y}, \boldsymbol{x}_t) dt = -\left(1 - \frac{1}{V}\right) L \int_0^1 \sigma'(t) dt$.*

*Proof.* In all cases,

$$-\int_0^1 \mathbb{E}_{\boldsymbol{x}_t \sim p_t} \sum_{\boldsymbol{y} \neq \boldsymbol{x}_t} \boldsymbol{Q}_t(\boldsymbol{y}, \boldsymbol{x}_t) dt = -\mathbb{E}_{t \sim U(0,1)} \mathbb{E}_{\boldsymbol{x}_t \sim p_t(\boldsymbol{x}_t)} \sum_{i=1}^L \sum_{\boldsymbol{y}^i \neq \boldsymbol{x}_t^i} \boldsymbol{Q}_t^{tok}(\boldsymbol{y}^i, \boldsymbol{x}_t^i) \quad (58)$$

$$= -\mathbb{E}_{t \sim U(0,1)} \sum_{i=1}^L \mathbb{E}_{\boldsymbol{x}_t \sim p_t(\boldsymbol{x}_t)} \sum_{\boldsymbol{y}^i \neq \boldsymbol{x}_t^i} \boldsymbol{Q}_t^{tok}(\boldsymbol{y}^i, \boldsymbol{x}_t^i) \quad (59)$$

$$= -\mathbb{E}_{t \sim U(0,1)} \sum_{i=1}^L \sum_{\boldsymbol{x}_t^i} p_t(\boldsymbol{x}_t^i) \sum_{\boldsymbol{y}^i \neq \boldsymbol{x}_t^i} \boldsymbol{Q}_t^{tok}(\boldsymbol{y}^i, \boldsymbol{x}_t^i) \quad (60)$$

$$= -\mathbb{E}_{t \sim U(0,1)} \sum_{i=1}^L \sum_{\boldsymbol{x}_0^i} p_0(\boldsymbol{x}_0^i) \sum_{\boldsymbol{x}_t^i} p_{t|0}(\boldsymbol{x}_t^i | \boldsymbol{x}_0^i) \sum_{\boldsymbol{y}^i \neq \boldsymbol{x}_t^i} \boldsymbol{Q}_t^{tok}(\boldsymbol{y}^i, \boldsymbol{x}_t^i) \quad (61)$$

For the roulette case, if $\boldsymbol{x}_t^i$ is the masked token, that is, $\boldsymbol{x}_t^i = n$, then $\sum_{\boldsymbol{y}^i \neq \boldsymbol{x}_t^i} \boldsymbol{Q}_t^{tok}(\boldsymbol{y}^i, \boldsymbol{x}_t^i) = 0$. Otherwise, $\sum_{\boldsymbol{y}^i \neq \boldsymbol{x}_t^i} \boldsymbol{Q}_t^{tok}(\boldsymbol{y}^i, \boldsymbol{x}_t^i) = \sigma_t' \left(1 - \frac{1-p_m}{n-1}\right)$. Thus,

$$-\int_0^1 \mathbb{E}_{\boldsymbol{x}_t \sim p_t} \sum_{\boldsymbol{y} \neq \boldsymbol{x}_t} \boldsymbol{Q}_t(\boldsymbol{y}, \boldsymbol{x}_t) = \quad (62)$$

$$= -\left(1 - \frac{1 - p_m}{n - 1}\right) \mathbb{E}_{t \sim U(0,1)} \sum_{i=1}^L \sum_{\boldsymbol{x}_0^i} p_0(\boldsymbol{x}_0^i) \sum_{\boldsymbol{x}_t^i \neq n} p_{t|0}(\boldsymbol{x}_t^i | \boldsymbol{x}_0^i) \sigma_t' = \quad (63)$$

$$= -\left(1 - \frac{1 - p_m}{n - 1}\right) \mathbb{E}_{t \sim U(0,1)} \sum_{i=1}^L \sum_{\boldsymbol{x}_0^i} p_0(\boldsymbol{x}_0^i) \sigma_t' \sum_{\boldsymbol{x}_t^i \neq n} p_{t|0}(\boldsymbol{x}_t^i | \boldsymbol{x}_0^i) \quad (64)$$

$$= -\left(1 - \frac{1 - p_m}{n - 1}\right) \mathbb{E}_{t \sim U(0,1)} \sum_{i=1}^L \sum_{\boldsymbol{x}_0^i} p_0(\boldsymbol{x}_0^i) \sigma_t' \left(1 - p(\boldsymbol{x}_t^i = n | \boldsymbol{x}_0^i)\right) \quad (65)$$

$$= -\left(1 - \frac{1 - p_m}{n - 1}\right) \mathbb{E}_{t \sim U(0,1)} \sum_{i=1}^L \sum_{\boldsymbol{x}_0^i} p_0(\boldsymbol{x}_0^i) \sigma_t' e^{-\sigma_t p_m} = -\left(1 - \frac{1 - p_m}{n - 1}\right) L \int_0^1 \sigma_t' e^{-\sigma_t p_m} dt \quad (66)$$

$$= \left(1 - \frac{1 - p_m}{n - 1}\right) \frac{L}{p_m} (e^{-\sigma_1 p_m} - e^{-\sigma_0 p_m}) \quad (67)$$

Since $\sigma_t$ is roulette-loglinear, that is $\sigma_t = -\frac{1}{p_m} \log\left(1 - (1 - \epsilon)t\right)$, we get the result stated in the Proposition.

In the Uniform case, we notice that Equation (61) can be rewritten as

$$\mathbb{E}_{t \sim U(0,1)} \sum_{i=1}^L \sum_{\boldsymbol{x}_0^i} \sum_{\boldsymbol{x}_t^i} p(\boldsymbol{x}_t^i, \boldsymbol{x}_0^i) \boldsymbol{Q}_t^{tok}(\boldsymbol{x}_t^i, \boldsymbol{x}_t^i) \quad (68)$$

$$= \mathbb{E}_{t \sim U(0,1)} \sum_{i=1}^L \sum_{\boldsymbol{x}_0^i} \sum_{\boldsymbol{x}_t^i} p(\boldsymbol{x}_t^i, \boldsymbol{x}_0^i) \sigma'(t) \left(\frac{1}{V} - 1\right) = -L \left(1 - \frac{1}{V}\right) \int_0^1 \sigma'(t) dt. \quad (69)$$

$\square$

## A.2 CROSS ENTROPY DISCRETE DIFFUSION (CEDD)

### A.2.1 REDERIVING THE DENOSING SCORE ENTROPY LOSS

In what follows, we rederive the Denosing Score Entropy Loss for completeness (Lou et al., 2024). Training a model using ratio matching is performed by minimizing the error of a network that receives as input $\boldsymbol{x}_t$ and tries to predict the ratios of $\frac{p_t(\boldsymbol{y})}{p_t(\boldsymbol{x}_t)}$, where $\boldsymbol{y}$ is a neighbour of $\boldsymbol{x}_t$ with Hamming distance of 1. In other words, one tries to minimize

$$\bar{\ell}\left(\frac{p_t(\boldsymbol{y})}{p_t(\boldsymbol{x}_t)}, s_\theta(\boldsymbol{x}_t, t)_{\boldsymbol{y}}\right), \tag{70}$$

for all $\boldsymbol{x}_t$ following distribution $p_t(\boldsymbol{x}_t)$, that is:

$$\mathbb{E}_{\boldsymbol{x}_t \sim p_t(\boldsymbol{x}_t)} \bar{\ell}\left(\frac{p_t(\boldsymbol{y})}{p_t(\boldsymbol{x}_t)}, s_\theta(\boldsymbol{x}_t, t)_{\boldsymbol{y}}\right). \tag{71}$$

The main bottleneck is that the ratios $\frac{p_t(\boldsymbol{y})}{p_t(\boldsymbol{x}_t)}$ are unknown, as $p_t(\boldsymbol{x}_t) = \int p_{t|0}(\boldsymbol{x}_t|\boldsymbol{x}_0)p_0(\boldsymbol{x}_0)d\boldsymbol{x}_0$ and $p_0(\boldsymbol{x}_0)$ is what we are trying to model in the first place. Luckily, the denoising trick can be employed:

$$\mathbb{E}_{\boldsymbol{x}_t \sim p_t(\boldsymbol{x}_t)} \bar{\ell}\left(\frac{p_t(\boldsymbol{y})}{p_t(\boldsymbol{x}_t)}, s_\theta(\boldsymbol{x}_t, t)_{\boldsymbol{y}}\right) = \mathbb{E}_{\boldsymbol{x}_t \sim p_t(\boldsymbol{x}_t)} \bar{\ell}\left(\Sigma_{\boldsymbol{x}_0} p_{t|0}(\boldsymbol{y}|\boldsymbol{x}_0) \frac{p_0(\boldsymbol{x}_0)}{p_t(\boldsymbol{x}_t)}, s_\theta(\boldsymbol{x}_t, t)_{\boldsymbol{y}}\right) = \tag{72}$$

$$= \mathbb{E}_{\boldsymbol{x}_t \sim p_t(\boldsymbol{x}_t)} \bar{\ell}\left(\Sigma_{\boldsymbol{x}_0} \frac{p_{t|0}(\boldsymbol{y}|\boldsymbol{x}_0)}{p_{t|0}(\boldsymbol{x}_t|\boldsymbol{x}_0)} \frac{p_{t|0}(\boldsymbol{x}_t|\boldsymbol{x}_0)p_0(\boldsymbol{x}_0)}{p_t(\boldsymbol{x}_t)}, s_\theta(\boldsymbol{x}_t, t)_{\boldsymbol{y}}\right) \tag{73}$$

$$= \mathbb{E}_{\boldsymbol{x}_t \sim p_t(\boldsymbol{x}_t)} \bar{\ell}\left(\Sigma_{\boldsymbol{x}_0} \frac{p_{t|0}(\boldsymbol{y}|\boldsymbol{x}_0)}{p_{t|0}(\boldsymbol{x}_t|\boldsymbol{x}_0)} p_{0|t}(\boldsymbol{x}_0|\boldsymbol{x}_t), s_\theta(\boldsymbol{x}_t, t)_{\boldsymbol{y}}\right) \tag{74}$$

$$= \mathbb{E}_{\boldsymbol{x}_t \sim p_t(\boldsymbol{x}_t)} \bar{\ell}\left(\mathbb{E}_{\boldsymbol{x}_0 \sim p_{0|t}(\boldsymbol{x}_0|\boldsymbol{x}_t)} \frac{p_{t|0}(\boldsymbol{y}|\boldsymbol{x}_0)}{p_{t|0}(\boldsymbol{x}_t|\boldsymbol{x}_0)}, s_\theta(\boldsymbol{x}_t, t)_{\boldsymbol{y}}\right). \tag{75}$$

When $\bar{\ell}$ is linear with respect to ratios, like in the case of Score Entropy (Lou et al., 2024), then we can pull the inner expectation (sum) outside and have

$$\mathbb{E}_{\boldsymbol{x}_t \sim p_t(\boldsymbol{x}_t)} \bar{\ell}\left(\frac{p_t(\boldsymbol{y})}{p_t(\boldsymbol{x}_t)}, s_\theta(\boldsymbol{x}_t, t)_{\boldsymbol{y}}\right) = \mathbb{E}_{\boldsymbol{x}_t \sim p_t(\boldsymbol{x}_t)} \mathbb{E}_{\boldsymbol{x}_0 \sim p_{0|t}(\boldsymbol{x}_0|\boldsymbol{x}_t)} \bar{\ell}\left(\frac{p_{t|0}(\boldsymbol{y}|\boldsymbol{x}_0)}{p_{t|0}(\boldsymbol{x}_t|\boldsymbol{x}_0)}, s_\theta(\boldsymbol{x}_t, t)_{\boldsymbol{y}}\right), \tag{76}$$

which of course is equal to

$$\mathbb{E}_{\boldsymbol{x}_0 \sim p_0(\boldsymbol{x}_0)} \mathbb{E}_{\boldsymbol{x}_t \sim p_{t|0}(\boldsymbol{x}_t|\boldsymbol{x}_0)} \bar{\ell}\left(\frac{p_{t|0}(\boldsymbol{y}|\boldsymbol{x}_0)}{p_{t|0}(\boldsymbol{x}_t|\boldsymbol{x}_0)}, s_\theta(\boldsymbol{x}_t, t)_{\boldsymbol{y}}\right). \tag{77}$$

### A.2.2 DERIVING CROSS-ENTROPY FROM RATIO MATCHING

In order to go from Equation (72) to (74) we used the fact that

$$\frac{p_t(\boldsymbol{y})}{p_t(\boldsymbol{x}_t)} = \Sigma_{\boldsymbol{x}_0} \frac{p_{t|0}(\boldsymbol{y}|\boldsymbol{x}_0)}{p_{t|0}(\boldsymbol{x}_t|\boldsymbol{x}_0)} p_{0|t}(\boldsymbol{x}_0|\boldsymbol{x}_t). \tag{78}$$

Now, we select position $i$ on the sequence with length $L$. We want to find an expression of the ratios $\frac{p_t(\boldsymbol{y})}{p_t(\boldsymbol{x}_t)}$ for all sequences $\boldsymbol{y}$, which differ with $\boldsymbol{x}_t$ only on position $i$. From above we have

$$\frac{p_t(\boldsymbol{y})}{p_t(\boldsymbol{x}_t)} = \Sigma_{\boldsymbol{x}_0} \frac{p_{t|0}(\boldsymbol{y}|\boldsymbol{x}_0)}{p_{t|0}(\boldsymbol{x}_t|\boldsymbol{x}_0)} p_{0|t}(\boldsymbol{x}_0|\boldsymbol{x}_t), \tag{79}$$

where

$$\frac{p_{t|0}(\boldsymbol{y}|\boldsymbol{x}_0)}{p_{t|0}(\boldsymbol{x}_t|\boldsymbol{x}_0)} = \frac{p_{t|0}\left((\boldsymbol{y}^{(0)}, \boldsymbol{y}^{(1)}, ..., \boldsymbol{y}^{(i)}, ..., \boldsymbol{y}^{(V-1)})|(\boldsymbol{x}_0^{(0)}, \boldsymbol{x}_0^{(1)}, ..., \boldsymbol{x}_0^{(i)}, ..., \boldsymbol{x}_0^{(V-1)})\right)}{p_{t|0}\left((\boldsymbol{x}_t^{(0)}, \boldsymbol{x}_t^{(1)}, ..., \boldsymbol{x}_t^{(i)}, ..., \boldsymbol{x}_t^{(V-1)})|(\boldsymbol{x}_0^{(0)}, \boldsymbol{x}_0^{(1)}, ..., \boldsymbol{x}_0^{(i)}, ..., \boldsymbol{x}_0^{(V-1)})\right)} \tag{80}$$

and due to independence between entries in the forward process we have

$$\frac{p_{t|0}(\boldsymbol{y}|\boldsymbol{x}_0)}{p_{t|0}(\boldsymbol{x}_t|\boldsymbol{x}_0)} = \frac{\prod p_{t|0}(\boldsymbol{y}^j|\boldsymbol{x}_0{}^j)}{\prod p_{t|0}(\boldsymbol{x}_t{}^j|\boldsymbol{x}_0{}^j)} = \prod \frac{p_{t|0}(\boldsymbol{y}^j|\boldsymbol{x}_0{}^j)}{p_{t|0}(\boldsymbol{x}_t{}^j|\boldsymbol{x}_0{}^j)} = \frac{p_{t|0}(\boldsymbol{y}^i|\boldsymbol{x}_0{}^i)}{p_{t|0}(\boldsymbol{x}_t{}^i|\boldsymbol{x}_0{}^i)}, \tag{81}$$

as the rest of these ratios are 1, since $\boldsymbol{y}$ differs with $\boldsymbol{x}_t$ only on position $i$. Therefore we have

$$\frac{p_t(\boldsymbol{y})}{p_t(\boldsymbol{x}_t)} = \Sigma_{\boldsymbol{x}_0} \frac{p_{t|0}(\boldsymbol{y}^i|\boldsymbol{x}_0{}^i)}{p_{t|0}(\boldsymbol{x}_t{}^i|\boldsymbol{x}_0{}^i)} p_{0|t}(\boldsymbol{x}_0|\boldsymbol{x}_t), \tag{82}$$

where $p_{0|t}(\boldsymbol{x}_0|\boldsymbol{x}_t) = p_{0|t}(\boldsymbol{x}_0{}^{(0)}, \boldsymbol{x}_0{}^{(1)}, ..., \boldsymbol{x}_0{}^{(i)}, ..., \boldsymbol{x}_0{}^{(V-1)}|\boldsymbol{x}_t)$ and where the expression inside the sum $\frac{p_{t|0}(\boldsymbol{y}^i|\boldsymbol{x}_0{}^i)}{p_{t|0}(\boldsymbol{x}_t{}^i|\boldsymbol{x}_0{}^i)}$, depends only on $\boldsymbol{x}_0{}^i$ and not on the rest of $\boldsymbol{x}_0{}^j$ for $j \neq i$. Thus,

$$\Sigma_{\boldsymbol{x}_0} \frac{p_{t|0}(\boldsymbol{y}^i|\boldsymbol{x}_0{}^i)}{p_{t|0}(\boldsymbol{x}_t{}^i|\boldsymbol{x}_0{}^i)} p_{0|t}(\boldsymbol{x}_0|\boldsymbol{x}_t) = \Sigma_{\boldsymbol{x}_0{}^i} \frac{p_{t|0}(\boldsymbol{y}^i|\boldsymbol{x}_0{}^i)}{p_{t|0}(\boldsymbol{x}_t{}^i|\boldsymbol{x}_0{}^i)} \Sigma_{\boldsymbol{x}_0{}^{(0)},...,\boldsymbol{x}_0{}^{(i-1)},\boldsymbol{x}_0{}^{(i+1)},...,\boldsymbol{x}_0{}^{(V-1)}} p_{0|t}(\boldsymbol{x}_0|\boldsymbol{x}_t), \tag{83}$$

which implies that

$$\frac{p_t(\boldsymbol{y})}{p_t(\boldsymbol{x}_t)} = \Sigma_{\boldsymbol{x}_0{}^i} \frac{p_{t|0}(\boldsymbol{y}^i|\boldsymbol{x}_0{}^i)}{p_{t|0}(\boldsymbol{x}_t{}^i|\boldsymbol{x}_0{}^i)} p_{0|t}^i(\boldsymbol{x}_0|\boldsymbol{x}_t), \tag{84}$$

or more clearly

$$\frac{p_t(\boldsymbol{y})}{p_t(\boldsymbol{x}_t)} = \Sigma_{h \in [V]} \frac{p_{t|0}(\boldsymbol{y}^i|h)}{p_{t|0}(\boldsymbol{x}_t{}^i|h)} p_{0|t}^i(h|\boldsymbol{x}_t). \tag{85}$$

If we learn the $V$ probabilities $[p_{0|t}^i(0|\boldsymbol{x}_t), p_{0|t}^i(1|\boldsymbol{x}_t), ..., p_{0|t}^i((V-1)|\boldsymbol{x}_t)]$, since we analytically have $\frac{p_{t|0}(\boldsymbol{y}^i|h^i)}{p_{t|0}(\boldsymbol{x}_t{}^i|h^i)}$ (from the matrix exponential), then we we will have the ratio $\frac{p_t(\boldsymbol{y})}{p_t(\boldsymbol{x}_t)}$. We can choose another sequence $\boldsymbol{z}$ which also differs only at position $i$ from $\boldsymbol{x}_t$. Then we will have

$$\frac{p_t(\boldsymbol{z})}{p_t(\boldsymbol{x}_t)} = \Sigma_{h \in [V]} \frac{p_{t|0}(\boldsymbol{z}^i|h)}{p_{t|0}(\boldsymbol{x}_t{}^i|h)} p_{0|t}^i(h|\boldsymbol{x}_t), \tag{86}$$

which highlights the fact that even though the ratios might change, the same $[p_{0|t}^i(0|\boldsymbol{x}_t), p_{0|t}^i(1|\boldsymbol{x}_t), ..., p_{0|t}^i((V-1)|\boldsymbol{x}_t)]$ appear as long as the selected position $i$ in the sequence remains unchanged. Therefore if we learn these probabilities for a position, we will have the ratios of all neighbours that differ only in that position, that is, we have $V$ ratios, by modeling these $V$ probabilities.

There is nothing special about position $i$ and we can choose to model $V$ probabilities for each $L$ positions. In order to do so, we define a neural network $f_\theta(\boldsymbol{x}_t, t)$ whose output is $L \times V$, where entry $(i, h)$ of the output (that is $f_\theta^i(\boldsymbol{x}_t, t)[h]$), predicts $p_{0|t}^i(h|\boldsymbol{x}_t)$ the probability of the entry $i$ of $\boldsymbol{x}_t$ (position $i$ of the current sequence), having being perturbed from token with id $h$ in the original sequence $\boldsymbol{x}_0$, given that we are at sequence $\boldsymbol{x}_t$ right now. That is, the model is directly trying to predict from where each of the tokens in the current sequence came from. Therefore we will reparametrize our ratio model as follows

$$s_\theta^i(\boldsymbol{x}_t, t)[h] = s_\theta^i(\boldsymbol{x}_t, t)_h = \Sigma_{h \in [V]} \frac{p_{t|0}(\boldsymbol{z}^i|h)}{p_{t|0}(\boldsymbol{x}_t{}^i|h)} f_\theta^i(\boldsymbol{x}_t, t)[h], \tag{87}$$

where $f_\theta^i(\boldsymbol{x}_t, t)$ are the outputs of the softmax at the end, and where for each $i$, distribution $f_\theta^i(\boldsymbol{x}_t, t)$ should match $p_{0|t}^i(\cdot|\boldsymbol{x}_t)$. This happens when $D_{KL}(p_{0|t}^i(\cdot|\boldsymbol{x}_t)|f_\theta^i(\boldsymbol{x}_t, t)) = 0$, thus we minimize,

$$\sum_{i=1}^{L} \mathbb{E}_{t \sim U(0,1)} w(t) \mathbb{E}_{\boldsymbol{x}_t \sim p_t(\boldsymbol{x}_t)} D_{KL}(p_{0|t}^i(\cdot|\boldsymbol{x}_t)||f_\theta^i(\boldsymbol{x}_t, t)). \tag{88}$$

Since $D_{KL}(p_{0|t}^i(\cdot|\boldsymbol{x}_t)||f_\theta^i(\boldsymbol{x}_t, t)) = H(p_{0|t}^i(\cdot|\boldsymbol{x}_t), f_\theta^i(\boldsymbol{x}_t, t)) - C = \mathbb{E}_{p_{0|t}^i(h|\boldsymbol{x}_t)} \log f_\theta^i(\boldsymbol{x}_t, t)[h] - C$ the loss function above has the same gradients with regards to network parameters as

$$-\sum_{i=1}^{L} \mathbb{E}_{t \sim U(0,1)} w(t) \mathbb{E}_{\boldsymbol{x}_t \sim p_t(\boldsymbol{x}_t)} \mathbb{E}_{h \sim p_{0|t}^i(h|\boldsymbol{x}_t)} \log f_\theta^i(\boldsymbol{x}_t, t)[h], \tag{89}$$

which is equal to

$$-\sum_{i=1}^{L} \mathbb{E}_{t\sim U(0,1)} \mathbb{E}_{\boldsymbol{x}_t \sim p_t(\boldsymbol{x}_t)} \mathbb{E}_{\boldsymbol{x}_0 \sim p_{0|t}(\boldsymbol{x}_0|\boldsymbol{x}_t)} w(t) \log f_\theta^i(\boldsymbol{x}_t, t)[\boldsymbol{x}_0^i]. \tag{90}$$

Thus, as in (Campbell et al., 2022), the loss we utilize in our training is $L_{ll}$,

$$L_{ll} = -\mathbb{E}_{t\sim U(0,1)} \mathbb{E}_{\boldsymbol{x}_0 \sim p_0(\boldsymbol{x}_0)} \mathbb{E}_{\boldsymbol{x}_t \sim p_{t|0}(\boldsymbol{x}_t|\boldsymbol{x}_0)} \sum_{i=1}^{L} w(t) \log f_\theta^i(\boldsymbol{x}_t, t)[\boldsymbol{x}_0^i]. \tag{91}$$

We can also motivate this loss from the score entropy in Equation (77). Indeed, since $\ell$ and $\bar{\ell}$ differ by a constant, its gradients are the same as those of

$$\mathbb{E}_{\boldsymbol{x}_0 \sim p_0(\boldsymbol{x}_0)} \mathbb{E}_{\boldsymbol{x}_t \sim p_{t|0}(\boldsymbol{x}_t|\boldsymbol{x}_0)} \ell\left(\frac{p_{t|0}(\boldsymbol{y}|\boldsymbol{x}_0)}{p_{t|0}(\boldsymbol{x}_t|\boldsymbol{x}_0)}, \Sigma_{h\in[V]} \frac{p_{t|0}(\boldsymbol{y}^i|h)}{p_{t|0}(\boldsymbol{x}_t^i|h)} f_\theta^i(\boldsymbol{x}_t, t)[h]\right), \tag{92}$$

and as before since $\boldsymbol{y}$ and $\boldsymbol{x}_t$ differ at only one token we get

$$= \mathbb{E}_{\boldsymbol{x}_0 \sim p_0(\boldsymbol{x}_0)} \mathbb{E}_{\boldsymbol{x}_t \sim p_{t|0}(\boldsymbol{x}_t|\boldsymbol{x}_0)} \ell\left(\frac{p_{t|0}(\boldsymbol{y}^i|\boldsymbol{x}_0^i)}{p_{t|0}(\boldsymbol{x}_t^i|\boldsymbol{x}_0^i)}, \Sigma_{h\in[V]} \frac{p_{t|0}(\boldsymbol{y}^i|h)}{p_{t|0}(\boldsymbol{x}_t^i|h)} f_\theta^i(\boldsymbol{x}_t, t)[h]\right). \tag{93}$$

The function $\ell(a, b)$ is clearly minimized when $a = b$, in which case $\ell(a, b) = 0$. In our case,

$$\ell\left(\frac{p_{t|0}(\boldsymbol{y}^i|\boldsymbol{x}_0^i)}{p_{t|0}(\boldsymbol{x}_t^i|\boldsymbol{x}_0^i)}, \Sigma_{h\in[V]} \frac{p_{t|0}(\boldsymbol{y}^i|h)}{p_{t|0}(\boldsymbol{x}_t^i|h)} f_\theta^i(\boldsymbol{x}_t, t)[h]\right) = 0, \tag{94}$$

for

$$\frac{p_{t|0}(\boldsymbol{y}^i|\boldsymbol{x}_0^i)}{p_{t|0}(\boldsymbol{x}_t^i|\boldsymbol{x}_0^i)} = \Sigma_{h\in[V]} \frac{p_{t|0}(\boldsymbol{y}^i|h)}{p_{t|0}(\boldsymbol{x}_t^i|h)} f_\theta^i(\boldsymbol{x}_t, t)[h], \tag{95}$$

which happens when $f_t^i(\boldsymbol{x}_t, \theta)[h \neq \boldsymbol{x}_0^i] = 0$ and $f_t^i(\boldsymbol{x}_t, \theta)[\boldsymbol{x}_0^i] = 1$, that is when our probability prediction matches the one hot encoding used in cross entropy. Thus we train our model with cross entropy, which simplifies the job of the network of learning complex ratios, as the conditional ratios now do not participate in the loss. We only add them during sampling to the weighted sum in order to predict the marginal ratios, since we have analytic expressions for the conditional ratios. Therefore, we train the model by minimizing

$$-\mathbb{E}_{t\sim U(0,1)} \mathbb{E}_{\boldsymbol{x}_0 \sim p_0(\boldsymbol{x}_0)} \mathbb{E}_{\boldsymbol{x}_t \sim p_{t|0}(\boldsymbol{x}_t|\boldsymbol{x}_0)} \sum_{i=1}^{L} w(t) \log f_\theta^i(\boldsymbol{x}_t, t)[\boldsymbol{x}_0^i]. \tag{96}$$

### A.2.3 RELATION TO CROSS ENTROPY IN CONTINUOUS DIFFUSION

Cross entropy has also been used in continuous diffusion models applied to Language Modelling (Dieleman et al., 2022). Here, we draw parallels between the two approaches. We denote a sequence of $L$ tokens at time $t$ as $\boldsymbol{x}_t = (\boldsymbol{x}_t^1, ..., \boldsymbol{x}_t^L)$, where each token is embedded in a $D-$dimensional space $\boldsymbol{x}_t^i \in \mathbb{R}^D$. That is, each sequence can be seen as an $L \times D$ vector, created from concatenating the embeddings of each token. The score that generates the reverse process, is therefore a $L \times D$ vector $s_\theta(x_t, t)$, which approximates:

$$\nabla_{\boldsymbol{x}_t} \log p_t(\boldsymbol{x}_t) = \int \nabla_{\boldsymbol{x}_t} \log p_t(\boldsymbol{x}_t|\boldsymbol{x}_0) p_t(\boldsymbol{x}_0|\boldsymbol{x}_t) d\boldsymbol{x}_0 = \tag{97}$$

$$\int \nabla_{\boldsymbol{x}_t} \log e^{\frac{-||\boldsymbol{x}_t - \boldsymbol{x}_0||^2}{2\sigma_t^2}} p_t(\boldsymbol{x}_0|\boldsymbol{x}_t) d\boldsymbol{x}_0 = \int \frac{\boldsymbol{x}_0 - \boldsymbol{x}_t}{\sigma_t^2} p_t(\boldsymbol{x}_0|\boldsymbol{x}_t) d\boldsymbol{x}_0 = \int \frac{\boldsymbol{x}_0}{\sigma_t^2} p_t(\boldsymbol{x}_0|\boldsymbol{x}_t) d\boldsymbol{x}_0 - \frac{\boldsymbol{x}_t}{\sigma_t^2}. \tag{98}$$

From above, it is clear that

$$\nabla_{\boldsymbol{x}_t^i} \log p_t(\boldsymbol{x}_t) = \int \frac{\boldsymbol{x}_0^i}{\sigma_t^2} p_t(\boldsymbol{x}_0|\boldsymbol{x}_t) d\boldsymbol{x}_0 - \frac{\boldsymbol{x}_t^i}{\sigma_t^2} = \int \frac{\boldsymbol{x}_0^i}{\sigma_t^2} \left(\int p_t(\boldsymbol{x}_0|\boldsymbol{x}_t) d\boldsymbol{x}_0^{-i}\right) d\boldsymbol{x}_0^i - \frac{\boldsymbol{x}_t^i}{\sigma_t^2}, \tag{99}$$

where $\boldsymbol{x}_0^{-i}$ denotes all entries of $\boldsymbol{x}_0$ without the ones of $\boldsymbol{x}_0^i$. Therefore

$$s_\theta(\boldsymbol{x}_t, t)_{(i1, i2, .., iD)} \approx \nabla_{\boldsymbol{x}_t^i} \log p_t(\boldsymbol{x}_t) = \int \frac{\boldsymbol{x}_0^i}{\sigma_t^2} p_t(\boldsymbol{x}_0^i|\boldsymbol{x}_t) d\boldsymbol{x}_0^i - \frac{\boldsymbol{x}_t^i}{\sigma_t^2}. \tag{100}$$

Clearly, all that need to be learned in order to model the score for the dimensions of the token at position $i$ are the $V$ probabilities $p_t(\boldsymbol{x}_0^i|\boldsymbol{x}_t)$. For $L$ such tokens (to model the score of the entire sequence) one simply needs to model $L \times V$ probabilities using cross entropy as in the discrete case.

### A.3 ROULETTE DISCRETE DIFFUSION

The roulette transition-rate matrix is

$$
\boldsymbol{Q}_{roulette}^{tok} =
\begin{pmatrix}
\frac{1}{n-1}(1-p_m)-1 & \frac{1}{n-1}(1-p_m) & ... & \frac{1}{n-1}(1-p_m) & 0 \\
\frac{1}{n-1}(1-p_m) & \frac{1}{n-1}(1-p_m)-1 & ... & \frac{1}{n-1}(1-p_m) & 0 \\
... & ... & ... & ... & ... \\
\frac{1}{n-1}(1-p_m) & \frac{1}{n-1}(1-p_m) & ... & \frac{1}{n-1}(1-p_m)-1 & 0 \\
p_m & p_m & ... & p_m & 1-1
\end{pmatrix}_{n \times n},
\tag{101}
$$

where $n-1 = V$ is the number of tokens in our vocabulary. We add a special token for the absorb (mask) state, therefore increasing the number of total states to $n$, with the token id $n$ corresponding to the absorbed state. Unfortunately, we cannot derive the matrix exponential as in the case of absorb and uniform diffusion, since the corresponding matrix $\boldsymbol{P}$ of $\boldsymbol{Q}$ is not idempotent. Therefore, we will first derive the exponential matrix of the uniform and absorb diffusion manually, in order to motivate the manual derivation of the exponential matrix in the roulette case.

#### A.3.1 DERIVING THE EXPONENTIAL MATRIX OF THE ABSORB DIFFUSION

The transition matrix in the absorb case is

$$
\boldsymbol{Q}_{abs}^{tok} =
\begin{pmatrix}
0-1 & 0 & ... & 0 & 0 \\
0 & 0-1 & ... & 0 & 0 \\
... & ... & ... & ... & ... \\
0 & 0 & ... & 0-1 & 0 \\
1 & 1 & ... & 1 & 1-1
\end{pmatrix}_{n \times n}.
\tag{102}
$$

Clearly this is $\boldsymbol{Q}_{roulette}^{tok}$ when $p_m = 1$.
If we are at state $i$, the probability of moving at state $n$ (the absorb/mask state) for a time-step of size $\epsilon$ is $\epsilon$. We discretize the time interval into discrete steps that are multiples of $\epsilon$, that is $[0, \tau] \to \{\epsilon, ..., j\epsilon, ... \lfloor \frac{\tau}{\epsilon} \rfloor \epsilon\}$ and denote the event of jumping from $i \neq n$ to $n$ at time $j\epsilon$ as $a_j$. The intersection of any of these two events is empty and their union is the space of all possibilities, thus the probability of being at $n$ at time $\tau$ is

$$
p(x_\tau = n) = p(\{x_\tau = n\} \cap \Omega) = p(\{x_\tau = n\} \cap \{\cup a_j\}) = p(\cup_{j\epsilon < \tau}\{a_j\})
\tag{103}
$$

$$
p(x_\tau = n) = \Sigma_{j < \frac{\tau}{\epsilon}} p(a_j) = \epsilon + \epsilon(1-\epsilon) + \epsilon(1-\epsilon)^2 + ... + \epsilon(1-\epsilon)^{\lfloor \frac{\tau}{\epsilon} \rfloor - 1}
\tag{104}
$$

$$
p(x_\tau = n) = \epsilon \frac{1 - (1-\epsilon)^{\lfloor \frac{\tau}{\epsilon} \rfloor}}{1 - (1-\epsilon)} = 1 - (1-\epsilon)^{\lfloor \frac{\tau}{\epsilon} \rfloor}
\tag{105}
$$

Taking the limit $\epsilon \to 0$, we get $p(x_\tau = n) = 1 - e^{-\tau}$. Thus, assuming that we start at state $i$, we have $p_{\tau|0}(j \notin \{n, i\}|i \neq n) = 0$, $p_{\tau|0}(j = i|i \neq n) = e^{-\tau}$ and $p_{\tau|0}(j = n|i \neq n) = 1 - e^{-\tau}$. This defines the column $i \neq n$ of the exponential $e^{\tau \boldsymbol{Q}^{tok}}$. Iterating though different $i$ we construct the entire exponential matrix, except its last column which is a simple one hot encoding at $n$. Setting $\tau = \sigma_t$ finishes the derivation.

#### A.3.2 DERIVING THE EXPONENTIAL OF THE UNIFORM DIFFUSION

$$
\boldsymbol{Q}_{unif}^{tok} =
\begin{pmatrix}
\frac{1}{n-1}-1 & \frac{1}{n-1} & ... & \frac{1}{n-1} \\
\frac{1}{n-1} & \frac{1}{n-1}-1 & ... & \frac{1}{n-1} \\
... & ... & ... & ... \\
\frac{1}{n-1} & \frac{1}{n-1} & ... & \frac{1}{n-1}-1
\end{pmatrix}_{V \times V}
\tag{106}
$$

Clearly this is $\boldsymbol{Q}_{roulette}^{tok}$ when $p_m = 0$, without the absorption row and column, as our vocabulary size is $V = n - 1$ since we do not have the special token. As before we discretize the time interval into discrete steps that are multiples of $\epsilon$, that is $[0, \tau] \to \{\epsilon, ..., j\epsilon, ... \lfloor \frac{\tau}{\epsilon} \rfloor \epsilon\}$. We assume that initially (at time 0) we start at position $i$. We wish to find the probability of being at state $k \neq i$ at time $\lfloor \frac{\tau}{\epsilon} \rfloor \epsilon$. For the sake of simplicity, we abuse notation by denoting $p_{(\lfloor \frac{\tau}{\epsilon} \rfloor \epsilon - m\epsilon)|0}(j|i)$ as $\bar{p}_{\tau - m\epsilon}(j|i)$. By symmetry we know that for every $j \neq i$ the probability $\bar{p}_{\tau - \epsilon}(j|i)$ is the same. Thus the probability of $\bar{p}_\tau(j|i)$ is the sum of the following components:

- probability of being at $j \notin \{i, k\}$ at time $\tau - \epsilon$, i.e, $(\bar{p}_{\tau-\epsilon}(j|i))$, times $n - 3 = V - 2$ such states, times probability of moving $(\epsilon)$, times probability of hitting $k$ on that move $(\frac{1}{n-1})$.

- probability of being at $j = i$ at time $\tau - \epsilon$, i.e, $(1 - (n-2) \cdot \bar{p}_{\tau-\epsilon}(j|i))$, times probability of moving $(\epsilon)$, times probability of hitting $k$ on that move $(\frac{1}{n-1})$.

- probability of being at $j = k$ at time $\tau - \epsilon$, i.e, $(\bar{p}_{\tau-\epsilon}(j|i))$, times of staying there $(1 - \epsilon)$.

- probability of being at $j = k$ at time $\tau - \epsilon$, i.e, $(\bar{p}_{\tau-\epsilon}(j|i))$, times of moving $(\epsilon)$, times of hitting itself on this move $(\frac{1}{n-1})$.

We write them down mathematically and get:

$$\bar{p}_\tau(k|i) = \bar{p}_{\tau-\epsilon}(j|i)\epsilon\frac{1}{n-1}(n-3) + (1 - (n-2)\bar{p}_{\tau-\epsilon}(j|i))\epsilon\frac{1}{n-1} + \tag{107}$$

$$+\bar{p}_{\tau-\epsilon}(j|i)(1-\epsilon) + \bar{p}_{\tau-\epsilon}(j|i)\frac{1}{n-1}\epsilon, \tag{108}$$

thus

$$\bar{p}_\tau(k|i) = \frac{1}{n-1}\epsilon + \bar{p}_{\tau-\epsilon}(j|i)(1-\epsilon). \tag{109}$$

Since by definition $k \neq i$ and $j \neq i$, then by symmetry we have $\bar{p}_\tau(k|i) = \bar{p}_\tau(j|i)$. Therefore we get the recursion:

$$p_{\lfloor\frac{\tau}{\epsilon}\rfloor\epsilon|0}(k|i) = \frac{1}{n-1}\epsilon + p_{(\lfloor\frac{\tau}{\epsilon}\rfloor\epsilon-\epsilon)|0}(k|i), \tag{110}$$

which if we fully develop becomes:

$$p_{\lfloor\frac{\tau}{\epsilon}\rfloor\epsilon|0}(k|i) = \frac{1}{n-1}\epsilon(1 + (1-\epsilon) + (1-\epsilon)^2 + ... + (1-\epsilon)^{\lfloor\frac{\tau}{\epsilon}\rfloor-1}), \tag{111}$$

which is equal to:

$$p_{\lfloor\frac{\tau}{\epsilon}\rfloor\epsilon|0}(k|i) = \frac{1}{n-1}\epsilon\frac{1 - (1-\epsilon)^{\lfloor\frac{\tau}{\epsilon}\rfloor}}{1 - (1-\epsilon)} = \frac{1}{n-1}(1 - (1-\epsilon)^{\lfloor\frac{\tau}{\epsilon}\rfloor}). \tag{112}$$

Taking the limit $\epsilon \to 0$ gives $p_{\tau|0}(k \neq i|i) = \frac{1}{n-1}(1 - e^{-\tau})$ and $p_{\tau|0}(i|i) = 1 - \frac{n-2}{n-1}(1 - e^{-\tau})$, which gives the $i$th column of the exponential matrix. Setting $\tau = \sigma_t$ finishes the derivation.

### A.3.3 DERIVING THE EXPONENTIAL OF THE ROULETTE DIFFUSION

First, we notice that we must start at a state that is different from the absorb state (there are no masked tokens in the training data). We can split the states into two groups, the non-absorbing states, and the absorbing state. We consider the non-absorbing states as a single super-state and the absorbing state as the other super-state. From our derivations in Appendix A.3.1, we know how to derive the probability of being at the absorb super-state at time $\tau$. The only difference is that previously, if we moved, it would be certain we would move to the absorption super-state, while in our case, we can move to a different state in our non-absorb super-state. The possibility of moving to the absorb super-state in a $\epsilon$ time step, changes from $\epsilon$ to $\epsilon p_m$, and the formula becomes

$$p(x_\tau = n) = \Sigma_{j<\frac{\tau}{\epsilon}}p(a_j) = \epsilon p_m + \epsilon p_m(1-\epsilon p_m) + \epsilon p_m(1-\epsilon p_m)^2 + ... + \epsilon p_m(1-\epsilon p_m)^{\lfloor\frac{\tau}{\epsilon}\rfloor-1} \tag{113}$$

$$p(x_\tau = n) = \epsilon p_m\frac{1 - (1-\epsilon p_m)^{\frac{1}{\epsilon p_m}\lfloor\frac{\tau}{\epsilon}\rfloor\epsilon p_m}}{1 - (1-\epsilon p_m)} = 1 - (1-\epsilon p_m)^{\frac{1}{\epsilon p_m}\lfloor\frac{\tau}{\epsilon}\rfloor\epsilon p_m} \tag{114}$$

which implies that $p(x_\tau = n) = 1 - e^{-\tau p_m}$. This means that at time $\tau$ we are in one of the non-absorption states with a probability of $e^{-\tau p_m}$. Now, as before (Appendix A.3.2), we want to find the probability of being at each non-absorption state at time $\tau$ given that we stated a state $i$. We can construct these probabilities using a similar approach as before, but with two key differences. First, because we are conditioning on being in the non-absorption super-state, all probabilities must be multiplied by $e^{-\tau p_m}$. Second, this conditioning implicitly provides some information about the probability of moving when going from time $\tau - \epsilon$ to time $\tau$. In the uniform case, for a time step $\epsilon$ this probability was $\epsilon$, but now conditioning on the fact that we are not at the absorption state at time $\tau$ provides extra information on whether we moved when going from $\tau - \epsilon$ to $\tau$. Indeed, given

this extra information, we expect the probability of having moved when going form $\tau - \epsilon$ to $\tau$ to be reduced. This becomes obvious when $p_m = 1$, as then we are in the case of the absorb diffusion, and saying that at time $\tau$ we are not at the absorption state, immediately implies that we did not move, as there are no other possibilites. As $p_m$ decreases, the magnitude of this information decreases. So instead of writing the probability of moving (during an $\epsilon$ time step) with $\epsilon$ as we did previously, we write this probability with $\delta(\epsilon)$, and as before we can derive that

$$\bar{p}_\tau(k|i) = \bar{p}_{\tau-\epsilon}(j|i)\delta\frac{1}{n-1}(n-3) + (1-(n-2)\bar{p}_{\tau-\epsilon}(j|i))\delta\frac{1}{n-1}+ \tag{115}$$

$$+\bar{p}_{\tau-\epsilon}(j|i)(1-\delta) + \bar{p}_{\tau-\epsilon}(j|i)\frac{1}{n-1}\delta \tag{116}$$

thus

$$\bar{p}_\tau(k|i) = \frac{1}{n-1}\delta + \bar{p}_{\tau-\epsilon}(k|i)(1-\delta) \tag{117}$$

and by the same recursion trick as before,

$$p_{\lfloor\frac{\tau}{\epsilon}\rfloor\epsilon|0}(k|i) = \frac{1}{n-1}\delta\frac{1-(1-\delta)^{\lfloor\frac{\tau}{\epsilon}\rfloor}}{1-(1-\delta)} = \frac{1}{n-1}(1-(1-\delta)^{\lfloor\frac{\tau}{\epsilon}\rfloor}). \tag{118}$$

Now we derive the expression of $\delta$. We write the move event when going from $\tau - \epsilon$ to $\tau$ with $A$ and not going at the absorbed state at time $\tau$ with $B$

$$\delta = p(A|B) = \frac{p(A)}{p(B)}p(B|A) = \frac{p(B|A)p(A)}{p(B|A)p(A) + p(B|A^C)p(A^C)} = \tag{119}$$

$$= \frac{(1-p_m)\epsilon}{(1-p_m)\epsilon + 1(1-\epsilon)} = \alpha\epsilon \tag{120}$$

where $\alpha = \frac{(1-p_m)}{(1-p_m)\epsilon+1(1-\epsilon)}$ goes to $1 - p_m$ when $\epsilon$ goes to 0, therefore finally

$$p_{\lfloor\frac{\tau}{\epsilon}\rfloor\epsilon|0}(k|i) = \frac{1}{n-1}(1-(1-\delta)^{\lfloor\frac{\tau}{\epsilon}\rfloor}) = \frac{1}{n-1}(1-(1-\delta)^{\frac{1}{\delta}\delta\lfloor\frac{\tau}{\epsilon}\rfloor}) = \frac{1}{n-1}(1-(1-\delta)^{\frac{1}{\delta}\alpha\epsilon\lfloor\frac{\tau}{\epsilon}\rfloor}). \tag{121}$$

Therefore $p_{\tau|0}(j \notin \{i,n\}|i) = e^{-\tau p_m}\frac{1}{n-1}(1 - e^{(1-p_m)\tau})$, and $p_{\tau|0}(n|i) = 1 - e^{-\tau p_m}$, and $p_{\tau|0}(i|i) = e^{-\tau p_m}(1 - \frac{n-2}{n-1}(1 - e^{(1-p_m)\tau}))$, which gives the $i$th column of the exponential matrix, when $i \neq n$. In case $i = n$, the column of the exponential matrix is simply the one hot encoding at $n$. Setting $\tau = \sigma_t$ finishes the derivation.

**Proposition 5** *If we denote with $\boldsymbol{Y}_t$ the matrix exponential of $\sigma_t\boldsymbol{Q}^{tok}_{roulette} = \sigma_t(\boldsymbol{I} - \boldsymbol{P}^{tok}_{roulette})$, then $\boldsymbol{Y}_t(i \notin \{j,n\}, j \neq n) = e^{-\sigma_t p_m}\frac{1}{n-1}(1 - e^{-(1-p_m)\sigma_t})$, $\boldsymbol{Y}_t(i \neq n, i \neq n) = e^{-\sigma_t p_m}(1 - \frac{n-2}{n-1}(1 - e^{-(1-p_m)\sigma_t}))$, $\boldsymbol{Y}_t(n, j \neq n) = 1 - e^{-\sigma_t p_m}$, $\boldsymbol{Y}_t(i \neq n, n) = 0$, and $\boldsymbol{Y}_t(n, n) = 1$.*

### A.3.4 CONTROLLING THE NUMBER OF THE CORRECTED TOKENS IN THE REVERSE PROCESS

We wish to see how the choice of $p_m$ corresponds to the probability of a token moving uniformly at least once before being masked, given some diffusion interval from $[0, T]$. This should correspond to the probability of a token being corrected after it is unmasked in the reverse process, therefore enabling us to control the expected number of tokens to be corrected.

As before, we discretize the time interval into subintervals of length $\epsilon$. The event of a token moving before getting masked will be denoted by $X$, and the event of a token being masked at time $j\epsilon$ is denoted with $A_j$. Therefore

$$p(X) = p(X \cap \Omega) = p(X \cap \{\cup A_j\}) = p(\cup_{j\epsilon < T} X \cap A_j) \tag{122}$$

$$p(X) = \Sigma_{j<\lfloor\frac{T}{\epsilon}\rfloor}p(X|A_j)p(A_j). \tag{123}$$

$p(A_j)$ is the probability of being at the absorbtion state at $j\epsilon$ and not being there at $(j-1)\epsilon$. This probability is $p(A_j) = (1-e^{-j\epsilon p_m}) - (1-e^{-(j-1)\epsilon p_m}) = e^{-(j-1)\epsilon p_m} - e^{-j\epsilon p_m}$ therefore $p(A_j) = e^{-(j-1)\epsilon p_m}(1 - e^{-\epsilon p_m}) \approx e^{-(j-1)\epsilon p_m}\epsilon p_m$. We write $\tau_i = i\epsilon$, and thus we have

$$p(X) = \Sigma_{j<\lfloor\frac{T}{\epsilon}\rfloor}p(X|A_j)p(A_j) = \Sigma_{j<\lfloor\frac{T}{\epsilon}\rfloor}\frac{n-2}{n-1}(1 - e^{-(1-p_m)\tau_{j-1}}) \cdot e^{-\tau_{j-1}p_m}\epsilon p_m \tag{124}$$

which is a Riemann sum. Taking the limit we get the following integral

$$p(X) = p_m \frac{n-2}{n-1} \int_0^T \left(1 - e^{-(1-p_m)x}\right) \cdot e^{-xp_m} dx, \tag{125}$$

whose solution is

$$p(X) = p_m \frac{n-2}{n-1}\left(e^{-T} - \frac{1}{p_m}e^{-Tp_m} + \frac{1}{p_m} - 1\right). \tag{126}$$

Setting $t = \sigma_\tau$, we get

$$p(X) = \frac{n-2}{n-1}\left(e^{-\sigma_1}p_m - e^{-\sigma_1 p_m} + 1 - p_m\right). \tag{127}$$

We can see that when $\sigma_1$ and $\sigma_1 p_m$ are relatively large then $p(X) \approx 1 - p_m$. That is, the ratio of tokens moving before absorption is $p(X) \approx 1 - p_m$, or in other words the probability of tokens being masked without ever moving is $p_m$ which is precisely the probability of a token being masked in our transition-rate matrix $\boldsymbol{Q}^{tok}$. Thus the last row of our matrix directly controls the percentage of corrected tokens in the reverse process for large enough diffusion times.

### A.3.5  TIME EVOLVING ROULETTE

Here, we generalize the results of the previous subsections for the case that $p_m$ varies with respect to time. We start by defining

$$\boldsymbol{Q}^{tok}_{eroulette}(t) = [(1 - p_m(t))\sigma(t)]'_t \boldsymbol{Q}^{tok}_{uniform} + [p_m(t)\sigma(t)]'_t \boldsymbol{Q}^{tok}_{absorb}. \tag{128}$$

First, we point out that since the size of $\boldsymbol{Q}_{eroulette}$ is $(n \times n)$ and that of $\boldsymbol{Q}_{uniform}$ is $(V \times V)$, where $V = n-1$, in order for them to have the same size, we add a row of zeros and a column of zeros to the bottom and right of $\boldsymbol{Q}_{uniform}$ respectively. Clearly, by definition, these are transition rate matrices, and therefore so is $\boldsymbol{Q}^{tok}_{eroulette}(t)$. Indeed, the elements in each of its columns add to 0, and the only negative elements are in the diagonals. If we define, $p_m(t)$ such that $p_m(0) = 0$ and $p_m(1) = 1$, then the limiting distribution will be the one-hot encoding at the absorb state. Furthermore, we can compute the exponential matrix of $\boldsymbol{Q}^{tok}_{eroulette}(t)$. It is easy to prove that $(1 - p_m(t))\sigma(t)\boldsymbol{Q}^{tok}_{uniform}$ and $p_m(t)\sigma(t)\boldsymbol{Q}^{tok}_{absorb}(t)$ commute with each other, thus

$$e^{\boldsymbol{Q}^{tok}_{eroulette}(t)} = e^{(1-p_m(t))\sigma(t)\boldsymbol{Q}^{tok}_{uniform} + p_m(t)\sigma(t)\boldsymbol{Q}^{tok}_{absorb}}. \tag{129}$$

$$= e^{p_m(t)\sigma_t \boldsymbol{Q}^{tok}_{absorb}} e^{(1-p_m(t))\sigma_t \boldsymbol{Q}^{tok}_{uniform}}. \tag{130}$$

Writing $\alpha_t = p_m(t)\sigma_t$ and $\beta_t = (1 - p_m(t))\sigma_t$, shows that we can calculate each of these exponential matrices using the strategy below

$$e^{c(t)\boldsymbol{Q}^{tok}} = \boldsymbol{I} + \sum_{k=1}^{\infty} \frac{(\boldsymbol{Q}^{tok})^k c(t)^k}{k!} = \boldsymbol{I} + \sum_{k=1}^{\infty} \frac{(-1)^{k+1} c(t)^k \boldsymbol{Q}^{tok}}{k!} = \boldsymbol{I} + \boldsymbol{Q}^{tok}(1 - e^{-c(t)}). \tag{131}$$

Indeed,

$$e^{\alpha_t \boldsymbol{Q}^{tok}_{absorb}} = \boldsymbol{I} + \boldsymbol{Q}^{tok}_{absorb}(1 - e^{-\alpha_t}) \text{ and } e^{\beta_t \boldsymbol{Q}^{tok}} = \boldsymbol{I} + \boldsymbol{Q}^{tok}_{uniform}(1 - e^{-\beta_t}). \tag{132}$$

Multiplying them together, we get the following proposition:

**Proposition 7.** *If we denote with $\boldsymbol{Y}_t$ the matrix exponential of $\sigma_t \boldsymbol{Q}^{tok}_{eroulette}$, then $\boldsymbol{Y}_t(i \notin \{j, n\}, j \neq n) = e^{-\sigma_t p_m(t)} \frac{1}{n-1}(1 - e^{-(1-p_m(t))\sigma_t})$, $\boldsymbol{Y}_t(i \neq n, i \neq n) = e^{-\sigma_t p_m(t)}(1 - \frac{n-2}{n-1}(1 - e^{-(1-p_m(t))\sigma_t}))$, $\boldsymbol{Y}_t(n, j \neq n) = 1 - e^{-\sigma_t p_m(t)}$, $\boldsymbol{Y}_t(i \neq n, n) = 0$, and $\boldsymbol{Y}_t(n, n) = 1$.*

One can see that this is almost identical to Proposition 5, with the only difference being that $p_m$ varies with $t$. Indeed, if we fix $p_m$ so that it is constant with respect to time, Equation (128) becomes

$$\boldsymbol{Q}^{tok}_{eroulette}(t) = p_m \boldsymbol{Q}^{tok}_{absorb}(t) + (1 - p_m)\boldsymbol{Q}^{tok}_{uniform}(t) = \boldsymbol{Q}^{tok}_{roulette}(t), \tag{133}$$

meaning that $\boldsymbol{Q}^{tok}_{eroulette}(t)$, coincides with $\boldsymbol{Q}^{tok}_{roulette}(t)$ roulette in this case. This highlights that the roulette diffusion is an interpolation between the roulette and uniform diffusion.

One possible choice of $p_m(t)$ is $t^{\frac{1}{at}}$ for a positive constant $a$.

## B EXPERIMENTAL DETAILS

### B.1 ALGORITHMS

---

**Algorithm 1** Cross Entropy Training Algorithm

---

**Require:** Network $f_\theta$, (total) noise schedule $\sigma_t$, data distribution $p_{\text{data}}$, token transition matrix $\boldsymbol{Q}^{tok}$, and time $t \in [0, 1]$.
  **Sample** $\boldsymbol{x}_0 \sim p_0$, $t \sim \mathcal{U}([0, 1])$.
  **Construct** $\boldsymbol{x}_t$ from $\boldsymbol{x}_0$. In particular, $\boldsymbol{x}_t^i \sim p_{t|0}(\cdot|\boldsymbol{x}_0^i) = \exp(\sigma_t \boldsymbol{Q}^{tok})_{[:,\boldsymbol{x}_0^i]}$.
  **Compute** $L_{ll} = -\sum_{i=1}^L \log f_\theta^i(\boldsymbol{x}_t, t)[\boldsymbol{x}_0^i]$.
  **Backpropagate** $\nabla_\theta L_{ll}$.
  **Run** optimizer.

---

### B.2 NETWORK ARCHITECTURE AND HYPER-PARAMETERS

This core model is grounded in the diffusion transformer architecture introduced by Peebles & Xie (2023), which integrates time conditioning into the conventional encoder-only transformer framework as established by Vaswani et al. (2017); Devlin et al. (2019). However, it includes minor adjustments, such as the use of rotary positional encoding (Su et al., 2024). The model contains approximately 5% more parameters than a standard transformer (utilized in the case of GPT-2), attributed to the incorporation of time conditioning. Additionally, the same tokenizers and data splits as in prior work are employed to avoid introducing artifacts.

The network is configured with 12 transformer blocks, each featuring 12 attention heads and a hidden size of 768, aligning with the "small" variant of GPT-2. It includes conditioning dimensions set at 128 to facilitate the diffusion process by encoding time-dependent features. Notably, the architecture excludes masking, typical of generative models that generate all tokens simultaneously rather than sequentially. It uses standard scaled dot-product attention mechanisms and incorporates a dropout rate of 0.1 to mitigate overfitting.

In terms of hyperparameters, the model was trained on a single H100 when the sequence length is set at 128, while in the case of sequence lengths of 1024 the model is trained using 8×H100 with a vocabulary size of 50,257 tokens. Training involves a batch size of 512. The training regime is designed for a total of 400,000 iterations.

Training utilizes the OpenWebText dataset, while evaluation is conducted on WikiText-103, with data managed locally to speed up access times. The noise schedule for the diffusion process is log-linear (uniform, absorb), and roulette log-linear (roulette) controlling the variance of noise added incrementally. In both cases we set $\epsilon = 0.001$ as in (Lou et al., 2024). Sampling for evaluation during training employs an Euler predictor over 128 (and 1024 when $L = 1024$) steps, with noise removal enabled.

For optimization, the model uses the AdamW optimizer with a learning rate of 0.0003, beta parameters of 0.9 and 0.999, and epsilon set to 1e-8. It features no weight decay, focusing on adapting learning without additional regularization. The optimizer includes a warm-up phase of 2,500 steps to stabilize learning dynamics, and employs gradient clipping at a threshold of 1 to prevent gradients from exploding during training. The log-linear noise schedule was used in the absorb and uniform case, while the roulette log-linear one was used in the roulette case.

### B.3 EFFICIENT IMPLEMENTATION IN PRACTICE

#### B.3.1 EFFICIENT ESTIMATION OF THE SCORE IN PRACTICE

For our choice of sparse matrices $\boldsymbol{Q}_t$ which can only modify one position at each step, the loss function

$$\mathbb{E}_{t\sim U(0,1)}\mathbb{E}_{\boldsymbol{x}_0\sim p_0(\boldsymbol{x}_0)}\mathbb{E}_{\boldsymbol{x}_t\sim p_{t|0}(\cdot|\boldsymbol{x}_0)} \sum_{\boldsymbol{y}\neq\boldsymbol{x}_t} \boldsymbol{Q}_t(\boldsymbol{x}_t,\boldsymbol{y})\ell\left(\frac{p_{t|0}(\boldsymbol{y}|\boldsymbol{x}_0)}{p_{t|0}(\boldsymbol{x}_t|\boldsymbol{x}_0)}, s_\theta(\boldsymbol{x}_t,t)_{\boldsymbol{y}}\right), \tag{134}$$

becomes

$$\mathbb{E}_{t\sim U(0,1)}\mathbb{E}_{\boldsymbol{x}_0\sim p_0(\boldsymbol{x}_0)}\mathbb{E}_{\boldsymbol{x}_t\sim p_{t|0}(\cdot|\boldsymbol{x}_0)}\sum_{i=1}^{L}\sum_{\boldsymbol{y}^i\neq\boldsymbol{x}_t^i}\boldsymbol{Q}_t^{tok}(\boldsymbol{x}_t^i,\boldsymbol{y}^i)\ell\left(\frac{p_{t|0}(\boldsymbol{y}^i|\boldsymbol{x}_0^i)}{p_{t|0}(\boldsymbol{x}_t^i|\boldsymbol{x}_0^i)},s_\theta^i(\boldsymbol{x}_t,t)[\boldsymbol{y}^i]\right). \quad (135)$$

Now we focus on calculating the loss at position $i$, as the total loss across the sequence, will be simply the sum of such individual losses at each position. We notice that the term $\sum_{\boldsymbol{y}^i\neq\boldsymbol{x}_t^i}\boldsymbol{Q}_t^{tok}(\boldsymbol{x}_t^i,\boldsymbol{y}^i)s_\theta^i(\boldsymbol{x}_t,t)[\boldsymbol{y}^i]$ in the expression above is trivial to calculate. We simply add the exponentiated outputs of the neural network across the last dimension weighted by $\boldsymbol{Q}_t^{tok}(\boldsymbol{x}_t^i,\boldsymbol{y}^i)$, and then substract $s_\theta(\boldsymbol{x}_t,t)_{\boldsymbol{x}_t^i}^i\cdot\boldsymbol{Q}_t^{tok}(\boldsymbol{x}_t^i,\boldsymbol{x}_t^i)$. Since in all cases the terms of $\boldsymbol{Q}_t^{tok}$ are either mostly 0 or mostly the same, this can be done efficiently. The expression

$$\sum_{\boldsymbol{y}^i\neq\boldsymbol{x}_t^i}\boldsymbol{Q}_t^{tok}(\boldsymbol{x}_t^i,\boldsymbol{y}^i)\frac{p_{t|0}(\boldsymbol{y}^i|\boldsymbol{x}_0^i)}{p_{t|0}(\boldsymbol{x}_t^i|\boldsymbol{x}_0^i)}\log s_\theta^i(\boldsymbol{x}_t,t)[\boldsymbol{y}^i] \quad (136)$$

is more challenging to be calculated efficiently. We follow the approach of (Lou et al., 2024), when using SEDD for training. To reiterate, for each position $i$, if we use SEDD for training, we need to calculate the sum of the product $\frac{p_{t|0}(\boldsymbol{y}^i|\boldsymbol{x}_0^i)}{p_{t|0}(\boldsymbol{x}_t^i|\boldsymbol{x}_0^i)}\log s_\theta^i(\boldsymbol{x}_t,t)[\boldsymbol{y}^i]$ over $n$ ratios, where $n=V$ if $\boldsymbol{Q}^{tok}=\boldsymbol{Q}_{uniform}^{tok}$ and $n=V+1$ otherwise. Luckily, for the choices of $\boldsymbol{Q}^{tok}$ in this paper (uniform, absorb and roulette), such ratios have relatively simple form and they are almost all the same. For example, in the case of the **absorb diffusion**, there are two cases, either $\boldsymbol{x}_t^i$ has not moved ($\boldsymbol{x}_t^i=\boldsymbol{x}_0^i$) or it has been masked ($\boldsymbol{x}_t^i=n\neq\boldsymbol{x}_0^i$), which can be seen in Figure 4.

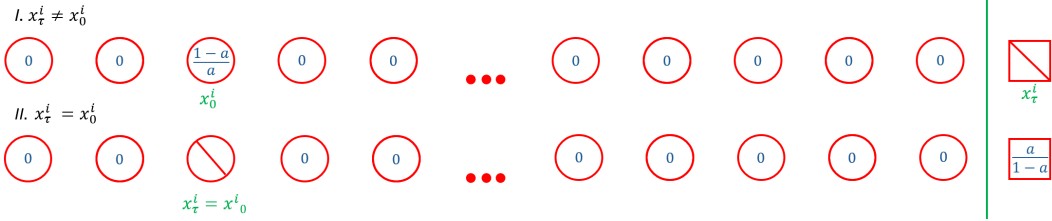

Figure 4: The conditional ratios at position $i$ over the vocabulary in the two cases. The square represents the absorb state. The value of $a$ is $1-e^{-\sigma_t}$.

To simplify notation, when iterating through different tokens in the vocabulary we write $\boldsymbol{x}_0^i=k$, $\boldsymbol{y}^i=j$ and $\log s_\theta^i(\boldsymbol{x}_t,t)[\boldsymbol{y}^i]=s_j$. In the first case, when $\boldsymbol{x}_t^i=n\neq\boldsymbol{x}_0^i$ we get

$$\sum_{\boldsymbol{y}^i\neq\boldsymbol{x}_t^i}\boldsymbol{Q}_t^{tok}(\boldsymbol{x}_t^i,\boldsymbol{y}^i)\frac{p_{t|0}(\boldsymbol{y}^i|\boldsymbol{x}_0^i)}{p_{t|0}(\boldsymbol{x}_t^i|\boldsymbol{x}_0^i)}\log s_\theta^i(\boldsymbol{x}_t,t)[\boldsymbol{y}^i]=\frac{1-a}{a}s_k. \quad (137)$$

On the other hand, in the second case ($\boldsymbol{x}_t^i=\boldsymbol{x}_0^i$) one has

$$\sum_{\boldsymbol{y}^i\neq\boldsymbol{x}_t^i}\boldsymbol{Q}_t^{tok}(\boldsymbol{x}_t^i,\boldsymbol{y}^i)\frac{p_{t|0}(\boldsymbol{y}^i|\boldsymbol{x}_0^i)}{p_{t|0}(\boldsymbol{x}_t^i|\boldsymbol{x}_0^i)}\log s_\theta^i(\boldsymbol{x}_t,t)[\boldsymbol{y}^i]= \quad (138)$$

$$\sum_{j\notin\{\boldsymbol{x}_t^i,n\}}\boldsymbol{Q}_t^{tok}(\boldsymbol{x}_t^i,j)0s_j+\boldsymbol{Q}_t^{tok}(\boldsymbol{x}_t^i,n)\frac{a}{1-a}s_n=0, \quad (139)$$

as $\boldsymbol{Q}_t^{tok}(\boldsymbol{x}_t^i,n)=0$. Thus when $\boldsymbol{x}_t^i=\boldsymbol{x}_0^i\neq n$, we can simply not calculate the loss. For the **uniform diffusion**, there are also two cases, namely $\boldsymbol{x}_t^i=\boldsymbol{x}_0^i$ and $\boldsymbol{x}_t^i\neq\boldsymbol{x}_0^i$ as illustrated in Figure 5.

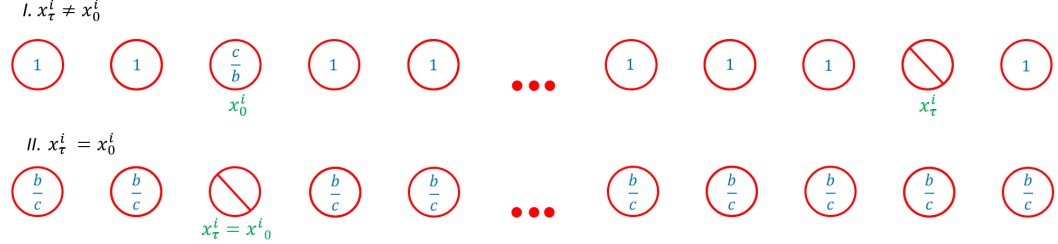

Figure 5: The conditional ratios at position $i$ over the vocabulary in the two cases. We define $b = \frac{1}{n=V}(1 - e^{-\sigma_t})$ and $c = 1 - (n-1)b$.

If $x_t^i \neq x_0^i$, then

$$\sum_{\boldsymbol{y}^i \neq \boldsymbol{x}_t^i} \boldsymbol{Q}_t^{tok}(\boldsymbol{x}_t^i, \boldsymbol{y}^i) \frac{p_{t|0}(\boldsymbol{y}^i|\boldsymbol{x}_0^i)}{p_{t|0}(\boldsymbol{x}_t^i|\boldsymbol{x}_0^i)} s_\theta^i(\boldsymbol{x}_t, t)[\boldsymbol{y}^i] = \sum_{j \notin \{\boldsymbol{x}_t^i, k\}} \boldsymbol{Q}_t^{tok}(\boldsymbol{x}_t^i, j)s_j + \boldsymbol{Q}_t^{tok}(\boldsymbol{x}_t^i, k)\frac{c}{b}s_k \quad (140)$$

$$= \sum_{j \neq \boldsymbol{x}_t^i} \boldsymbol{Q}_t^{tok}(\boldsymbol{x}_t^i, j)s_j + \boldsymbol{Q}_t^{tok}(\boldsymbol{x}_t^i, k)\left(\frac{c}{b} - 1\right)s_k = \sum_{j \neq \boldsymbol{x}_t^i} \frac{1}{n}s_j + \frac{1}{n}\left(\frac{c}{b} - 1\right)s_k. \quad (141)$$

Otherwise for $x_t^i = x_0^i$, we have

$$\sum_{\boldsymbol{y}^i \neq \boldsymbol{x}_t^i} \boldsymbol{Q}_t^{tok}(\boldsymbol{x}_t^i, \boldsymbol{y}^i) \frac{p_{t|0}(\boldsymbol{y}^i|\boldsymbol{x}_0^i)}{p_{t|0}(\boldsymbol{x}_t^i|\boldsymbol{x}_0^i)} s_\theta^i(\boldsymbol{x}_t, t)[\boldsymbol{y}^i] = \frac{b}{c}\sum_{j \notin \{\boldsymbol{x}_t^i\}} \boldsymbol{Q}_t^{tok}(\boldsymbol{x}_t^i, j)s_j = \frac{1}{n}\frac{b}{c}\sum_{j \neq \boldsymbol{x}_t^i} s_j. \quad (142)$$

Finally writing $S^i = \sum_j s_j$, we get $\frac{1}{n}S^i + \frac{1}{n}\left(\frac{c}{b} - 1\right)s_k - \frac{1}{n}s_{\boldsymbol{x}_t^i}$ in the first case, and $\frac{1}{n}\frac{b}{c}S^i - \frac{1}{n}\frac{b}{c}s_{\boldsymbol{x}_t^i}$ in the second one.

For the **roulette diffusion** we proceed similarly. Figure 6 shows the ratios in all three possible cases.

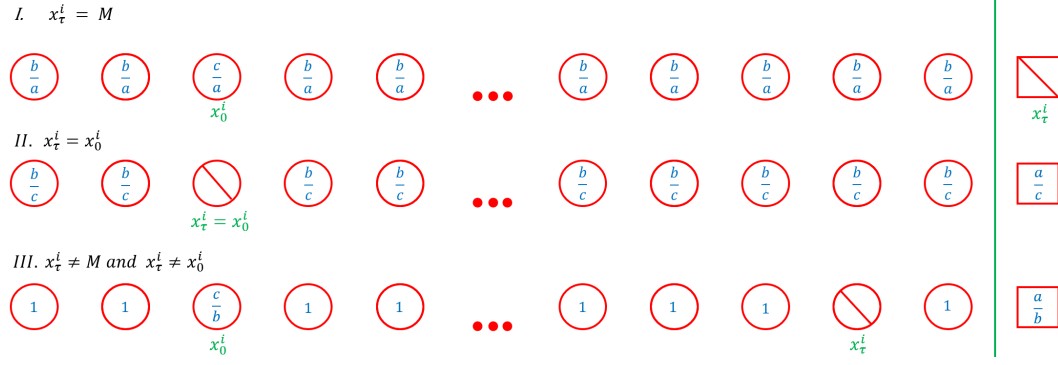

Figure 6: The conditional ratios at position $i$ over the vocabulary in the three cases. Since $\boldsymbol{x}_0^i$ is a token from the data it cannot be masked. The square represents the absorb state. We define $a = 1 - e^{-p_m \sigma_t}$, $b = e^{-p_m \sigma_t} \cdot \frac{1 - e^{-\sigma_t(1-p_m)}}{n-1}$ and $c = e^{-p_m \sigma_t}\left[\frac{1}{n-1} + \left(1 - \frac{1}{n-1}\right)e^{-\sigma_t(1-p_m)}\right]$.

In the first case, the sum at position $i$ is $p_m\left[\frac{b}{a}\left(S^i - s_k - s_{\boldsymbol{x}_t^i}\right) + \frac{c}{a}s_k\right]$. In the second it is $\frac{1-p_m}{V}\frac{b}{c}\left(S^i - s_k - s_n\right)$, while in the third case we have $\frac{1-p_m}{V}\left(S^i - s_k - s_n - s_{\boldsymbol{x}_t^i}\right) + \frac{1-p_m}{V}s_k\frac{c}{b}$.

If one chooses to incorporate the constant term $K(a)$, in the loss, then a similar approach is followed. However, considering Theorem 4 for evaluation, and the fact that a constant term does not affect the gradient during training, implementing this term is not necessary.

### B.3.2 SEDD SCALING

During training, some scores $\frac{p_{t|0}(\boldsymbol{y}^i|\boldsymbol{x}_0^i)}{p_{t|0}(\boldsymbol{x}_t^i|\boldsymbol{x}_0^i)}$, on expectation with respect to $\boldsymbol{x}_0^i$, can vary significantly for different $\boldsymbol{y}^i$. This makes the task of the network more complex and reduces performance. In order to

alleviate this issue, in the case of the **absorb diffusion**, (Lou et al., 2024) add $\log \sum_{\boldsymbol{x}_0^i} \frac{1}{n} \frac{p_{t|0}(\boldsymbol{y}^i|\boldsymbol{x}_0^i)}{p_{t|0}(\boldsymbol{x}_t^i|\boldsymbol{x}_0^i)}$ to the output of the network $\log s_\theta^i(\boldsymbol{x}_t, t)[\boldsymbol{y}^i]$. In this case, since we only need to calculate scores when $\boldsymbol{x}_t^i = n$, this sum equals

$$\sum_{\boldsymbol{x}_0^i} \frac{1}{n-1} \frac{p_{t|0}(\boldsymbol{y}^i|\boldsymbol{x}_0^i)}{p_{t|0}(\boldsymbol{x}_t^i|\boldsymbol{x}_0^i)} = \sum_{\boldsymbol{x}_0^i} \frac{1}{n-1} \frac{p_{t|0}(\boldsymbol{y}^i \neq n|\boldsymbol{x}_0^i)}{p_{t|0}(n|\boldsymbol{x}_0^i)} = 0 + \frac{1}{n-1} \frac{p_{t|0}(\boldsymbol{y}^i \neq n|\boldsymbol{y}^i \neq n)}{p_{t|0}(n|\boldsymbol{y}^i \neq n)} \quad (143)$$

$$= \frac{1}{n-1} \frac{e^{-\sigma_t}}{1 - e^{-\sigma_t}} = \frac{1}{n-1} \frac{1}{e^{\sigma_t} - 1}. \quad (144)$$

We notice that in effect, this changes the prediction of the score, from $\log s_\theta^i(\boldsymbol{x}_t, t)[\boldsymbol{y}^i]$ to $\log s_\theta^i(\boldsymbol{x}_t, t)[\boldsymbol{y}^i] - \log(n-1) - \log(e^{\sigma_t} - 1)$.

We extend this strategy for the uniform and the roulette diffusion. In the case of the **uniform diffusion**, we get

$$\sum_{\boldsymbol{x}_0^i} \frac{1}{V} \frac{p_{t|0}(\boldsymbol{y}^i|\boldsymbol{x}_0^i)}{p_{t|0}(\boldsymbol{x}_t^i|\boldsymbol{x}_0^i)} = \sum_{\boldsymbol{x}_0^i \notin \{\boldsymbol{y}^i, \boldsymbol{x}_t^i\}} \frac{1}{V} \frac{p_{t|0}(\boldsymbol{y}^i|\boldsymbol{x}_0^i)}{p_{t|0}(\boldsymbol{x}_t^i|\boldsymbol{x}_0^i)} + \frac{1}{V} \frac{p_{t|0}(\boldsymbol{y}^i|\boldsymbol{y}^i)}{p_{t|0}(\boldsymbol{x}_t^i|\boldsymbol{y}^i)} + \frac{1}{V} \frac{p_{t|0}(\boldsymbol{y}^i|\boldsymbol{x}_t^i)}{p_{t|0}(\boldsymbol{x}_t^i|\boldsymbol{x}_t^i)}. \quad (145)$$

Now,

$$\frac{p_{t|0}(\boldsymbol{y}^i|\boldsymbol{y}^i)}{p_{t|0}(\boldsymbol{x}_t^i|\boldsymbol{y}^i)} = \frac{1 - \frac{V-1}{V}(1 - e^{-\sigma_t})}{\frac{1}{V}(1 - e^{-\sigma_t})} = 1 + \frac{V}{e^{\sigma_t} - 1}, \quad (146)$$

and

$$\frac{1}{V} \frac{p_{t|0}(\boldsymbol{y}^i|\boldsymbol{x}_t^i)}{p_{t|0}(\boldsymbol{x}_t^i|\boldsymbol{x}_t^i)} = \frac{\frac{1}{V}(1 - e^{-\sigma_t})}{1 - \frac{V-1}{V}(1 - e^{-\sigma_t})} = 1 - \frac{V}{e^{\sigma_t} - 1 + V}. \quad (147)$$

Since $\frac{p_{t|0}(\boldsymbol{y}^i|\boldsymbol{x}_0^i \neq \boldsymbol{y}^i)}{p_{t|0}(\boldsymbol{x}_t^i|\boldsymbol{x}_0^i \neq \boldsymbol{x}_t^i)} = 1$, we conclude

$$\sum_{\boldsymbol{x}_0^i} \frac{1}{V} \frac{p_{t|0}(\boldsymbol{y}^i|\boldsymbol{x}_0^i)}{p_{t|0}(\boldsymbol{x}_t^i|\boldsymbol{x}_0^i)} = \frac{V-2}{V} + \frac{1}{V} + \frac{1}{e^{\sigma_t} - 1} + \frac{1}{V} - \frac{1}{e^{\sigma_t} - 1 + V} = 1 + \frac{1}{e^{\sigma_t} - 1} - \frac{1}{e^{\sigma_t} - 1 + V}.$$

We then modify the output of the network $\log s_\theta^i(\boldsymbol{x}_t, t)[\boldsymbol{y}^i]$ accordingly

$$\log s_\theta^i(\boldsymbol{x}_t, t)[\boldsymbol{y}^i] + \log \left( \sum_{\boldsymbol{x}_0^i} \frac{1}{V} \frac{p_{t|0}(\boldsymbol{y}^i|\boldsymbol{x}_0^i)}{p_{t|0}(\boldsymbol{x}_t^i|\boldsymbol{x}_0^i)} \right), \text{ where } n := V. \quad (148)$$

Similarly, we can calculate this scaling factor for the **roulette diffusion**. There are two cases, either $\boldsymbol{x}_t^i = n$ or $\boldsymbol{x}_t^i \neq n$. In the first case, with little modifications from before we can show that

$$\sum_{\boldsymbol{x}_0^i} \frac{1}{n-1} \frac{p_{t|0}(\boldsymbol{y}^i \neq n|\boldsymbol{x}_0^i)}{p_{t|0}(n|\boldsymbol{x}_0^i)} = \frac{1}{n-1} \frac{1 - (1 - e^{-\sigma_t p_m})}{1 - e^{-\sigma_t p_m}} = \frac{1}{n-1} \frac{1}{e^{\sigma_t p_m} - 1}. \quad (149)$$

In the second case,

$$\sum_{\boldsymbol{x}_0^i} \frac{1}{n-1} \frac{p_{t|0}(\boldsymbol{y}^i \neq n|\boldsymbol{x}_0^i)}{p_{t|0}(\boldsymbol{x}_t^i \neq n|\boldsymbol{x}_0^i)} = 1 + \frac{1}{e^{(1-p_m)\sigma_t} - 1} - \frac{1}{e^{(1-p_m)\sigma_t} - 1 + n - 1}. \quad (150)$$

We then modify the output of the network $\log s_\theta^i(\boldsymbol{x}_t, t)[\boldsymbol{y}^i]$ accordingly

$$\log s_\theta^i(\boldsymbol{x}_t, t)[\boldsymbol{y}^i] + \log \left( \sum_{\boldsymbol{x}_0^i} \frac{1}{n} \frac{p_{t|0}(\boldsymbol{y}^i|\boldsymbol{x}_0^i)}{p_{t|0}(\boldsymbol{x}_t^i|\boldsymbol{x}_0^i)} \right). \quad (151)$$

We take a moment to recall that if $\boldsymbol{y}$ and $\boldsymbol{x}_t$ differ only at one position $i$ then

$$\frac{p_t(\boldsymbol{y})}{p_t(\boldsymbol{x}_t)} = \sum_{\boldsymbol{x}_0^i} p^i(\boldsymbol{x}_0^i|\boldsymbol{x}_t) \frac{p_{t|0}(\boldsymbol{y}^i|\boldsymbol{x}_0^i)}{p_{t|0}(\boldsymbol{x}_t^i|\boldsymbol{x}_0^i)}. \quad (152)$$

Thus, above we are scaling the output by a naive estimation of this expectation where $p^i(\boldsymbol{x}_0^i|\boldsymbol{x}_t)$ is assumed to be $\frac{1}{n-1}$. In fact, the study of the scaling approach and the realization that the entity $\frac{1}{n-1}$ can be more properly estimated, is what originally motivated the derivation and usage of CEDD in this paper.

### B.3.3 SCORE ESTIMATION THROUGH CEDD IN PRACTICE

In this subsection, similarly to Section B.3.1, we show how to convert the learned probabilities into learned ratios in practice. Suppose that the perturbed token at position $i$ at time $t$ is $\boldsymbol{x}_t^i$. If $\boldsymbol{y}$ is a sequence that differs from $\boldsymbol{x}_t$ only at position $i$ then from before one can write:

$$\frac{p_t(\boldsymbol{y})}{p_t(\boldsymbol{x}_t)} = \sum_{\boldsymbol{x}_0^i} p^i(\boldsymbol{x}_0^i|\boldsymbol{x}_t)\frac{p_{t|0}(\boldsymbol{y}^i|\boldsymbol{x}_0^i)}{p_{t|0}(\boldsymbol{x}_t^i|\boldsymbol{x}_0^i)}. \tag{153}$$

We point out that since $\boldsymbol{x}_t$ and $\boldsymbol{y}$ only differ at position $i$, then $\boldsymbol{x}_t^i \neq \boldsymbol{y}^i$. Thus,

$$\frac{p_t(\boldsymbol{y})}{p_t(\boldsymbol{x}_t)} = \sum_{\boldsymbol{x}_0^i \notin \{\boldsymbol{x}_t^i,\boldsymbol{y}^i\}} p^i(\boldsymbol{x}_0^i|\boldsymbol{x}_t)\frac{p_{t|0}(\boldsymbol{y}^i|\boldsymbol{x}_0^i)}{p_{t|0}(\boldsymbol{x}_t^i|\boldsymbol{x}_0^i)} + p^i(\boldsymbol{y}^i|\boldsymbol{x}_t)\frac{p_{t|0}(\boldsymbol{y}^i|\boldsymbol{y}^i)}{p_{t|0}(\boldsymbol{x}_t^i|\boldsymbol{y}^i)} + p^i(\boldsymbol{x}_t^i|\boldsymbol{x}_t)\frac{p_{t|0}(\boldsymbol{y}^i|\boldsymbol{x}_t^i)}{p_{t|0}(\boldsymbol{x}_t^i|\boldsymbol{x}_t^i)}. \tag{154}$$

In the case of the **absorb diffusion**, from Section B.3.1, we can see that we do not need the ratio when $\boldsymbol{y}^i = n$, and furthermore, we only need the ratios when $\boldsymbol{x}_t^i = n$. Since also the data is not masked, then $\boldsymbol{x}_0^i \neq n$. Therefore, the first term (the sum) in Equation (154) is 0, since $\frac{p_{t|0}(\boldsymbol{y}^i \neq n|\boldsymbol{x}_0^i \notin \{n,\boldsymbol{y}^i\})}{p_{t|0}(n|\boldsymbol{x}_0^i \notin \{n,\boldsymbol{y}^i\})} = 0$. Hence

$$\frac{p_t(\boldsymbol{y})}{p_t(\boldsymbol{x}_t)} = p^i(\boldsymbol{y}^i|\boldsymbol{x}_t)\frac{p_{t|0}(\boldsymbol{y}^i \neq n|\boldsymbol{y}^i \neq n)}{p_{t|0}(n|\boldsymbol{y}^i \neq n)} + p^i(\boldsymbol{x}_t^i|\boldsymbol{x}_t)\frac{p_{t|0}(\boldsymbol{y}^i \neq n|n)}{p_{t|0}(n|n)}, \tag{155}$$

where $p_{t|0}(\boldsymbol{y}^i \neq n|n) = 0$ since a masked token cannot be unmasked in the forward process. Thus our approximation of the score is simply

$$s_\theta^i(\boldsymbol{x}_t,t)[\boldsymbol{y}^i] = f_\theta^i(\boldsymbol{x}_t,t)[\boldsymbol{y}^i]\frac{e^{-\sigma_t}}{1-e^{-\sigma_t}}. \tag{156}$$

To conclude $s_\theta(\boldsymbol{x}_t,t) = f_\theta(\boldsymbol{x}_t,t)\frac{1}{e^{\sigma_t}-1}$.

In the case of the **uniform diffusion**, we write $b = \frac{1}{n=V}(1-e^{-\sigma_t})$ and $c = 1-(n-1)b$ as in Section B.3.1, and Equation (154) becomes

$$\frac{p_t(\boldsymbol{y})}{p_t(\boldsymbol{x}_t)} = \sum_{\boldsymbol{x}_0^i \notin \{\boldsymbol{x}_t^i,\boldsymbol{y}^i\}} p^i(\boldsymbol{x}_0^i|\boldsymbol{x}_t) + p^i(\boldsymbol{y}^i|\boldsymbol{x}_t)\frac{c}{b} + p^i(\boldsymbol{x}_t^i|\boldsymbol{x}_t)\frac{b}{c}. \tag{157}$$

$$\frac{p_t(\boldsymbol{y})}{p_t(\boldsymbol{x}_t)} = \sum_{\boldsymbol{x}_0^i} p^i(\boldsymbol{x}_0^i|\boldsymbol{x}_t) + p^i(\boldsymbol{y}^i|\boldsymbol{x}_t)\left(\frac{c}{b}-1\right) + p^i(\boldsymbol{x}_t^i|\boldsymbol{x}_t)\left(\frac{b}{c}-1\right). \tag{158}$$

Therefore, our approximation of the score is simply

$$s_\theta^i(\boldsymbol{x}_t,t)[\boldsymbol{y}^i] = 1 + f_\theta^i(\boldsymbol{x}_t,t)[\boldsymbol{y}^i]\left(\frac{c}{b}-1\right) + f_\theta^i(\boldsymbol{x}_t,t)[\boldsymbol{x}_t^i]\left(\frac{b}{c}-1\right), \tag{159}$$

or

$$s_\theta^i(\boldsymbol{x}_t,t) = \boldsymbol{1} + f_\theta^i(\boldsymbol{x}_t,t)\left(\frac{c}{b}-1\right) + f_\theta^i(\boldsymbol{x}_t,t)[\boldsymbol{x}_t^i]\left(\frac{b}{c}-1\right)\boldsymbol{1}, \tag{160}$$

in vector form.

Finally, in the case of **roulette diffusion**, we can discern two possibilities, namely $\boldsymbol{x}_t^i = n$ and $\boldsymbol{x}_t^i \neq n$. In the first one, writing as in Section B.3.1, $a = 1 - e^{-p_m\sigma_t}$, $b = e^{-p_m\sigma_t} \cdot \frac{1-e^{-\sigma_t(1-p_m)}}{n-1}$ and $c = e^{-p_m\sigma_t}\left[\frac{1}{n-1} + \left(1-\frac{1}{n-1}\right)e^{-\sigma_t(1-p_m)}\right]$, Equation 154 becomes

$$\frac{p_t(\boldsymbol{y})}{p_t(\boldsymbol{x}_t)} = \sum_{\boldsymbol{x}_0^i \notin \{\boldsymbol{x}_t^i,\boldsymbol{y}^i\}} p^i(\boldsymbol{x}_0^i|\boldsymbol{x}_t)\frac{b}{a} + p^i(\boldsymbol{y}^i|\boldsymbol{x}_t)\frac{c}{b} = \frac{b}{a} + p^i(\boldsymbol{y}^i|\boldsymbol{x}_t)\left(\frac{c}{b}-\frac{b}{a}\right). \tag{161}$$

Hence, if the perturbed token $\boldsymbol{x}_t^i$ at position $i$ is masked, the predicted ratios over the vocabulary are:

$$s_\theta^i(\boldsymbol{x}_t,t) = \boldsymbol{1}\frac{b}{a} + f_\theta^i(\boldsymbol{x}_t,t)\left(\frac{c}{b}-\frac{b}{a}\right). \tag{162}$$

Very similarly, if the perturbed token $\boldsymbol{x}_t^i$ at position $i$ is not masked, the predicted ratios over the vocabulary are:

$$s_\theta^i(\boldsymbol{x}_t, t) = \boldsymbol{1} + f_\theta^i(\boldsymbol{x}_t, t)\left(\frac{c}{b} - 1\right) + f_\theta^i(\boldsymbol{x}_t, t)[\boldsymbol{x}_t^i]\left(\frac{b}{c} - 1\right)\boldsymbol{1}, \qquad (163)$$

This shows that we can calculate the ratios by scaling and translating the learned probabilities through CEDD.

**Modification of the reparametrization in (18): Re-scaling conditional ratios when $t \to 0$.**

In the case of the uniform diffusion dynamics, when $t \to 0$ then $\frac{b}{c} \to 0$ and $\frac{c}{b} \to \infty$. This does not cause any issues when sampling, however, it does negatively impact the perplexity bound. We explain informally the issue below. We fix position $i$ and assume we are at $\boldsymbol{x}_t^i$ and that originally we were at $\boldsymbol{x}_0^i$. When the time $t$ is close to 0, the model $f_\theta^i(\boldsymbol{x}_t, t)[\boldsymbol{y}^i]$ is typically very confident and can predict that some token $\boldsymbol{y}^i$ is $\boldsymbol{x}_0^i$, that is, $f_\theta^i(\boldsymbol{x}_t, t)[\boldsymbol{y}^i] \approx 1 - \epsilon$ and $f^i$ is $\approx \frac{\epsilon}{V-1}$ everywhere else. Furthermore, most of the time $\boldsymbol{x}_t^i = \boldsymbol{x}_0^i$. This means the score $s_\theta^i(\boldsymbol{x}_t, t)[\boldsymbol{y}^i] \approx \frac{c}{b}$ and is $\approx \frac{b}{c}$ everywhere else. If the prediction is correct then the loss becomes $\approx \frac{1}{V-1}(V-2)[-\frac{b}{c}\log(\frac{b}{c}) + \frac{b}{c} + \frac{b}{c}(\log(\frac{b}{c}) - 1)] + [-\frac{c}{b}\log(\frac{c}{b}) + \frac{c}{b} + \frac{c}{b}(\log(\frac{c}{b}) - 1)] \approx 0$. If it misses however, then we have $\approx \frac{1}{V-1}(V-3)[-\frac{b}{c}\log(\frac{b}{c}) + \frac{b}{c} + \frac{b}{c}(\log(\frac{b}{c}) - 1)] + [-\frac{b}{c}\log(\frac{c}{b}) + \frac{c}{b} + \frac{b}{c}(\log(\frac{c}{b}) - 1)] + [-\frac{c}{b}\log(\frac{b}{c}) + \frac{b}{c} + \frac{c}{b}(\log(\frac{c}{b}) - 1)] \approx 2\frac{c}{b}\log\frac{c}{b}$. Considering the large magnitude of $\frac{c}{b}$ when $t \approx 0$, this loss is extremely punitive. SEDD does not suffer from this issue, as the conditional ratios $\frac{c}{b}$ are implicitly learned by the network, which due to its limited flexibility likely acts as a regularizer, not allowing the values of the ratios to rise steeply as $t \to 0$. Motivated by this, in the uniform case, we rescale $\sigma_t < 0.0015$, by setting $\sigma_t = 0.0015$. This significantly reduces the magnitude of $\frac{c}{b}$ as $t \to 0$. We highlight that only the $\sigma_t$ that is used to calculate $\frac{c}{b}$ and $\frac{b}{c}$ are scaled while the $\sigma_t$ that is fed to the model is not touched. Naturally, for the sake of rigor, this is also the model we employ to generate samples. Finally, we note that this problem also appears in the case of the roulette diffusion dynamics, as when time is close to 0 most tokens are unmasked. The same strategy is applied, as before, only to ratios $\frac{c}{b}$ and $\frac{b}{c}$, by rescaling $\sigma_t$ when $\sigma_t < 0.5$ as follows: $\sigma_t^{scaled} = \log(1.1\sigma_t + 1.1)$.

### B.3.4 Analytic Sampling in Practice

In (Lou et al., 2024) an alternative sampling scheme (Equation (18)) is provided. This method is called the analytic method and it performs better than Euler sampling, in particular when the number of sampling steps is small:

$$p_{t-\epsilon|t}(\boldsymbol{x}_{t-\epsilon}^i|\boldsymbol{x}_t^i) = \left(e^{\sigma_t^{\Delta t}\boldsymbol{Q}^{tok}}(\boldsymbol{x}_t^i, \boldsymbol{x}_{t-\epsilon}^i)\right)\sum_{\boldsymbol{y}^i=1}^n\left(e^{-\sigma_t^{\Delta t}\boldsymbol{Q}^{tok}}(\boldsymbol{x}_{t-\epsilon}^i, \boldsymbol{y}^i)\right)s_\theta^i(\boldsymbol{x}_t, t)[\boldsymbol{y}^i], \qquad (164)$$

where $\sigma_t^{\Delta t} = \sigma_t - \sigma_{t-\epsilon}$.

Since we have an analytic expression of the matrix exponential $e^{\sigma_t^{\Delta t}\boldsymbol{Q}^{tok}}$, we can easily derive the expression of $e^{-\sigma_t^{\Delta t}\boldsymbol{Q}^{tok}}$, by simply substituting $\sigma_t^{\Delta t}$ with $-\sigma_t^{\Delta t}$ in each entry. One can efficiently calculate the sum above by following the strategy of (Lou et al., 2024) as below.

In the case of the **absorb diffusion**, $e^{-\sigma_t^{\Delta t}\boldsymbol{Q}^{tok}}(j \neq n, j \neq n) = 1 - \bar{a}$, $e^{-\sigma_t^{\Delta t}\boldsymbol{Q}^{tok}}(n, n) = 1$, $e^{-\sigma_t^{\Delta t}\boldsymbol{Q}^{tok}}(n, j \neq n) = \bar{a}$, with all other entries being 0, where $\bar{a} = 1 - e^{\sigma_t^{\Delta t}}$. As such if $\boldsymbol{x}_{t-\epsilon} \neq n$

$$\sum_{\boldsymbol{y}^i=1}^n\left(e^{-\sigma_t^{\Delta t}\boldsymbol{Q}^{tok}}(\boldsymbol{x}_{t-\epsilon}^i, \boldsymbol{y}^i)\right)s_\theta^i(\boldsymbol{x}_t, t)[\boldsymbol{y}^i] = (1 - \bar{a})s_\theta^i(\boldsymbol{x}_t, t)[\boldsymbol{x}_{t-\epsilon}^i], \qquad (165)$$

and if $\boldsymbol{x}_{t-\epsilon} = n$

$$\sum_{\boldsymbol{y}^i=1}^n\left(e^{-\sigma_t^{\Delta t}\boldsymbol{Q}^{tok}}(\boldsymbol{x}_{t-\epsilon}^i, \boldsymbol{y}^i)\right)s_\theta^i(\boldsymbol{x}_t, t)[\boldsymbol{y}^i] = \bar{a}\sum_{\boldsymbol{y}^i=1}^{n-1}s_\theta^i(\boldsymbol{x}_t, t)[\boldsymbol{y}^i] + s_\theta^i(\boldsymbol{x}_t, t)[n] = \qquad (166)$$

$$= \bar{a} \sum_{\boldsymbol{y}^i=1}^{n} s_\theta^i(\boldsymbol{x}_t, t)[\boldsymbol{y}^i] + (1-\bar{a})s_\theta^i(\boldsymbol{x}_t, t)[n] = \bar{a}S_\theta^i(\boldsymbol{x}_t, t) + (1-\bar{a})s_\theta^i(\boldsymbol{x}_t, t)[n], \tag{167}$$

where $S_\theta^i(\boldsymbol{x}_t, t) = \sum_{\boldsymbol{y}^i=1}^{n} s_\theta^i(\boldsymbol{x}_t, t)[\boldsymbol{y}^i]$.

In the case of the **uniform diffusion** $e^{-\sigma_t^{\Delta t} \boldsymbol{Q}^{tok}}(j,j) = \bar{c}$ while the rest of the entries are $\bar{b}$, where $\bar{b} = \frac{1}{n=V}(1 - e^{\sigma_t^{\Delta t}})$ and $\bar{c} = 1 - (n-1)\bar{b}$. Therefore

$$\sum_{\boldsymbol{y}^i=1}^{n} \left(e^{-\sigma_t^{\Delta t}\boldsymbol{Q}^{tok}}(\boldsymbol{x}_{t-\epsilon}^i, \boldsymbol{y}^i)\right) s_\theta^i(\boldsymbol{x}_t, t)[\boldsymbol{y}^i] = (\bar{c}-\bar{b})s_\theta^i(\boldsymbol{x}_t, t)[\boldsymbol{x}_{t-\epsilon}^i] + \bar{b}S_\theta^i(\boldsymbol{x}_t, t) = \tag{168}$$

$$= \frac{s_\theta^i(\boldsymbol{x}_t, t)[\boldsymbol{x}_{t-\epsilon}^i]}{e^{-\sigma_t^{\Delta t}}} + \frac{e^{-\sigma_t^{\Delta t}} - 1}{ne^{-\sigma_t^{\Delta t}}} S_\theta^i(\boldsymbol{x}_t, t). \tag{169}$$

Finally, in the case of the **roulette diffusion**, we have $e^{-\sigma_t^{\Delta t}\boldsymbol{Q}^{tok}}(j \neq n, j \neq n) = \bar{c}$, $e^{-\sigma_t^{\Delta t}\boldsymbol{Q}^{tok}}(j = n, j \neq n) = \bar{a}$, $e^{-\sigma_t^{\Delta t}\boldsymbol{Q}^{tok}}(j = n, j = n) = 1$, $e^{-\sigma_t^{\Delta t}\boldsymbol{Q}^{tok}}(j \neq n, j = n) = 0$ and the rest of the entries are $\bar{b}$, where $\bar{a} = 1 - e^{p_m \sigma_t^{\Delta t}}$, $\bar{b} = e^{p_m \sigma_t^{\Delta t}} \cdot \frac{1-e^{\sigma_t^{\Delta t}}}{n-1}$ and $\bar{c} = e^{p_m \sigma_t^{\Delta t}}\left[\frac{1}{n-1} + \left(1 - \frac{1}{n-1}\right)e^{\sigma_t^{\Delta t}(1-p_m)}\right]$. As in the absorb case, there are two cases, the first one being $\boldsymbol{x}_{t-\epsilon} \neq n$:

$$\sum_{\boldsymbol{y}^i=1}^{n} \left(e^{-\sigma_t^{\Delta t}\boldsymbol{Q}^{tok}}(\boldsymbol{x}_{t-\epsilon}^i, \boldsymbol{y}^i)\right) s_\theta^i(\boldsymbol{x}_t, t)[\boldsymbol{y}^i] = (\bar{c}-\bar{b})s_\theta^i(\boldsymbol{x}_t, t)[\boldsymbol{x}_{t-\epsilon}^i] + \bar{b}S_\theta^i(\boldsymbol{x}_t, t) - \bar{b}s_\theta^i(\boldsymbol{x}_t, t)[n], \tag{170}$$

while for $\boldsymbol{x}_{t-\epsilon} = n$, one derives:

$$\sum_{\boldsymbol{y}^i=1}^{n} \left(e^{-\sigma_t^{\Delta t}\boldsymbol{Q}^{tok}}(\boldsymbol{x}_{t-\epsilon}^i, \boldsymbol{y}^i)\right) s_\theta^i(\boldsymbol{x}_t, t)[\boldsymbol{y}^i] = \bar{a}S_\theta^i(\boldsymbol{x}_t, t) + (1-\bar{a})s_\theta^i(\boldsymbol{x}_t, t)[n] = \tag{171}$$

$$(\bar{c}-\bar{b})s_\theta^i(\boldsymbol{x}_t, t)[n] + \bar{b}S_\theta^i(\boldsymbol{x}_t, t) - \bar{b}s_\theta^i(\boldsymbol{x}_t, t)[n] + (\bar{a}-\bar{b})S_\theta^i(\boldsymbol{x}_t, t) + s_\theta^i(\boldsymbol{x}_t, t)[n](-\bar{a} + 2\bar{b} - \bar{c} + 1). \tag{172}$$

It is important to remark that in the case of the absorb and roulette diffusion, $s_\theta^i(\boldsymbol{x}_t, t)[n]$ plays a role in the quantities above only when $\boldsymbol{x}_t^i = n$. In the previous section, it is mentioned that $s_\theta^i(\boldsymbol{x}_t, t)[n]$ is not learned, but in this case when $\boldsymbol{x}_t^i = n$ is needed. However, this particular case is not problematic as $s_\theta^i(\boldsymbol{x}_t, t)[n] = s_\theta^i(\boldsymbol{x}_t, t)[\boldsymbol{x}_t^i] = 1$, thus we set $s_\theta^i(\boldsymbol{x}_t, t)[n]$ to 1 manually.

### B.4 Procedure for generating the plots in Figure 4

We sample a batch of 16 test points, and we perturb each datapoint with respect to a different $t$. Then, for example in the absorb case, we calculate the expressions

$$\sum_{\boldsymbol{y} \neq \boldsymbol{x}_t} \boldsymbol{Q}_t(\boldsymbol{x}_t, \boldsymbol{y})\ell\left(\frac{p_{t|0}(\boldsymbol{y}|\boldsymbol{x}_0)}{p_{t|0}(\boldsymbol{x}_t|\boldsymbol{x}_0)}, s_\theta(\boldsymbol{x}_t, t)_{\boldsymbol{y}}\right), \tag{173}$$

and

$$\sum_{\boldsymbol{y} \neq \boldsymbol{x}_t} \boldsymbol{Q}_t(\boldsymbol{x}_t, \boldsymbol{y})\bar{\ell}\left(\frac{p_{t|0}(\boldsymbol{y}|\boldsymbol{x}_0)}{p_{t|0}(\boldsymbol{x}_t|\boldsymbol{x}_0)}, s_\theta(\boldsymbol{x}_t, t)_{\boldsymbol{y}}\right) - 1 + \epsilon, \tag{174}$$

respectively for $J_1$ and $J_2$, where $\epsilon = 10^{-4}$. In each case, this returns a single point, therefore we repeat this procedure $64N$ times, where $N$ is the number of testing points. This ensures that in all cases ($J_1$ and $J_2$), we are estimating the loss in the same number of perturbed points. However, when $N$ is small, we keep testing $J_1$ and $J_2$ (extending their plots) for more points in order to provide to the reader more information about their (limiting) behaviour.

Table 5: Results comparing the roulette transition-matrices for different $p_m$ in terms of generative perplexity.

| | $p_m = 0.95$ | $p_m = 0.65$ | $p_m = 0.35$ | $p_m = 0.05$ |
|---|---|---|---|---|
| Roulette | 72.31 | 85.72 | 124.19 | 284.55 |

## B.5 Per sequence estimation of $J_1$

Here we explain a different method of calculating $J_1$. When applying this method one first samples a point (sequence) from the test set, and perturbs it for (say) 1024 time values $t$. Then the 1024 perturbed points are separated into (e.g.) 64 batches of size 16, and Expression 173 is computed for each batch which returns 64 values. This process is then repeated for the next test point (sequence). If the test set has N points then the process produces 64N values. These values can be averaged, divided by $L$ and exponentiated.

## B.6 Additional results

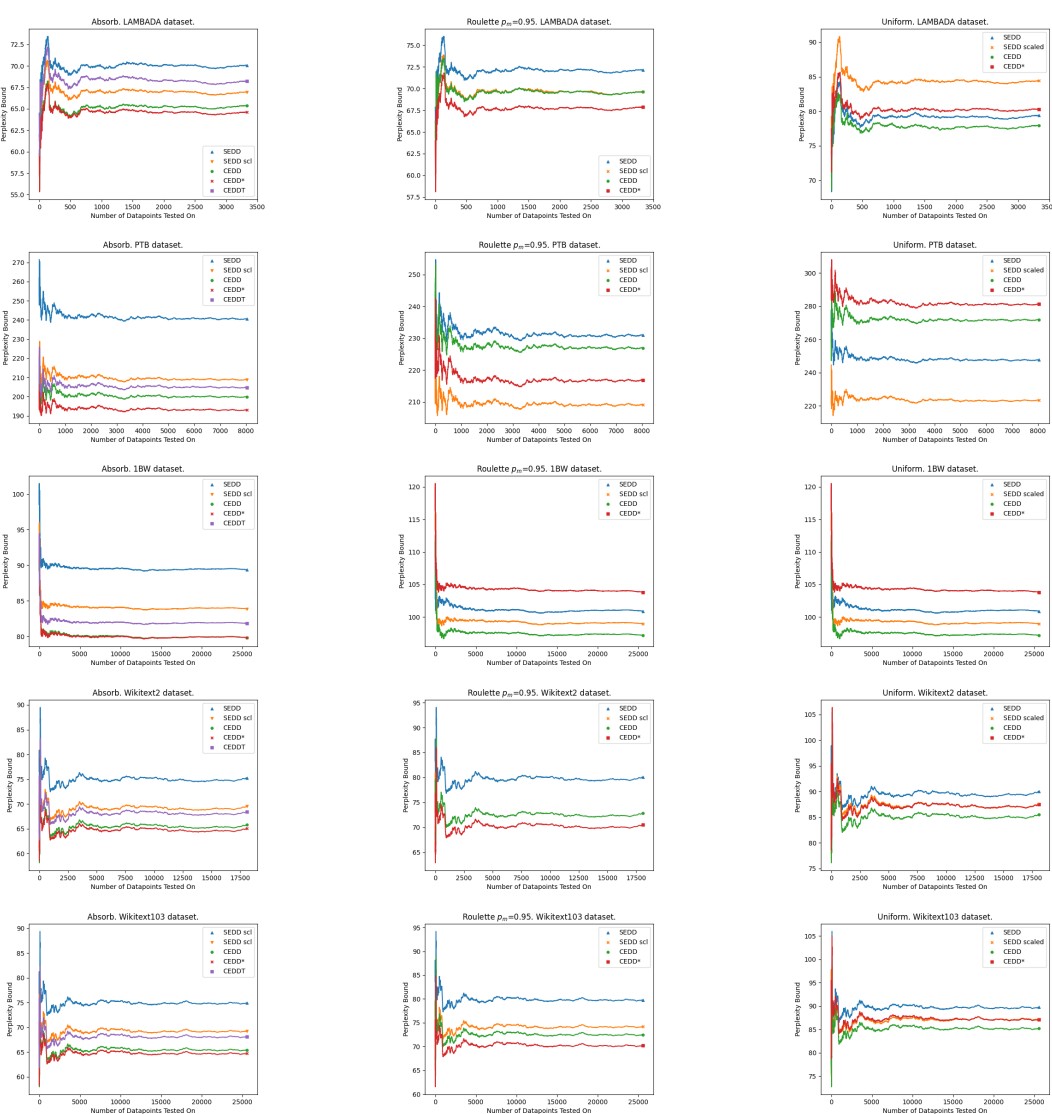

Figure 7: The comparison of the testing history of various methods on various datasets. Each row represents a different dataset, and each column a different type of diffusion transition-rate matrix. The way we calculated $J_1$ to produce the plots is explained in Appendix B.5. The curves presented are the exponentiated cummulative version of the array returned by the described process. Since we do not perturb the test set, this method enables a clearer visual comparison.

Table 6: Results comparing SEDD, SEDD scaled (SEDDs), CEDD and CEDD* using $J_2$. Lower is better.

| Model (L=128) | LAMBADA | WikiText2 | PTB | WikiText103 | 1BW |
|---|---|---|---|---|---|
| SEDD Absorb | 69.33 | 74.58 | 238.35 | 74.27 | 88.35 |
| SEDDs Absorb | 66.83 | 68.93 | 207.76 | 68.49 | 83.26 |
| CEDD Absorb | 64.87 | 65.07 | 198.00 | 64.97 | **78.93** |
| CEDD* Absorb | **64.11** | **64.54** | **191.38** | **64.30** | 79.17 |
| SEDD Roulette | 71.37 | 79.21 | 227.98 | 78.67 | 92.21 |
| SEDDs Roulette | 68.25 | 73.43 | 206.34 | 72.77 | 86.99 |
| CEDD Roulette | 68.92 | 72.16 | 224.18 | 71.52 | **85.37** |
| CEDD* Roulette | **67.27** | **69.61** | **213.90** | **69.45** | 85.64 |
| SEDD Uniform | 80.09 | 91.37 | 249.50 | 90.63 | 101.82 |
| SEDDs Uniform | 80.29 | 88.48 | **226.03** | 87.73 | 99.73 |
| CEDD Uniform | **79.46** | **86.82** | 276.61 | **86.52** | **98.44** |
| CEDD* Uniform | 82.43 | 89.68 | 289.09 | 88.90 | 106.32 |

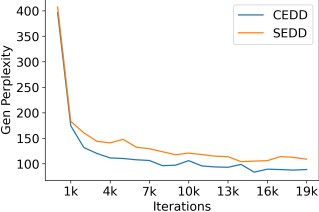

Figure 8: Comparison of CEDD* absorb L=1024 and SEDDs absorb 1024 in terms of generative perplexity (Euler, 128 steps), with respect to number of parameter updates.

Table 7: Results comparing SEDD (retrained), CEDD* trained for 20k parameter updates. For generation we use the analytic method with 1024 steps. Float 32 sampling.

| Model (Absorb) | GenPerp | LAMBADA | WikiText2 | PTB | WikiText103 | 1BW |
|---|---|---|---|---|---|---|
| SEDDs L=1024 | 55.27 | 67.92 | 64.91 | 173.38 | 64.13 | 107.64 |
| CEDD* L=1024 | 48.29 | 63.75 | 57.58 | 157.71 | 57.26 | 98.68 |

Table 8: Results comparing SEDD, SEDD scaled (SEDDs), CEDD and CEDD* in terms of generative perplexity.

| Model (L=128) | GenPerp Fl32-GPT2L | GenPerp Fl64-GPT2L | GenPerp Fl64-LLama8B |
|---|---|---|---|
| SEDD Absorb | 83.62 | 172.35 | 212.15 |
| SEDDs Absorb | 79.74 | 166.35 | 206.34 |
| CEDD Absorb | 74.19 | 148.21 | 185.90 |
| CEDD* Absorb | **72.13** | **143.86** | **183.74** |
| SEDD Roulette | 87.81 | 178.94 | 220.00 |
| SEDDs Roulette | 83.36 | 172.93 | 212.07 |
| CEDD Roulette | 76.71 | 167.67 | 208.84 |
| CEDD* Roulette | **72.31** | **158.56** | **197.79** |
| SEDD Uniform | 175.49 | 169.66 | 206.91 |
| SEDDs Uniform | 171.07 | 163.88 | 200.90 |
| CEDD Uniform | **168.35** | **161.84** | **200.09** |
| CEDD* Uniform | 179.30 | 175.42 | 213.99 |

Table 9: Results comparing SEDD, SEDD scaled (SEDDs), CEDD and CEDD* in terms of generative perplexity.

| Model (Absorb) | GenPerp Fl32-GPT2L | GenPerp Fl64-GPT2L | GenPerp Fl64-LLama8B |
|---|---|---|---|
| SEDDs L=128 | 79.74 | 166.35 | 206.34 |
| CEDD* L=128 | **72.13** | **143.86** | **183.74** |
| CEDDT L=128 | 74.07 | 154.04 | 195.41 |
| SEDDs L=1024 | 40.95 | 105.27 | 111.87 |
| CEDD* L=1024 | **40.93*** | **101.83** | **107.32** |
| CEDDT L=1024 | 42.18 | 108.88 | 115.60 |
| GPT-2 L=1024 | 41.02 | 41.02* | 50.25* |

Table 10: Correction accuracy percentages. 50k training iterations, batch size of 32 and $L = 128$.

| Model (L=128) | CEDD* Uniform | SEDD Uniform | CEDD* Roulette | SEDD Roulette |
|---|---|---|---|---|
| PAP | 90.7 | 89.9 | **90.8** | 87.9 |
| CAP | 91.2 | 90.3 | **91.3** | 88.6 |

Table 11: Percentage of mistakes corrected for additional models. 25k training iterations, batch size of 32 and $L = 128$.

| Model (L=128) | PAP | CAP | Model (L=128) | PAP | CAP |
|---|---|---|---|---|---|
| SEDD Roulette | 85.1 | 85.8 | SEDD Uniform | 86.9 | 87.5 |
| SEDDs Roulette | 86.5 | 87.2 | SEDDs Uniform | 86.4 | 87.1 |
| CEDD Roulette | 88.9 | 89.7 | CEDD Uniform | 88.5 | 89.2 |
| CEDD* Roulette | **89.7*** | **90.3*** | CEDD* Uniform | **89.5** | **89.5** |

Finally, we provide results (Table 12) when comparing CEDD* and Discrete Flow Matching that utilizes corrector sampling with the probability velocity $v_t^i(x^i, x_t) = \alpha_t \hat{u} - \beta_t \check{u}$, where $\check{u}(x^i, x_t) = \frac{\dot{k}_t}{k_t}(\delta_{x_t^i}(x^i) - \delta_m(x^i))$. We used $\alpha_t = 1 + \alpha t^a(1-t)^b$ for $\alpha = 1$ and $a = b = 0.5$. To compute the perplexity bound $e^B$ we utilized Equations (20) and (21) from Haxholli et al. (2024), and derived:

$$B = \frac{1}{L} \int_0^1 \sum_{x_t} p_{t|1}(x_t|x_1) \sum_{i=1}^{L} \left[ -\delta_{x_t^i \neq x_1^i} \frac{\dot{k}_t}{1 - k_t} \left( \log \alpha_t p_{1|t}^i(x_1^i|x_t; \theta) + 1 \right) + \right. \tag{175}$$

$$\left. \alpha_t \frac{\dot{k}_t}{1 - k_t}(1 - p_{1|t}^i(x_t^i|x_t; \theta)) + \beta_t \frac{\dot{k}_t}{k_t} \delta_{m \neq x_i} \right] dt. \tag{176}$$

We used $k_t = t$ and noticed that by increasing $\alpha$, generative perplexity improves but the quality of samples is reduced, to the point of generated sequences becoming simple repetitions of symbols and rare tokens at $\alpha = 10000$. This implies that $\alpha$ might modify the temperature of sampling. In addition, the measured perplexity bound quickly diverges to infinity, as $\alpha$ is increased.

Table 12: Generative perplexity was computed using GPT2 large, and sampling was performed at Float64 precision.

| Model (L=128) | GenPerp | LAMBADA | WT2 | PTB | WT103 | 1BW |
|---|---|---|---|---|---|---|
| SEDDs Absorb | 166.35 | 67.05 | 69.37 | 208.69 | 69.17 | 83.87 |
| CEDD* Absorb | 143.86 | **64.60** | **65.04** | **192.99** | **64.69** | **79.81** |
| Discrete flow $k_t = t$ | 145.48 | 71.90 | 71.20 | 221.15 | 70.84 | 82.63 |
| Discrete flow $k_t = t^2$ | 152.70 | 72.31 | 72.87 | 215.30 | 72.55 | 85.82 |
| Discrete Flow Correct | **143.38** | 114.40 | 113.05 | 352.32 | 112.87 | 130.41 |

## C GENERATED EXAMPLES

### C.1 GENERATED TEXT

Listing 1: Generated texts from Absorb diffusion trained using CEDD, L=128.

```
Any version of the bundle allows corruption. The campaign was delayed
    a year from testing ground, and because of other deficiencies.
    Perhaps this can be used to inform all the Xbox One games come
    with the same package. Whilst there will be no problems, there
    will probably be no corruption price-wise and we can expect the
    same with one Xbox One game package at the same time. My opinion
    is a solid one. Let's assume Japan as the entry point in the
    entire country, and PS3 probably supports only one of these. It is
     effectively competing (see video) against hardware arcade titles
    such as PS4 or XPS4 as well
================================================================
 his Achilles. I was ecstatic to be able to watch every one of them
    and keep them healthy tonight. You can look at the building
    blocks in college basketball, see that they play very hard and
    maybe feel superior to somebody. In my gut, however, you can
    argue that the one-year early good contracts come back and have a
     point. There has been a slight step-off with John Wall, which
    was a bit short on my mind. I now know Wall will be listed on the
     waiver, but I believe with Vic Beasley and Louren Maye this
    season, those two, with some really good shooters on the floor,
================================================================
 times when it is vital for Scotland's health and welfare The HRH
    Statement of Purpose and Development (SSTO) provides a framework
    for the government's creation of the Future of Europe as a single
     body independent of Scotland's national welfare system. The
    government's own intention is to actively develop its vision for
    the UK's welfare system based on its very nature and how the set
    of policies and key principals of the forum was established.

The Future of Europe Working Group (STO) Omissions of the UK Council
    on Europe's Audit Office (œsivishig
================================================================
 they go along with so many top players.

LAM: The lot of people in the parts they lose over are now than Jose
    Lopez, but if Affaar buys you deep and cheats Howard, Charo Colave
    , Bradley Donaldson, along with the rest of the opponents, we can
    see some of it.

LAM: With Santiago Ronaldo in South Florida. Vazal and Pedro Morales
    playing well, they don't help us.

LAIDHUL: And you can actually tell which sides are being switched. Of
    three moving sides can be quite accurate.

A pawn,
```

Listing 2: Generated texts from Uniform diffusion trained using CEDD, L=128.

```
 is to facilitate market conditions in jurisdictions. Also, not unlike
     that elsewhere where in terms of lack of in-house attorneys with
     no background qualified, or are looking to become in England and
     the whole country in the U.S., Arizona, New England, California,
     Connecticut, New Hampshire, New York, Massachusetts, Baltimore,
     Buffalo and Philadelphia, there are price measures set to rise to
     the model. And they come in handy as there are extraordinary
     levels of price in commodities, like alcohol and other drugs.
     Bankruptcycy approached, savings rate rigging now make even more
     dangerous. With broadest institutional licibility assets to sell
     and demand a
================================================================
 aircraft, under the Mohammed Boland project.[28][30][31] In 2010, the
     government collaborated with Oquantasch states hoping to build
     the first pilots and aircraft carriers,[33][34] announced that on
     16 October they decommissioned 25 acres of the Mozambique
     plantation leased by a Boeing 737, including several African
     pilots whom have been imparted on Brazilian soil and ended in. A
     military expedition - along with the arms embargo and tax evasion
     - has been delayed until federal Unabloprevalence halted part of
     the construction.[22]

Engineer units, and airport infrastructure [ edit ]

The
================================================================
 scene. Though a woman called, saying that whether it was being on the
     suspect's video is not the answer, Desiling says an undisclosed
     number were present at the mall, and that the police were ready.

In the end, two shoppers, Florimore, 27, sprinted outside the
     convenience store after struggling with downpour to get inside the
     shop.

When Florimore left the restroom at the store, Zimmerman entered the
     vendor's cell phone on Sunday and secured her keys Saturday,
     Desiling said. He helped he lock a key before being used to
     contract it. After the taped encounter with a current Illinois
     Bureau of Police
================================================================
, some of which were then scrutinized by the Bernanke Advisory
     Committee led by Michael Corm.

The U.K. Fed, eager.

Aboard all his most extreme concoctions careening the populace soon
     queued a shaky living net following weeks of the disastrous fiscal
     mess caused by the housing crash and the impending collapse in
     mortgage payments.

Can't anyone manage to usher it all the way. A savvy government could
     build "a serious living net - and that is much less than immediate
     relief." Instead of achieving a 50 percent reduction in mortgages
     , anyone at the Fed's table did it:
```

Listing 3: Generated texts from Roulette (0.95) diffusion trained using CEDD, L=128.

```
 deeper problems caused by America's growing deficit, and by our
    rapidly growing economy. The same was true today when the only
    way to undermine the nation's welfare was in a virtuous cycle of
    austerity, stimulus and lavish budgets. And the scorching reality
    , especially for millions, was that "every man is the child of
    his parents' mother"-not only abhorrent problem was the working
    age in Washington, which only saw 44% of the population working
    in the same household as network leaders and 79% of the workforce
     in their industry. We tapped these network leaders in polls on
    Election Day, causing favorable results
=======================================================================
 other federal government agencies had far greater control over water
    industry regulation. Today, drilling wasn't the equivalent of the
     Environmental Protection Agency championed since former
    president Ronald Reagan put Sen. Scott Desmond (CDE) on top of
    the Environmental Defense Agency, with a more funding area and
    rural programs under permit.

Like Moore, who spent so many a million dollars on his calls for the
    president to institute environmental protections on public lands
    and national parks, Trump himself had a nice shot on drilling,
    saying: "As passionate about the fight for property owners as we
    can be right here for Trump."

Canada XL - No
=======================================================================
Mistress: Next day next step, next step!" He's reply must not be
    translated. "What is - Guy's Your pronunciation? You still don't
    understand. Why am I saying that, don't mind: I've moved on to
    full set). Fitch word for move. Hey, that must have been long ago.
     What did you do going into our meeting? Everything changed, the
    rules changed... and you all admitted that Grand Auto Auto wasn't
    going to work, but you did. I need to know it. You've been in
    reference for the book, Notice."

Don't ignore him. He can't
=======================================================================
 like low income earners, raising the tax on below.

All Part D credits forgiven by at least twenty-one percent paying
    income tax, and no state interest payments to help the health
    insurance guarantee. If owners pay the credits, they spend money
    they owe on each payment, either. A fifth of incomes will pay the
    credits.

While it's perfectly up to the type of hardship, people can apply
    state or federal assistance to the credit program. It requires
    people to sell health insurance. The credits qualify and vary from
     individual person to enrollee.

Having decided to pay their state income taxes on the bill, a single
```

Listing 4: Generated texts from Absorb diffusion trained using CEDD*, L=128.

```
boat men: drift, and boom. Built, three 9th sailors found themselves
    losing an average of 12 inches of the underwater 50 feet in the
    delta base of S. M. Gielker Jr, U.S. captain (and pilot for the 11
    th Brigade), sailed deep into the canal on a watchdeck with his
    bow of thumb and lost at his anchor near shore.

Importantly, future "adillists should consider the alarming
    consequences that[t]here in the future the current rapprochement
    of the 9th would be had on the children of all areas of the coast
    ," re-ferenc
========================================================================
the world's private space program." (This mission, in James's words-
    rust is laced and revisited to the down-space fall and the
    commercial foundation-and eventually-software portable to what
    Lotus's proprietary operating system does now-and also new
    hardware-serves, itself, as a collective organization rather than
    a fragmented, coordinated copal: "NASA loves the space community
    at large.")

James retired as CEO on the company board in 2004. And longer,
    Planetary Resources's judoided business model is intimately linked
     to his own organizational record and by
========================================================================
 engaged in hacking American government and stealing cyber defense
     secrets from its political enemies.

"We know today that the celebrity and his partner in the leaks
    incident to WikiLeaks produce the true story about our government,
     content and the data shared by our intelligence services, and
    deserve accountability," said Mark Warner, Assistant Attorney
    General, at the key E.C.C conference on Wednesday in Washington, R
    .I. "In this case, the battle #WikiLeaks against WikiLeaks has
    touched both at a new level for the country, and finally has
    turning historic date.

"Our government has been without the power to hack
========================================================================
 non-committal after the third GOP debate.

Gia had already just commented on the upcoming status of her project
    on CNN, after Trump repeatedly asserted it would discuss politics
    for the first lady.

Trump said, "Look at this, on my part of Hillary Clinton, the mad
    country," Garcia continued.

"I took my position in a way President Trump didn't have before
    speaking :], he said it. I admitted some of those things, finally
    understanding that I stood for Oh Hillary as a businesswoman," she
     continued.

Garcia made the comment after training herself in Melania Trump's
    Daily
```

Listing 5: Generated texts from Uniform diffusion trained using CEDD*, L=128.

```
 the Registration of the Bank of Development despite the skills of
    Arasan Sance. unemployable novice is in high regard for Swatch Up
     Worker as there are large scale projects bylaws that have
    already been difficult in the sector.

Caught diving too deeper, Arasan Sance had a career at IIT built on
    Year Up in Science at the Wood,, Theatre during an instructor
    class at the university in Banance in 2017. The guide says that
    some jobs may have been finalised and it was understood well that,
     if efficacious, McCarley could decide to be invited to a perhaps
    larger as academic class.

========================================================================
 strength of subsequent fitness genes from mitochondria after
    maintenance of. This type of transition is called exalted
    mitochondrial, and although extended to long a feat is indeed
    amolecule strategy that requires stringent tests in selective
    binders.

In particular, one that abundant, modified mitochondria can enable
    recovery; first they produce mitochondria which is attached to the
     fuel cell, is rapidly moving essentially back to normal form,
    transitioning from the atrophy to a more muscular phenotype.
    Mitria such as L1L protein A (C1A and transporter R (Anaris-in
    proteinase), as NAD is used as an adaptation strategy, like many
========================================================================
 how things were in the later years. The truck was littered with a
    black pepper. It was overflowing with halal green and mo yellow
    corn pepperoni and a charcoal plate.

Pictured used to have a baseballs encased in the front driveway with
    the sports hooders. Guy's black ores tip, black bean bag, silver,
    cherry, and granny banger was ricocheted, and when the Appomeess
    the Chinese bullfly rolled me over to the Morrissey. O 44,
    everything fresh and natural were just different from those in
    earlier, inscriptions. This guy's big white hooded or blue jacket
========================================================================
 idea of increasing momentum, however, is to slow down a quarterback's
     strength. His is in the midst of a slow renaissance with the
    team's ensuing 17 games, but it is aging one who is getting the
    effectiveness of missing Greg Jennings and injuries. That this
    crucial area comes as the 49ers failed progressual Mac is forcing
     his fades down instead of sticking fully - unless he is getting
    better.

The Jets is approaching the top-10 in front but, as soon as they fill
    it in, an underdog-Dish seems to benefit from this slight fatigue
    in their upcoming game.

Negvious Matt Miller
```

Listing 6: Generated texts from Roulette (0.95) diffusion trained using CEDD*, L=128.

```
 such fraudulent statements and denials as, "the truth lies," and "
     white lies" could never be trusted. More than one among you have
     just known Williams about.

Many are the fictional experiences Williams shaped, in ways that
    embodied him in a generation. The Tielemin figure in James Joyce's
     white character largely was based on male gaze and information
    and, from Williams' perspective, reflecting to the lens on men
    from a different time was disappointing.

Williams's material images of how, in essence, we had occupied a not
    present world: beaten, destroyed, marginalized or
=============================================================
 an attempt to fully realize the potential for pedestrian safety where
      there is no dearticulate conduct for the traffic.

His own current plans, for which Lokxelli is the secretary of the
    planning drove changes with little progress through traditional
    detours as of late. "Not highways, but others are up quite a bit,"
    Akka forecasted "to make sure that one imp is not going to get
    torn. How to do that away from what it is becomes when much faster
     roads occur. There is all going to be of no serious consideration
    , at present, either in design or perspective."

=============================================================
 mad over and watching him laugh loudly enough to say so, and my point
      is that it's not just that I am not displeased with a politician
      or a CEO, that I think people looked in their shoes and paid
      well."

To see countries that see relevance to the show's multimillionaires,
    including so frustrating the small smile in the face of the world,
     Thomas Archifelius applied, but wanted to represent the image of
    man keeping ignorant out his protectionist mission.

ARTICLE CONTINUES BELOW

GEORGE STEPHANRAD: In a few weeks, Papacuzzi and so did Paul
=============================================================
 and made fun of the tactic. They acknowledged was just silly. Indeed,
      shortly after the books, a real Western painter and fellow
    broach type, said, to another wood miller, "he's straightened a
    nose. Skeptic got a hard chin and flies in the back." The winced
    to me is that you're a courtesan. You'll know what's up for you<|
    endoftext|>Get it done. Your friends will never see it

Parents of kids and old owners dogs yesterday said they are going to
    miss president-elect Donald Trump's look, 'Loving' for
```

Listing 7: Generated texts from Absorb diffusion trained using SEDD, L=128.

```
 likely to be the fact that Nephi calls God's beloved "The Synod Word
     ," a difference when it translates to an accident.

As viewers know, an Andesad man and his wife will never meet in Nephi'
    s time God begins to connect with others, Le Burkert said.

Source: Bonuses de jeunee included<|endoftext|>The concept of
    attunational fit occurs from the presence of perceptual
    experiences on the cortex of the brain, as exposed recently by
    research by senior Prof. Linda Perriene.[1] Since personal dynamic
     patterns don't exist within human subjects, cortical memory is
    often cognitive
========================================================================
 will look like a target.

97) J.O. Do you think of as a White House national coordinator as a
    Clinton political operative? Kitty|O. I sort of're still up-the-
    table the White House. P.O. That person who refuses to have the
    same meek with a client with concerns/lives just pretty boycotted
    by pulling off in the relationship. Glad let's get that out of the
     way tonight and point out his ability and his job promise.

98) Next to the President - David Grimsnek | In at least 100 years
    George W. Bush has
========================================================================
 last month invested in Ria Louise, New York through a wealth of $7
     billion this year.

Meanwhile, the committee is a part of Katzman & Girle Service
    Conference, the firm that licenses the annual facility. Frisco
    visits to CraftEarlier this year.

"We newg CraftyTags for philanthropic, recovery, trust, and use of our
     services, with immense support from federal government, Mossody
    said.

Under the partnership, Neoneys also is working on expansion as a
    partner among other not-for-profit resource providers, with an
    increasing amount of resources and core services covering
========================================================================
 at least one, but nothing qualified from any perspective they could
     be offering them.

"The response was from a very recent comment made in blue material by
    the public but it was decided ahead of time," Smith writes of the
    public's prurient. So we've found that a long and full response
    excludes the ones who didn't believe the policy would meet the
    government's standards for proof and hold."

Sure enough, their apologists could do what they want on hard science.
     But that being said, they'd like some insights into scientific
    ethics and
```

Listing 8: Generated texts from Uniform diffusion trained using SEDD, L=128.

```
 can generate a json of the library based on which one exists for the
    scenario instance.

<?php G require JetlinkViewer :: create ( ePub : 'Whu )' d. add(): c.
    app (). head () //Outputs from Sub_select from Google = str__ (
    http://www.google.com ); //Add it while that function starts down
    the path and results.Usingjson */ }

In order, Node handles repos(), several usable implementations in Mesh
    .js have been tested.

The JavaScript library actually has dire explanation for itls: it does
    not, but unlike
========================================================================
 using you or they use you taking people out to catch you or whatever.
       That was wonderful because of the ways I look. I had gone from
       anywhere from 20/22 here to 40 years old to 50/50. I also grew up
        feeling. 'True Blood Baby' has gave me a lot of body movements
       ... that's got me right there and I used to pop their first
       transformation. That show."

My sister really grew up as a nairety boy. Thrones in a while? No, it'
    s only sometimes, and since older people might even see us for 15
    days or 30 days per night, how long they
========================================================================
, but the ban on kidnapping remains the only difference.

Seyer has closed the case in a court letter seeking a response.

For more on the arrest video visit your Confused TMZ app page.

Updated Sept. 24, 2013. Tribute to Lawyer R Harrison.<|endoftext|>Gils
     Dahman is starting his season with the Red Bulls Crystal Palace.
    The midfielder rounded out his fifth season in the United States,
    missing three league meetings at U-a.

You can watch the interview with the media below. You can see an image
     for the camera below.

Gils Dahman has started
========================================================================
 Girl"). The earnestness of the ensemble is random, cheerful though
     frustrated with both the actions of themselves and the different
     perspectives on their journey. This classic masterpiece-writing
     in sequels and his imagined form of movies-is the tale of cinema
     that still has had aesthetics and is able to grow. But it also
     sees the fall of the film genre-though very serious yet complex-
     so we see a live show that can also find places lost by leading
     the way to work around intersectionality, where the lineage of
     Reed and Dormedy ("Iron Man" though) is applicable-and that she's
      starting
```

Listing 9: Generated texts from Roulette (0.95) diffusion trained using SEDD, L=128.

```
 its own territory. All the European Union as well as the U.S.
    government, including the British East Company, were issued coins
     to the Federal Reserve and a few days were in the open to
    support a foreign buyer. President Lee had adopted law rejecting
    donations from the country's tax code, preserving the spirit of
    the Constitution of the 1870 Act of self-rule in South America.
    In the caddition of the British banking system, the American Mint
     provided the first ammunition and smuggled guns to Europe.
    Nonetheless, a serious problem: the Union was secretly armed anti
    -communists with a formidable internal authority. The union
    government was
===========================================================================
 to blame?

It's full of wrongheaded sheen and bears expressions and carries a
    similar light ending. I'm not turned off on that Craig's Edge of
    Truth NYC guide if I don't see something wrong going on and do the
     right thing.

What's different about another story that can somehow tell the tale
    unfolds in Chicago so well with the ppl being approved by the
    General Service.

Like an Independent. Right? All right.

What Happens?

A Licensed Reader keeps all relationships private. They have no
    precedents. By all things
===========================================================================
 result it couldn't publish it.

No, my work is not the best way that all of us can be at home with the
     poll results. I acknowledge you believing it, and doing this poll
     is going to have to happen to make the difference.

But again, we are honestly happy with the results - let's see. We
    could not vote from all the experts - but that still could happen.
     What's your opinion on that debate?

The other most annoying - but the most upset is #MSNBC. In the Wall
    Street Journal, people called the polls slow Gore down, saying
    that even the ABC thought
===========================================================================
, 400 times the base capacity there has been priced to four. And now
    another station can do the same for capacity with two other lines
    to 1 per cent of the power meter factory circuit which also costs
    to $56 million for cancellation.

The next debate is on the schedule, but may not happen in late 2017
    without major delays.<|endoftext|>The French media reports that
    Chinese technology firm EtApple has today started claiming that it
     has all given back, in its software, a return to earlier this
    month's SmartEasy payment processing that will simply payment it
    for most Chinese users. And based on recent developments, the same
     company is sending
```

Listing 10: Generated text from Absorb diffusion trained using CEDD*, trained L=1024.

```
 two days later meeting with the Governmental Affairs Committee next
     Tuesday, replacing the committee's naps with an informal
     discussion.

In his Thursday hearing, Sessions said he had been "used" into
    criticizing the CIA.

"There has been some degree of transparency in looking at but that's
    not going to come cheap for himself or the American people, or
    really to undermine his independence, which I think, despite being
     in various positions, I've been very, very clear on the matter of
     that. I would be angry or very upset about that," he told
    senators at the meeting. "But I think as we go into Wednesday, and
     the United States Cabinet are finding ways to make very clear to
    citizens that they will respect that system."

Several high-level officials also called the appointment, saying it
    was "a bit of a quirk" since a revolving door of offices such as
    the CGA is already running rampant and has yet to hold any
    hearings.

"We've seen people make fairly much of their career life from the
    Justice Department out of this," said one GOP source. "And I don't
     think that choice is appropriate when that's a government entity
    that has given up a lot of accountability claims."

Speaking after noon another Republican member of the Senate Commerce
    Committee told. Todd Risch (R-Idaho) however, how he thought about
     Sessions, "and honestly, I didn't see a whole lot of discussion
    at one point suggesting a tone - a tone. There has been a lot of
    change since the removal of some five of these people ... I mean,
    I want people to find out that respect goes to the leadership of
    their department, and I'm not sure it's nice to move on down that
    precipice."<|endoftext|>When you try to put any pressure, you put
    more than the Pacers into you love watching. Before the Boston
    Celtics come to their games against Washington with.500 records on
     Tuesday night in Pittsburgh, you will likely watch a five-point
    game between Brad Stevens and Antetioun Gouden. Hollis have fourth
    -most competition in the game overall (49 points per game) but
    before games start their season of over 14.5 points per game has
    already passed.

That led Boston to an early 2-1 deficit.

No, not really. In addition to last season, however, the Celtics
    really struggled during the Wizards' first year of the NBA. For
    only the first 24 minutes, Washington had 18 minutes their D-man
    shooting guard who averaged at least four points, and as he
    chipped in another 11 points on at least 10 fourth quarter
    quarters Boston was outscored in the paint on five four-point
    shots, including two-point turnover margin.

If I included Stevens and Horford scoring 18 points in the only fourth
     quarter quarter from the paint, it would have probably been 2-1,
    and in the second-half-first quarter at half-time it would have
    been a 9-2 game thanks to the disastrous fourth-quarter
    performances. Yes, the Celtics have been unable to pull the ball
    away from us in the paint over the fourth quarter, but their
    success was probably in spite of the bad 2s, otherwise, our
    defense was weak.
```

Listing 11: Generated text from Absorb diffusion trained using SEDD, trained L=1024.

```
.

The principle of selling for a period of energy will make the most
    sense next year, as Tesla's record rapid sales may be sluggish but
     demand lives on.

Tesla Motors Corp's CTO Carl Conti said beginning a new conversation
    with Tesla about the future state of its workforce in 2014 may be
    a part of its strategy in which as the company races to reduce its
     price of luxury vehicle deliveries, the first units that get to
    market again in earnest are expected to begin to get "next level,"
     he said.

AP Photo

"I appreciate [the] ideas. These are not a fast break, but are ways to
     strengthen the margin at least knowing that if you get the right
    timeframe you make, its way ahead," Musk told reporters at The New
     York Times over the past month.

"None of these people have looked toward [s]ourcing the cost or the
    increase of the vehicle, unless you see it as a downward spiral,"
    he said.

Although Musk and other executives point to one to two auto drivers
    and a net purchaser of gasoline or other, the company's strategy
    is that a reduced price for vehicles will outstrip some of those
    buyers, Conti still said.

Conti said the price of Model S is projected to fall by at least a 33
    percent decline rate in 2015, on average. Conti expects Tesla
    expects 58.5 million to 59.5 million in 2015 and also for an
    average 15 percent decline to rise to February.

For the Model S, by contrast, Tesla's are expected to reach a write-
    down of about $500 million to 2,000,000 EVs in September and more
    than 700,000 currently, he said.

The first Model deliveries of the year is expected to nearly begin on
    March 22. That price is heavily influenced by dealer monitoring
    that includes a total inventory of 600,000, with 400,000 delivered
     next month, Conti said.

He noted Tesla expects its predicted upward trajectory to increase by
    2.4 percent by 2015 as production in Australia and China
    accelerate to 4.6 percent next year, respectively.

Musk also noted in The Wall Street Journal that while Tesla expects
    more than three consecutive in production next year, it will
    production a total of $8 million in U.S. roll-backs in the fall.

He said he has settled on a key question about the car: "If a customer
     needs it and is willing to pay the rates that we can keep things
    moving, then we ought to consider adding incentives for it."

"The price of Model S expectations for production is a driver'all,"
    Conti said. "When the profit is available, it means a car is not
    in its pocket. We are seeing this as a very good time after Jan.
    31."
```

Listing 12: Generated text from GPT2, trained L=1024.

```
power well past the NPC versus NPC meter for each unit type. Their use
    is varied, and thus allows for a considerable amount of
    flexibility with The Banner Saga, but whether you prefer your
    parent's roots to stay part of the community -- with allies
    everyone can fight and help each other -- remains to be seen. For
    that, you can always pick the steam community tab (not required,
    that's how the game is named -- no one likes spamming themselves
    for news now, I know).

However, you agree to terms of service that you agree to disable your
    favorite kind of multiplayer games (by default) to prevent
    unwanted conflicts: factions are only listed once in a game and
    cannot be disbanded or changed just because of a non-friendlier
    fleet. So it's a pretty straightforward, fairly simple toggle to
    get used to.

Ugh, it sounds like the HUD might get progressively messier as the
    game ends.

Shields charging slowly: active shields last several minutes, recharge
    fully in seconds, and charge every second they charge.

Shields wearing'shield' else doubles their size, can clip high above
    their skull, and are invincible to all other shields around your
    ship that border the shield and can deploy wall-mounted shields.
    Other shields like this one can only be used to shield the next
    vessel. Hit someone on the Pustules to activate shield shields. In
    an expedited, non-lethal, coordinated confrontation, it is fairly
    easy to compare stats to each shield for a ship to use.

Shield minus shields compensation: up to five shields are held at rank
    2 in your ship's shields. It's called 'delay,' but with a ton of
    fancy commands on screen, you don't have to know (or even see)
    their context or employment to make sure to compensate for some of
    the distortion inherent in that bit. Seconds before your shield
    starts charging (just seconds before it behaves like hiring
    another worker), the timer will start running indefinitely, which
    means shields can snap to the back of your ship if kept as low as
    possible. Avoid collisions caused by instantaneous cap.

Shield halves that match the length of the ship's hull: you can
    control the range between Shields, such as your Private Capital
    Signatures, where the ship's bulletproof liquor clock is set at
    half-space and your Heavy Construction Band Bump the Game's Game
    soundtrack to Fade let the corp count (if the corporation has
    sixty transports within one day of being prepped, it's probably
    already pretty close to ten minutes, but your mission selection
    loading speeds will be holding it back a little).

Surrounding explosions moving over the bow: it's deceivingly easy to
    line up targets like an Infallible Surface projection theater
    Explosion shield room high up in the sky: fire teams can run
    together and mean to demolish their own low-flying illions every
    once in a while, yet closely follow their targets like shadow
    warriors to the detriment of themselves. Covertly set to'square
    hit' for easier customization, Ubisoft's new ship weather HUD is
    the key to spreading effective damage throughout the entire ship
    using the standard meteorological updates.

Shaping your space: shape each bridge to create low-risk rockouts or
    pathogen/addage paths. This may be tedious, but it actually
    whizzes through by itself while simultaneously adding damage to
    giants mercenary ships.
```

# D    CTMCS PRELIMINARIES

## D.1    DISCRETE-TIME MARKOV CHAINS OVER FINITE-STATE SPACES

A discrete-time Markov Chain in a finite-state space is a stochastic process $X_1, X_2, \ldots, X_T$, where each state $X_t$ depends solely on the preceding state $X_{t-1}$. The states $X_t$ can take on any value from the set $\{1, 2, \ldots, S\}$, where $S$ denotes the total number of possible states, and $T$ represents the number of time steps. The probability of being in state $x$ at time $t$ is

$$p_t(X_t = x) = \sum_{y=1}^{S} p(X_t = x, X_{t-1} = y) = \sum_{y=1}^{S} p_{t|t-1}(X_t = x | X_{t-1} = y) p_{t-1}(X_{t-1} = y).$$
(177)

If we place all such probabilities $p_t(X_t = x)$ in a vector $\boldsymbol{s}_t$ of shape $S \times 1$, such that $\boldsymbol{s}_t(x) = p_t(X_t = x)$, then from above we can deduce that

$$\boldsymbol{s}_t = \boldsymbol{P} \boldsymbol{s}_{t-1},$$
(178)

where $\boldsymbol{P}(x, y) = p_{t|t-1}(X_t = x | X_{t-1} = y)$. Given an initial probability distribution $\boldsymbol{s}_0$ over states, the equation above fully determines the evolution of the probability over states with respect to time. If it is known that the state at time $t - 1$ is $y$, then $\boldsymbol{s}_t$ is simply column $y$ of $\boldsymbol{P}$. This implies that the sum of the elements of each column $y$ of $\boldsymbol{P}$ is one.

## D.2    CONTINUOUS-TIME MARKOV CHAINS OVER FINITE-STATE SPACES

It is possible to define a stochastic process with the Markov property in finite-state spaces, for $t \in [0, T]$, (Anderson, 2012). As previously, we can define a discrete-time process, on time points $\{0, \epsilon, \ldots, T - \epsilon, T\}$, such that there is $\epsilon$ probability of activating the previous transition mechanism when progressing from time $t - \epsilon$ to $t$, otherwise we stay where we are with probability $(1 - \epsilon)$. Removing the random variables to simplify notation, we have

$$p_t(x) = (1 - \epsilon) p_{t-\epsilon}(x) + \epsilon \sum_{y=1}^{S} p_{t|t-\epsilon}(x|y) p_{t-\epsilon}(y).$$
(179)

We notice that when $\epsilon = 1$ the equation above coincides with Equation (177), and in addition as before we can write Equation (179) in matrix form

$$\boldsymbol{s}_t = (1 - \epsilon) \boldsymbol{s}_{t-\epsilon} + \epsilon \boldsymbol{P} \boldsymbol{s}_{t-\epsilon} = (\boldsymbol{I} + \epsilon(\boldsymbol{P} - \boldsymbol{I})) \boldsymbol{s}_{t-\epsilon} = (\boldsymbol{I} + \epsilon \boldsymbol{Q}) \boldsymbol{s}_{t-\epsilon} \text{ , where } \boldsymbol{Q} = \boldsymbol{P} - \boldsymbol{I}. \quad (180)$$

From Equation (180), we see that $\frac{\boldsymbol{s}_t - \boldsymbol{s}_{t-\epsilon}}{\epsilon} = \boldsymbol{Q} \boldsymbol{s}_{t-\epsilon}$, which when taking the limit $\epsilon \to 0$ becomes

$$\frac{d\boldsymbol{s}_t}{dt} = \boldsymbol{Q} \boldsymbol{s}_t.$$
(181)

Given an initial probability distribution $\boldsymbol{s}_0$ over states, the equation above fully determines the evolution of the probability over states with respect to time. Indeed, the distribution over states at time $t$ is the solution of the linear ODE in (181): $\boldsymbol{s}_t = e^{t\boldsymbol{Q}} \boldsymbol{s}_0$. Therefore, if it is known that the state at time 0 is $j$, then $\boldsymbol{s}_t$ is simply column $j$ of $e^{t\boldsymbol{Q}}$. Naturally, one can rescale the time variable such that $t = \sigma(t)$, where $\sigma$ is monotonically increasing, $\sigma(0) = 0$ and $\lim_{t \to 1} \sigma(t) = T$, so that for $\boldsymbol{z}_0 := \boldsymbol{s}_0$, we get $e^{t\boldsymbol{Q}} \boldsymbol{s}_0 = e^{\sigma(t)\boldsymbol{Q}} \boldsymbol{z}_0 = \boldsymbol{z}_t$, and

$$\frac{d\boldsymbol{z}_t}{dt} = \sigma'(t) \boldsymbol{Q} \boldsymbol{z}_t = \boldsymbol{Q}_t \boldsymbol{z}_t, \text{ where } \boldsymbol{Q}_t = \sigma'(t) \boldsymbol{Q}.$$
(182)

Matrices $\boldsymbol{Q}_t = \sigma'(t)(\boldsymbol{P} - \boldsymbol{I})$ clearly satisfy the properties of transition-rate matrices (Suhov & Kelbert, 2008). Matrices $\boldsymbol{Q}_t$ are chosen such that: a) the matrix exponential $e^{\sigma(t)\boldsymbol{Q}}$ is easy to calculate, which is essential as $p_{t|0}(x|y) = e^{\sigma(t)\boldsymbol{Q}}(x, y)$; and b) $\boldsymbol{z}_1$ is an easy reference distribution to sample from (Austin et al., 2021; Campbell et al., 2022).

Finally, similar to diffusion processes in continuous spaces, the continuous-time Markov chain in Equation (182) also admits a reverse process (Kelly, 1979; Sun et al., 2023):

$$\frac{d\boldsymbol{z}_{1-t}}{dt} = \bar{\boldsymbol{Q}}_{1-t} \boldsymbol{z}_{1-t},$$
(183)

where $\bar{\boldsymbol{Q}}_t(x,y) = \boldsymbol{Q}_t(y,x)\frac{p_t(x)}{p_t(y)}$ for $x \neq y$, and $\bar{\boldsymbol{Q}}_t(x,x) = -\sum_{y\neq x}\bar{\boldsymbol{Q}}_t(y,x)$. Since we can easily sample from the reference distribution, the only unknowns preventing us from being able to run backwards are the ratios $\frac{p_t(x)}{p_t(y)}$ also known as concrete scores (Meng et al., 2022), which we desire to model using a neural network. Once such ratios are modeled we can generate samples from the learned data distribution $p_0^\theta$ by discretizing Equation (183) as follows:

$$p(x_{t-\epsilon} = y \mid x_t = x) = \delta_x(y) + \bar{\boldsymbol{Q}}_t(y,x)\epsilon + O(\epsilon^2). \tag{184}$$

