# OpenReview forum: "Efficient Perplexity Bound and Ratio Matching in Discrete Diffusion Language Models"
_ICLR.cc/2025/Conference — ICLR 2025 Poster_

### Official Review · Reviewer_78NZ · 2024-11-03

**Soundness:** 3
**Presentation:** 3
**Contribution:** 3
**Rating:** 8
**Confidence:** 3

**Summary:**

This paper proposes novel theoretical results and empirically validated improvements for diffusion language models in the continuous time discrete state space framework.  Specifically it establishes new bounds and relationships for KL divergence between empirical and learnt distributions, including an improved upper bound on perplexity of the estimated model.  Furthermore, it introduces a novel transition-rate matrix, termed Roulette diffusion, which generalizes both uniform transition-rate and absorb transition-rate matrices and admits an analytical matrix exponential.  Empirical results demonstrate that use of cross-entropy discrete diffusion (CEDD) to carry out ratio-matching (modeling of ‘concrete scores’) for uniform, absorbing, or roulette diffusion, is superior to ratio-matching using score-entropy (SEDD), and the proposed upper bound on perplexity is tighter than the previously published bound.

**Strengths:**

* Advances state of the art in continuous time discrete space diffusion language modeling with novel theoretical results pertaining to KL divergence between empirical and model distributions, including a tighter bound on model perplexity.
* Empirical results demonstrating the superiority of the proposed perplexity bound, as well as uniformly better perplexity performance of the CEDD approach over SEDD.
* Well written paper.

**Weaknesses:**

The largest weakness of the paper in my view centers around evaluation of the proposed models.  The paper presents perplexity results under GPT-2 as well as the perplexity bounds, but it is also acknowledged that GPT-2 based measures have their own potential issues, and with bounds a question of tightness always remains.  Thus, how good these models truly are, and how do they compare/contrast with more traditional auto-regressive LLMs remains an open question.

**Questions:**

Related to the weakness above, would it be possible to take these language models through the supervised fine-tuning steps and evaluate on set of tasks that LLMs are commonly evaluated on?

---

> ### Author Response · Authors · 2024-11-20
>
> We thank Reviewer **78NZ**  for the deep and careful reading of our work. We address the concerns as follows.
>
> ---
>
> __The largest weakness of the paper in my view centers around evaluation of the proposed models. The paper presents perplexity results under GPT-2 as well as the perplexity bounds...__
>
> We thank the reviewer for highlighting this important concern. We fully agree that `GPT-2` based measures have limitations. To address this to a degree, we reproduced the generative perplexity results (column 1 of Tables 1 and 2 in the main paper) using `LLama 3.1` with $8$ billion parameters. Importantly, the model rankings remained unchanged, supporting the consistency of our findings. Furthermore, following **R-wpbF**'s feedback regarding a recently identified issue in the sampling procedure of the original SEDD implementation by Lou et al. (2024), related to the use of  `Float32` (FL32) precision, we have conducted additional evaluations using `Float64` (FL64) precision. The aforementioned numerical precision issues only affected column 1 of Table 1 and column 1 of Table 2 in the paper. For transparency, we provide tables (A and B) showing both the original results (FL32 precision with `GPT-2 Large`) and the updated results (FL64 precision) evaluated using both `GPT-2 Large` and `LLama 3.1 8B`.
>
>
> **Table-A:** Results comparing SEDD, SEDD scaled (SEDDs), CEDD and CEDD* in terms of generative perplexity.
>
>
> | **Model (L=128)**    | **GenPerp `FL32-GPT2L`** | **GenPerp `FL64-GPT2L`** | **GenPerp `FL64-LLama8B`** |
> |-----------------------|------------------------|-------------------------|---------------------------|
> | SEDD Absorb      | 83.62                  | 172.35                  | 212.15                    |
> | SEDDs Absorb      | 79.74                  | 166.35                  | 206.34                    |
> | CEDD Absorb       | 74.19                  | 148.21                  | 185.90                    |
> | CEDD\* Absorb      | **72.13**              | **143.86**              | **183.74**                |
> |-----------------------|------------------------|-------------------------|---------------------------|
> | SEDD Roulette     | 87.81                  | 178.94                  | 220.00                    |
> | SEDDs Roulette    | 83.36                  | 172.93                  | 212.07                    |
> | CEDD Roulette     | 76.71                  | 167.67                  | 208.84                    |
> | CEDD\* Roulette    | **72.31**              | **158.56**              | **197.79**                |
> |-----------------------|------------------------|-------------------------|---------------------------|
> | SEDD Uniform      | 175.49                 | 169.66                  | 206.91                    |
> | SEDDs Uniform     | 171.07                 | 163.88                  | 200.90                    |
> | CEDD Uniform      | **168.35**             | **161.84**              | **200.09**                |
> | CEDD\* Uniform     | 179.30                 | 175.42                  | 213.99                    |
>
>
> **Table-B:** Results comparing SEDD, SEDD scaled (SEDDs), CEDD and CEDD* in terms of generative perplexity.
>
> | Model (Absorb)  | GenPerp `FL32-GPT2L` | GenPerp `FL64-GPT2L` | GenPerp `FL64-LLama8B` |
> |-----------------|---------------------|---------------------|-----------------------|
> | SEDDs L=128     | 79.74              | 166.35             | 206.34               |
> | CEDD\* L=128     | **72.13**              | **143.86**             | **183.74**               |
> | CEDDT L=128     | 74.07              | 154.04             | 195.41               |
> |-----------------|---------------------|---------------------|-----------------------|
> | SEDDs L=1024    | 40.95              | 105.27             | 111.87               |
> | CEDD\* L=1024    | **40.93**\*             | **101.83**             | **107.32**               |
> | CEDDT L=1024    | 42.18              | 108.88             | 115.60               |
> |-----------------|---------------------|---------------------|-----------------------|
> | GPT-2 L=1024    | 41.02              | 41.02\*             | 50.25\*               |
>
> Regarding the perplexity bounds, we acknowledge the inherent question of tightness. We demonstrate that when the model accurately learns the ratios, the bound becomes an exact measure of perplexity. However, as the reviewer correctly points out, this does not ensure that a model with a lower perplexity bound evaluation than another model necessarily exhibits a lower corresponding perplexity. Therefore the precise and practical measurement of perplexity in the discrete diffusion framework remains an important open topic.
>
> That said, our empirical results show that the relative rankings of methods based on perplexity bounds align closely with their generative perplexity performance. This alignment provides strong empirical evidence supporting the utility of these bounds in assessing the performance of discrete diffusion models.
>
> ---

---

> > ### Author Response · Authors · 2024-11-20
> >
> > __Related to the weakness above, would it be possible to take these language models through the supervised fine-tuning steps and evaluate on set of tasks that LLMs are commonly evaluated on?__
> >
> >
> >
> > This is indeed an important question. Due to the paper's current length and the short discussion period however, we are forced to address this in a follow-up study. That said, to demonstrate the framework's practical utility, and in line with Reviewer **cDEx**'s suggestion, we further evaluated our *uniform* and *roulette* diffusion models on an unsupervised spelling correction task. Following the character-level training practice outlined in [1], we trained CEDD and SEDD models on `War and Peace` ($3.3$ million tokens) and applied them to the spelling correction task for `Crime and Punishment (CAP)` and `Pride and Prejudice (PAP)`. Using a batch size of $32$ sequences with a sequence length of 128 tokens, we trained models for $25$ K and $50$ K iterations. The test sets were corrupted by perturbing 5\% of the characters and predictions were selected based on the tokens assigned the highest probability by the model. Accuracy scores are detailed below, which highlight the practical utility of our framework.
> >
> >
> > **Table C:** The percentages of characters corrected accurately after $25$ K training iterations.
> >
> > | **Model (L=128)** | CEDD\* Uniform | SEDD Uniform | CEDD\* Roulette | SEDD Roulette |
> > |-------------------|--------------|--------------|---------------|---------------|
> > | PAP               | 89.5         | 86.9         | **89.7**      | 85.1          |
> > | CAP               | 89.5         | 87.5         | **90.3**      | 85.8          |
> >
> > **Table D:** The percentages of characters corrected accurately after $50$ K training iterations.
> >
> > | **Model (L=128)** | CEDD\* Uniform | SEDD Uniform | CEDD\* Roulette | SEDD Roulette |
> > |-------------------|--------------|--------------|---------------|---------------|
> > | PAP               | 90.7         | 89.9         | **90.8**      | 87.9          |
> > | CAP               | 91.2         | 90.3         | **91.3**      | 88.6          |
> >
> >
> > ----
> >
> > We again sincerely thank the reviewer for the careful reading of the paper.
> >
> > __References__
> >
> > [1] Argmax Flows and Multinomial Diffusion: Learning Categorical Distributions, Hoogeboom et al.

---

### Official Review · Reviewer_wpbF · 2024-11-03

**Soundness:** 3
**Presentation:** 2
**Contribution:** 3
**Rating:** 5
**Confidence:** 4

**Summary:**

The paper presents, to the best of my knowledge, novel theorems that eventually lead to an enhanced performance compared to the baseline (SEDD). The authors further introduce a predictor-corrector version for the ratio matching framework called roulette diffusion.

**Strengths:**

The paper is well written and introduces, to the best of my knowledge, a novel set of theorems.

**Weaknesses:**

* My primary concern is that the authors appear to utilize the official implementation of SEDD, which has been demonstrated to have a flaw in the gumbel max trick used for sampling. In [1], the authors show that when using sampling in low precision for high dimensional distribution it leads to an effect similar to sampling with temperature. This is highly related here as the exponential in the proposed method may even anneal the temperature more.
* Prior Work: The authors only compare their results with SEDD, neglecting other relevant works in their analysis, such as [2] and [3].
* Predictor-Corrector: The aforementioned works incorporate some forms of predictor-corrector methods, and I believe it is necessary to include comparisons with these approaches.

[1] Masked Diffusion Models are Secretly Time-Agnostic Masked Models and Exploit Inaccurate Categorical Sampling - https://arxiv.org/pdf/2409.02908

[2] Generative Flows on Discrete State-Spaces: Enabling Multimodal Flows with Applications to Protein Co-Design https://arxiv.org/pdf/2402.04997

[3] Discrete Flow Matching https://arxiv.org/abs/2407.15595

**Questions:**

See above

---

> ### Author Response · Authors · 2024-11-20
>
> We sincerely thank the Reviewer (**wpbF**) for their time and efforts in carefully reviewing our paper.  We address the raised concerns below.
>
> ---
>
> __My primary concern is that the authors appear to utilize the official implementation of SEDD, which has been demonstrated to have a flaw in the gumbel max trick used for sampling. In [1], the authors show that when using sampling in low precision for high dimensional distribution it leads to an effect similar to sampling with temperature. This is highly related here as the exponential in the proposed method may even anneal the temperature more.__
>
> We thank the reviewer for this insightful feedback as [Lou et al 2024]'s (original SEDD) implementation of the sampling procedure appears to suffer from numerical issues. However, we emphasize that this issue only affects the results in Column 1 (GenPerp) of Table 1 and Column 1 of Table 2. Specifically, the perplexity bounds $e^\frac{{J_1}}{L}$ and $e^\frac{{J_2}}{L}$ are not affected by the sampling issue, as they do not require generated samples to calculate (the trained model is evaluated on test sets). The other columns of Table 1 and Table 2 present results with regard to the perplexity bound ($e^\frac{{J_1}}{L}$). Table 3 contains results in terms of $e^\frac{{J_1}}{L}$ and $e^\frac{{J_2}}{L}$ only. Table 4 contains the $e^\frac{{J_2}}{L}$ version of Table 2 and contains no results in terms of the flawed generative perplexity. Therefore, columns titled Lambada, Wikitext2, PTB, Wikitext103, and 1BW are not affected in any case.
>
>
> To address the reviewer's concerns, we recomputed the generative perplexity using proper sampling with `torch.float64` in the Gumbel to re-compute Columns 1 of Table 1 and 2. When we increase the precision using float 64 (Fl64), generative perplexity increases significantly as chances of sampling tokens with low probability increase to the proper level. Yet, the relative ranking of the models remained unchanged, as it can be seen below and in the updated paper. We expect such a result, as we use the learned probabilities to construct the scores, thus the flawed implementation in Lou et al. 2024 serves in both our and their favour by precisely the same mechanics.
>
>
> **Table-A:** Results comparing SEDD, SEDD scaled (SEDDs), CEDD and CEDD* in terms of generative perplexity.
>
>
> | **Model (L=128)**    | **GenPerp Fl32-GPT2L** | **GenPerp Fl64-GPT2L** | **GenPerp Fl64-LLama8B** |
> |-----------------------|------------------------|-------------------------|---------------------------|
> | SEDD Absorb      | 83.62                  | 172.35                  | 212.15                    |
> | SEDDs Absorb      | 79.74                  | 166.35                  | 206.34                    |
> | CEDD Absorb       | 74.19                  | 148.21                  | 185.90                    |
> | CEDD\* Absorb      | **72.13**              | **143.86**              | **183.74**                |
> |-----------------------|------------------------|-------------------------|---------------------------|
> | SEDD Roulette     | 87.81                  | 178.94                  | 220.00                    |
> | SEDDs Roulette    | 83.36                  | 172.93                  | 212.07                    |
> | CEDD Roulette     | 76.71                  | 167.67                  | 208.84                    |
> | CEDD\* Roulette    | **72.31**              | **158.56**              | **197.79**                |
> |-----------------------|------------------------|-------------------------|---------------------------|
> | SEDD Uniform      | 175.49                 | 169.66                  | 206.91                    |
> | SEDDs Uniform     | 171.07                 | 163.88                  | 200.90                    |
> | CEDD Uniform      | **168.35**             | **161.84**              | **200.09**                |
> | CEDD\* Uniform     | 179.30                 | 175.42                  | 213.99                    |

---

> ### Author Response · Authors · 2024-11-20
>
> **Table-B:** Results comparing SEDD, SEDD scaled (SEDDs), CEDD and CEDD* in terms of generative perplexity.
>
> | Model (Absorb)  | GenPerp Fl32-GPT2L | GenPerp Fl64-GPT2L | GenPerp Fl64-LLama8B |
> |-----------------|---------------------|---------------------|-----------------------|
> | SEDDs L=128     | 79.74              | 166.35             | 206.34               |
> | CEDD\* L=128     | **72.13**              | **143.86**             | **183.74**               |
> | CEDDT L=128     | 74.07              | 154.04             | 195.41               |
> |-----------------|---------------------|---------------------|-----------------------|
> | SEDDs L=1024    | 40.95              | 105.27             | 111.87               |
> | CEDD\* L=1024    | **40.93**\*             | **101.83**             | **107.32**               |
> | CEDDT L=1024    | 42.18              | 108.88             | 115.60               |
> |-----------------|---------------------|---------------------|-----------------------|
> | GPT-2 L=1024    | 41.02              | 41.02\*             | 50.25\*               |
>
>
>
> In the Tables above we used both GPT2 large (GPT2L) and LLama 3.1 8B for measuring the generative perplexity. In addition, in the case of Uniform diffusion, due to the change in the sampling process, we found that a more optimal hyperparameter $\sigma^\theta$ is $0.0015$ which better balances the trade-off between generative perplexity and the perplexity bounds ($e^\frac{{J_1}}{L}$ and $e^\frac{{J_2}}{L}$), thus we report the results for this choice.
>
> ---
>
> __Prior Work: The authors only compare their results with SEDD, neglecting other relevant works in their analysis, such as [2] and [3].__
>
>
> We thank the reviewer for this remark. We concentrated our experiments on comparing SEDD and CEDD across various settings, demonstrating the empirical superiority of the latter. We did not focus on surpassing the state-of-the-art which is still autoregressive, and thus exluded other important approaches. Additionally, the lack of a perplexity bound for discrete flows until recently made comparisons challenging.
>
> We appreciate the reviewer’s suggestion as it provides a meaningful extension to our work. To address this, we have now compared our models against Discrete Flow Matching [2] which generalizes [1], focusing on sequence lengths of 128 (as in Table 1 of the main paper). We report both generative perplexity, and the perplexity bounds, where we employed the recently proposed bound in Equation (24) of [3] for discrete flows. Specifically, we compare discrete flows when  convex interpolants are used (Equation 9, in [2]), with schedules $k_t=t$ as in [1], as well as $k_t=t^2$. For fairness, all comparisons were conducted under identical settings, including network architecture, number of parameters, training updates, sampling steps, batch size, and other hyperparameters.
>
> | Model (L=128)          | GenPerp (Fl64-GPT2L) | LAMBADA | WikiText2 | PTB    | WikiText103 | 1BW    |
> |-------------------------|-----------------------|---------|-----------|--------|-------------|--------|
> | SEDDs Absorb            | 166.35               | 67.05   | 69.37     | 208.69 | 69.17       | 83.87  |
> | CEDD\* Absorb         | **143.86**           | **64.60** | **65.04** | **192.99** | **64.69** | **79.81** |
> | Discrete flow $k_t=t$   | 145.48               | 71.90   | 71.20     | 221.15 | 70.84       | 82.63  |
> | Discrete flow $k_t=t^2$ | 152.70               | 72.31   | 72.87     | 215.30 | 72.55       | 85.82  |
>
> ---

---

> ### Author Response · Authors · 2024-11-21
>
> __Predictor-Corrector: The aforementioned works incorporate some forms of predictor-corrector methods, and I believe it is necessary to include comparisons with these approaches.__
>
> *EDITED (additional experiments):* Unfortunately a fair comparison is challenging in general. The predictor-corrector method in [2] requires learning two probability velocities  $\hat{u}$ and $\check{u}$. Our correction strategy, on the other hand, by definition necessitates the usage of a single neural network.  In addition, a perplexity bound is still needed for comparison but does not exist in the literature.
>
> However, in the specific case where the source distribution is $\delta_m(x^i)$, as used in the experiments addressing the second concern, $\check{u}$ has an analytical expression $\check{u}(x^i, x_t) = \frac{1}{t}(\delta_{x_t^i}(x^i) - \delta_m(x^i))$. Yet, a perplexity bound is still needed for comparison but does not exist in the literature. Nevertheless, to address the reviewer’s concerns, and inspired by Expression (21) in [3], we derived a bound and conducted additional experiments.
>
>
> We evaluated the corrector model using using generative perplexity and the derived perplexity bound. The perplexity bound was derived as follows:
>
> By substituting $v_{t}^i(x^i, x_{t})=\alpha_t\hat{u}-\beta_t \check{u}$ in Equation (21) of [3], where $\check{u}(x^i, x_t) = \frac{1}{t}(\delta_{x_t^i}(x^i) - \delta_m(x^i))$, the bound in Equation (24) of [3] becomes
>
> $\int_0^1 \sum_{x_{t}}p_{t|1}(x_{t}|x_1) \sum_{i=1}^L \left[ -\delta_{x_1^i\neq x_t^i} \frac{\dot{k_t}}{1-k_t}\left(\log{\alpha_t  p_{1|t}^i(x^i_1| x_{t};\theta)}+1\right)+
>     \alpha_t \frac{\dot{k_t}}{1-k_t}\left( 1- p_{1|t}^i(x_t^i| x_{t};\theta) \right)+\beta_t\frac{\dot{k_t}}{k_t}\delta_{m\neq x_t^i}\right]dt.$
>
> We chose $\alpha_t=1+\alpha t^a(1-t)^b$ and $\beta_t=\alpha_t-1$ as in [2]. Furthermore based on Figure 7 in [2] we chose $\alpha=1$ and $a=0.5$ and $b=0.5$. As before, the choice $k_t=t$ was made.
>
> While this model performs roughly the same as our CEDD* absorb in terms of generative perplexity, its perplexity bound scores are considerably worse. The results are summarized in the table below.
>
> | Model (L=128)          | GenPerp (Fl64-GPT2L) | LAMBADA | WikiText2 | PTB    | WikiText103 | 1BW    |
> |-------------------------|-----------------------|---------|-----------|--------|-------------|--------|
> | Discrete Flow Corrector   | 143.38             | 114.40   | 113.05    | 352.32 | 112.87       | 130.41  |
>
>
>
> We again sincerely thank the reviewer for the constructive comments which have improved the integrity of our paper.
>
> __References__
>
> [1] Generative Flows on Discrete State-Spaces: Enabling Multimodal Flows with Applications to Protein Co-Design https://arxiv.org/pdf/2402.04997
>
> [2] Discrete Flow Matching https://arxiv.org/abs/2407.15595
>
> [3] Minibatch Optimal Transport and Perplexity Bound Estimation in Discrete Flow Matching https://arxiv.org/pdf/2411.00759

---

### Official Review · Reviewer_cDEx · 2024-11-04

**Soundness:** 3
**Presentation:** 3
**Contribution:** 3
**Rating:** 8
**Confidence:** 3

**Summary:**

This paper presents new results for discrete diffusion models trying to estimate density ratios for language modeling task. The authors' contributions are threefold: (i) they provide an upper bound $J_2$ on the perplexity of discrete ratio-matching diffusion language models which is computationally more efficient than the bound $J_1$ known from the literature, and, as shown through empirical evaluation, is consistently tighter; (ii) they derive a novel training objective called cross-entropy discrete diffusion (CEDD) loss by making use of reparametrization of concrete scores; (iii) they propose an alternative to known transition-rate matrices referred to as "absorb" and "uniform" and derive exponential of this new "roulette" transition matrix necessary to compute conditional probabilities efficiently.

**Strengths:**

The novelty of the contents of this paper consists of several theoretical findings and undoubtfully is the main strength of this paper. These findings include a novel type of transition matrices for discrete diffusion models along with all the facts necessary to use it in practice. The feasibility of using this type of forward diffusions was shown through experiments. Second, a score reparameterization was used to derive a CEDD model competitive with a common SEDD model. Lastly, the novel bound $J_2$ on the perplexity of the generated text is proposed, and the experiments demonstrate that it is consistenly better than previously used bound $J_1$.
Overall, I think this is a good paper from the point of view of originality and significance of questions raised in it since discrete diffusion language modeling is the topic that so far receives relatively little attention compared to continuous-domain diffusion models.

**Weaknesses:**

1. I am not sure about the correctness of perplexity evaluation in Section 4.1. To evaluate likelihood (which is equivalent to perplexity evaluation) with an imperfect model may lead to unreliable results as you mention in line 362. I would suggest to use some better and more modern language models instead of GPT-2 and report some aggregated score.
2. CEDD* differs from CEDD by some hand-crafted time-dependent weights (line 324) and the results across different transition matrices types differ as follows from Table 1. It raises some concerns about CEDD training objective in terms of stability and sensitivity to the choice of $\omega(t)$.
3. Language models trained on language modeling task can be evaluated not only from the point of view of perplexity on this task. I am not talking about some NLU tasks we can finetune language models to - I've never seen such attempts in the context of diffusion models. But, e.g. in [1] diffusion language models are shown to be capable of spell checking / grammatical error correction. So, maybe it could be more impactful if the authors evaluated their CEDD also on tasks like this.
4. When it comes to discrete diffusion language models, I always wonder about their scalability in terms of, say, model/batch size, or data size, or vocabulary size. Common language models based on transformers typically have some useful scalability properties and their performance increases with the increase in data/model size for carefully chosen training hyperparameters. Papers on discrete diffusion language models usually do not discuss these issues, but in my opinion it is important if we want to prove that such models really can compete with modern language models.

[1] Argmax Flows and Multinomial Diffusion: Learning Categorical Distributions, Hoogeboom et al.

**Questions:**

The experiments reveal that $J_2$ is tighter that $J_1$ in all cases you studied. Do you have intuitive explanation of this phenomenon? Comparing the formulae (13) and (16) I can see that $J_2$ contains function $\bar{l}$ differing from $l$ by the $K$ function taken at conditional scores. This $K$ can have different signs depending on its arguments, and $J_2$ also has additional integral term and entropy of the reference distribution. Do you have any idea on how to explain the superior performance of $J_2$ compared to $J_1$?

---

> ### Author Response · Authors · 2024-11-19
>
> We thank Reviewer **cDEx** for the time and efforts in carefully reviewing our paper. We address the specific concerns as follows:
>
> ---
>
> __I am not sure about the correctness of perplexity evaluation in Section 4.1. To evaluate likelihood (which is equivalent to perplexity evaluation) with an imperfect model may lead to unreliable results as you mention in line 362. I would suggest to use some better and more modern language models instead of GPT-2 and report some aggregated score.__
>
> We used GPT-2 Large in our evaluations to remain consistent with the baseline from the original SEDD implementation. That said, we agree that GPT-2 Large may not be the most suitable evaluation model for this task. Following the reviewer's suggestion, we reproduced the generative perplexity results presented in column 1 of Table 1 and column 1 of Table 2 in the main paper using LLama 3.1 with 8 billion parameters (Tables A and B below). The results indicate that the model rankings remained unchanged.
>
> Additionally, as Reviewer **wpbF** noted, an issue in the sampling procedure within the original SEDD implementation by Lou et al. (2024) was identified very recently, which only impacts the generative perplexity (GenPerp) results in column 1 of Tables 1 and 2. Importantly, this issue does not affect the perplexity bound results for columns representing measurements using
> $J_1$ and $J_2$ (that is, columns titled: Lambada, Wikitext2, PTB, Wikitext103, and 1BW).
>
> For transparency, we have provided tables with both the original results—where sampling used Float32 (FL32) precision and evaluation employed GPT-2 Large (GPT2L)—and the corrected results utilizing Float64 (FL64) precision, where quality is measured by using both GPT2L and LLama 3.1 8B.
>
> **Table-A:** Results comparing SEDD, SEDD scaled (SEDDs), CEDD and CEDD* in terms of generative perplexity.
>
>
> | **Model (L=128)**    | **GenPerp Fl32-GPT2L** | **GenPerp Fl64-GPT2L** | **GenPerp Fl64-LLama8B** |
> |-----------------------|------------------------|-------------------------|---------------------------|
> | SEDD Absorb      | 83.62                  | 172.35                  | 212.15                    |
> | SEDDs Absorb      | 79.74                  | 166.35                  | 206.34                    |
> | CEDD Absorb       | 74.19                  | 148.21                  | 185.90                    |
> | CEDD\* Absorb      | **72.13**              | **143.86**              | **183.74**                |
> |-----------------------|------------------------|-------------------------|---------------------------|
> | SEDD Roulette     | 87.81                  | 178.94                  | 220.00                    |
> | SEDDs Roulette    | 83.36                  | 172.93                  | 212.07                    |
> | CEDD Roulette     | 76.71                  | 167.67                  | 208.84                    |
> | CEDD\* Roulette    | **72.31**              | **158.56**              | **197.79**                |
> |-----------------------|------------------------|-------------------------|---------------------------|
> | SEDD Uniform      | 175.49                 | 169.66                  | 206.91                    |
> | SEDDs Uniform     | 171.07                 | 163.88                  | 200.90                    |
> | CEDD Uniform      | **168.35**             | **161.84**              | **200.09**                |
> | CEDD\* Uniform     | 179.30                 | 175.42                  | 213.99                    |
>
>
> **Table-B:** Results comparing SEDD, SEDD scaled (SEDDs), CEDD and CEDD* in terms of generative perplexity.
>
> | Model (Absorb)  | GenPerp Fl32-GPT2L | GenPerp Fl64-GPT2L | GenPerp Fl64-LLama8B |
> |-----------------|---------------------|---------------------|-----------------------|
> | SEDDs L=128     | 79.74              | 166.35             | 206.34               |
> | CEDD\* L=128     | **72.13**              | **143.86**             | **183.74**               |
> | CEDDT L=128     | 74.07              | 154.04             | 195.41               |
> |-----------------|---------------------|---------------------|-----------------------|
> | SEDDs L=1024    | 40.95              | 105.27             | 111.87               |
> | CEDD\* L=1024    | **40.93**\*             | **101.83**             | **107.32**               |
> | CEDDT L=1024    | 42.18              | 108.88             | 115.60               |
> |-----------------|---------------------|---------------------|-----------------------|
> | GPT-2 L=1024    | 41.02              | 41.02\*             | 50.25\*               |
>
> ---

---

> ### Author Response · Authors · 2024-11-19
>
> __CEDD* differs from CEDD by some hand-crafted time-dependent weights (line 324) and the results across different transition matrices types differ as follows from Table 1. It raises some concerns about CEDD training objective in terms of stability and sensitivity to the choice of $w(t)$.__
>
> We thank the reviewer for their careful reading of the paper. We would like to clarify that the weighting term in the cross-entropy loss does not compromise stability; rather it serves as an important design choice that can enhance the performance in specific scenarios. For example, in the case of absorb dynamics, increasing the weight when $t$ is close to zero can lead to improved results while overincreasing it, as in the case of CEDDT, can lead to a drop in performance, albeit a relatively modest one.
>
> That said, we acknowledge that the effectiveness of this term is influenced by the diffusion dynamics. Overall, while the selection of $w(t)$ must be made empirically—an inherent challenge in most diffusion models—the loss function remains stable across all tested configurations.
>
> ---
>
> __Language models trained on language modeling task can be evaluated not only from the point of view of perplexity on this task. I am not talking about some NLU tasks we can finetune language models to - I've never seen such attempts in the context of diffusion models. But, e.g. in [1] diffusion language models are shown to be capable of spell checking / grammatical error correction. So, maybe it could be more impactful if the authors evaluated their CEDD also on tasks like this.__
>
> We thank the reviewer for this valuable suggestion. We agree that evaluating CEDD on unsupervised spell-checking task provides a meaningful extension to our work. Following the character-level training practice outlined in [1], we trained CEDD and SEDD models on `War and Peace` (3.3 million tokens) and applied them to the spelling correction task for `Crime and Punishment (CAP)` and `Pride and Prejudice (PAP)` texts. Using a batch size of 32 sequences with a sequence length of 128 tokens, we trained models for $25$ K and $50$ K iterations. The test sets were corrupted by perturbing 5\% of the characters and predictions were selected based on the tokens assigned the highest probability by the model. Accuracy scores are detailed below, which highlight the practical utility of our framework.
>
>
> **Table C:** The percentages of characters corrected accurately after $25$ K training iterations.
>
> | **Model (L=128)** | CEDD\* Uniform | SEDD Uniform | CEDD\* Roulette | SEDD Roulette |
> |-------------------|--------------|--------------|---------------|---------------|
> | PAP               | 89.5         | 86.9         | **89.7**      | 85.1          |
> | CAP               | 89.5         | 87.5         | **90.3**      | 85.8          |
>
> **Table D:** The percentages of characters corrected accurately after $50$ K training iterations.
>
> | **Model (L=128)** | CEDD\* Uniform | SEDD Uniform | CEDD\* Roulette | SEDD Roulette |
> |-------------------|--------------|--------------|---------------|---------------|
> | PAP               | 90.7         | 89.9         | **90.8**      | 87.9          |
> | CAP               | 91.2         | 90.3         | **91.3**      | 88.6          |
>
>
> We will expand the experimental section to incorporate the results presented above.
>
> ---
>
> __When it comes to discrete diffusion language models, I always wonder about their scalability in terms of, say, model/batch size, or data size, or vocabulary size. Common language models based on transformers typically have some useful scalability properties and their performance increases with the increase in data/model size for carefully chosen training hyperparameters. Papers on discrete diffusion language models usually do not discuss these issues, but in my opinion it is important if we want to prove that such models really can compete with modern language models.__
>
> We thank the reviewer for raising this important point. Scalability is a critical aspect of discrete diffusion language models and presents exciting opportunities for future exploration. While we believe these models should follow similar scaling laws as their autoregressive counterparts, given the shared network architecture and loss function, the validation of this hypothesis remains to be addressed in future work through thorough empirical testing. We will expand on this discussion in the *Future Outlook* section of the paper.
>
> ---

---

> > ### Author Response · Authors · 2024-11-19
> >
> > __The experiments reveal that $J_2$ is tighter than $J_1$ in all cases you studied. Do you have an intuitive explanation of this phenomenon? Comparing the formulae (13) and (16), I can see that $J_2$ contains function $\tilde{l}$ differing from $l$ by the $K$ function taken at conditional scores. This $K$ can have different signs depending on its arguments, and $J_2$ also has an additional integral term and entropy of the reference distribution. Do you have any idea on how to explain the superior performance of $J_2$ compared to $J_1$?__
> >
> > As the reviewer points out, $J_2$ excludes the $K$ term, which can take both positive and negative values. Instead, $J_2$ incorporates an entropy term (always non-negative) and subtracts a non-negative integral term.
> >
> > Our intuition and primary motivation for deriving $J_2$ was to find a direct bound of the cross entropy by extending the discrete diffusion framework to parallel the results in [2]. We expected this direct bound to be tighter, as it avoids averaging over bounds on the negative likelihood of individual data points, which is inherent in $J_1$. This structural difference could contribute to its tighter performance, as $J_2$ leverages a more global evaluation in contrast to the more pointwise $J_1$.
> >
> > ---
> >
> > #### References
> >
> > [1] Argmax Flows and Multinomial Diffusion: Learning Categorical Distributions, Hoogeboom et al.
> >
> > [2] Maximum Likelihood Training of Score-Based Diffusion Models, Song et al.

---

> ### Comment · Reviewer_cDEx · 2024-11-26
>
> Thank you for the detailed responses. I think that the revision of the paper during the rebuttal phase improved it. I will increase my score to reflect that.

---

> > ### Author Response · Authors · 2024-11-28
> >
> > Thank you for your thoughtful feedback and for recognizing the improvements. We truly appreciate your time and consideration!

---

### Official Review · Reviewer_5C4V · 2024-11-05

**Soundness:** 4
**Presentation:** 3
**Contribution:** 3
**Rating:** 6
**Confidence:** 4

**Summary:**

This paper provides three new theorems concerning the KL divergence between the data and the learned distribution, improving model evaluation through a bound.
A new transition-rate matrix (roulette diffusion matrix) that allows for token correction after unmasking in the reverse process is introduced, deriving its exponential matrix to enable efficient training/sampling. Finally, denoising cross entropy loss over score entropy is proposed for training discrete diffusion models, with up to 10% lower perplexity/generative-perplexity, and 15% faster training steps.

**Strengths:**

The paper is well-written and clear.
Developing efficient perplexity bound and the training algorithm for discrete diffusion language models is an important yet challenging problem.

The particular forms of the bounds, the roulette diffusion matrix, the denoising cross entropy loss and are interesting and new. Both theoretical development and empirical evaluations are conducted.

**Weaknesses:**

From Table 2, overall GPT-2 performs best over SEDDs, CEDD*, and CEDDT. The practical usefulness of the new bound and training algorithm is not completely clear.

**Questions:**

Line 147: \bf{x}^i_0 = x^i_0, is it a typo?

---

> ### Author Response · Authors · 2024-11-19
>
> We thank the Reviewer **5C4V** for their time and valuable feedback. We address the concerns as follows:
>
> ---
>
> __From Table 2, overall GPT-2 performs best over SEDDs, CEDD*, and CEDDT. The practical usefulness of the new bound and training algorithm is not completely clear.__
>
> We acknowledge that discrete diffusion models have not yet outperformed autoregressive models in the realm of language modeling. Nonetheless, we believe in the promise of this approach, and, as highlighted by the reviewer, this paper offers several significant contributions to the framework. Moreover, this framework can be particularly advantageous when applied to data without a left-to-right bias, such as protein design. Recent research [1] also suggests that discrete diffusion models have the potential to accelerate the generation of categorical data. Additionally, these models are naturally suited for specific tasks, including spelling correction.
>
> To demonstrate this, as advised by Reviewer **cDEx**,  we further evaluated our *uniform* and *roulette* diffusion models on an unsupervised spell-checking task. Following the character-level training practice outlined in [2], we trained CEDD and SEDD models on `War and Peace` (3.3 million tokens) and applied them to spelling correction task for `Crime and Punishment (CAP)` and `Pride and Prejudice (PAP)`. Using a batch size of 32 sequences with a sequence length of 128 tokens, we trained models for $25$ K and $50$ K iterations. The test sets were corrupted by perturbing 5\% of the characters and predictions were selected based on the tokens assigned the highest probability by the model. Accuracy scores are detailed below, which highlight the practical utility of our framework.
>
> **Table A:** The percentages of characters corrected accurately after $25$ K training iterations.
>
> | **Model (L=128)** | CEDD\* Uniform | SEDD Uniform | CEDD\* Roulette | SEDD Roulette |
> |-------------------|--------------|--------------|---------------|---------------|
> | PAP               | 89.5         | 86.9         | **89.7**      | 85.1          |
> | CAP               | 89.5         | 87.5         | **90.3**      | 85.8          |
>
> **Table B:** The percentages of characters corrected accurately after $50$ K training iterations.
>
> | **Model (L=128)** | CEDD\* Uniform | SEDD Uniform | CEDD\* Roulette | SEDD Roulette |
> |-------------------|--------------|--------------|---------------|---------------|
> | PAP               | 90.7         | 89.9         | **90.8**      | 87.9          |
> | CAP               | 91.2         | 90.3         | **91.3**      | 88.6          |
>
> ---
>
> __Line 147: $\bf{x}^i_0 = x^i_0$ , is it a typo?__
>
> This is actually redundant and has now been removed from the paper. We thank the reviewer for pointing this out.
>
> If the reviewer has additional concerns we would be happy to address them.
>
> __References__
>
> [1] Speculative Diffusion Decoding: Accelerating Language Generation through Diffusion
>
> [2] Argmax Flows and Multinomial Diffusion: Learning Categorical Distributions, Hoogeboom et al.

---

### Meta-Review · Area_Chair_4dvh · 2024-12-22

**Metareview:**

> This paper presents new results for discrete diffusion models trying to estimate density ratios for language modeling task. The authors' contributions are threefold: (i) they provide an upper bound  on the perplexity of discrete ratio-matching diffusion language models which is computationally more efficient than the bound  known from the literature, and, as shown through empirical evaluation, is consistently tighter; (ii) they derive a novel training objective called cross-entropy discrete diffusion (CEDD) loss by making use of reparametrization of concrete scores; (iii) they propose an alternative to known transition-rate matrices referred to as "absorb" and "uniform" and derive exponential of this new "roulette" transition matrix necessary to compute conditional probabilities efficiently.

Reviewer wpbF (marginally below) discussed 3 weaknesses of the original paper that the authors addressed. With 2 strong accepts, I lean accept.

**Additional Comments On Reviewer Discussion:**

The authors took advantage of the rebuttal period to improve their paper.

---

### Decision · Program_Chairs · 2025-01-22

Accept (Poster)